# Mechanical confinement governs phenotypic plasticity in melanoma

Miranda V. Hunter[1✉], Eshita Joshi[2], Sydney Bowker[3], Emily Montal[1], Yilun Ma[1,4,5], Young Hun Kim[6], Zhifan Yang[2], Laura Tuffery[7], Zhuoning Li[7], Eric Rosiek[6], Alexander Browning[8], Reuben Moncada[9], Itai Yanai[9], Helen Byrne[8], Mara Monetti[7], Elisa de Stanchina[3], Pierre-Jacques Hamard[2], Richard P. Koche[2] & Richard M. White[1,10✉]

Phenotype switching is a form of cellular plasticity in which cancer cells reversibly move between two opposite extremes: proliferative versus invasive states[1,2]. Although it has long been hypothesized that such switching is triggered by external cues, the identity of these cues remains unclear. Here we demonstrate that mechanical confinement mediates phenotype switching through chromatin remodelling. Using a zebrafish model of melanoma coupled with human samples, we profiled tumour cells at the interface between the tumour and surrounding microenvironment. Morphological analysis of interface cells showed elliptical nuclei, suggestive of mechanical confinement by the adjacent tissue. Spatial and single-cell transcriptomics demonstrated that interface cells adopted a gene program of neuronal invasion, including the acquisition of an acetylated tubulin cage that protects the nucleus during migration. We identified the DNA-bending protein HMGB2 as a confinement-induced mediator of the neuronal state. HMGB2 is upregulated in confined cells, and quantitative modelling revealed that confinement prolongs the contact time between HMGB2 and chromatin, leading to changes in chromatin configuration that favour the neuronal phenotype. Genetic disruption of HMGB2 showed that it regulates the trade-off between proliferative and invasive states, in which confined HMGB2[high] tumour cells are less proliferative but more drug-resistant. Our results implicate the mechanical microenvironment as a mechanism that drives phenotype switching in melanoma.

The ability of cancer cells to adopt new phenotypes without further DNA mutations is now well understood to substantially influence tumour behaviour. Such plasticity has long been observed in melanoma, where early studies identified transcriptomic and phenotypic states not linked to specific genetic lesions[1]. More recent evidence indicates that most tumours encompass a heterogeneous yet reproducible number of transcriptional states[3,4]. The extent to which tumour cells transition between states is an open area of investigation and has been hypothesized to be regulated by cues from the tumour microenvironment (TME). The identification of such cues has clinical relevance because they may enable the conversion of a superficial melanoma into an invasive and drug-resistant one[5]. Here using a combination of zebrafish transgenics and human samples, we show that mechanical confinement by the adjacent microenvironment induces stable changes in chromatin architecture that cause melanoma cells to transition from a proliferative to an invasive state.

## Tumour gene expression altered by TME

To study the influence of the local microenvironment on tumour invasion, we applied spatially resolved transcriptomics and single-cell RNA sequencing (scRNA-seq) to a transgenic zebrafish model of $BRAF^{V600E}$-driven melanoma (Fig. 1a). Tumours from this model frequently invade into adjacent tissues, including the underlying dermis and muscle. Across fish, we found a conserved 'interface' transcriptional cell state that occurs where tumour cells invade into the microenvironment[6] (12.1% of zebrafish tumour cells; Fig. 1a).

To investigate whether these interface cells occur in patients, we analysed a recently published human melanoma scRNA-seq dataset of 7,186 tumour and stromal cells from 31 patients with untreated or immunotherapy-resistant melanoma[7] (Fig. 1b and Extended Data Fig. 1a). We scored tumour cells for the relative expression of interface genes from our zebrafish transcriptomic datasets. Similar to

[1]Cancer Biology and Genetics Program, Memorial Sloan Kettering Cancer Center, New York, NY, USA. [2]Center for Epigenetics Research, Memorial Sloan Kettering Cancer Center, New York, NY, USA. [3]Antitumor Assessment Core Facility, Memorial Sloan Kettering Cancer Center, New York, NY, USA. [4]Weill Cornell/Rockefeller/Sloan Kettering Tri-Institutional MD-PhD Program, Memorial Sloan Kettering Cancer Center, New York, NY, USA. [5]Cell and Developmental Biology Program, Weill Cornell Graduate School of Medical Sciences, New York, NY, USA. [6]Molecular Cytology Core Facility, Memorial Sloan Kettering Cancer Center, New York, NY, USA. [7]Proteomics Core Facility, Memorial Sloan Kettering Cancer Center, New York, NY, USA. [8]Mathematical Institute, University of Oxford, Oxford, UK. [9]Institute for Systems Genetics, NYU Langone Health, New York, NY, USA. [10]Nuffield Department of Medicine, Ludwig Institute for Cancer Research, University of Oxford, Oxford, UK. ✉e-mail: hunterm@mskcc.org; richard.white@ludwig.ox.ac.uk

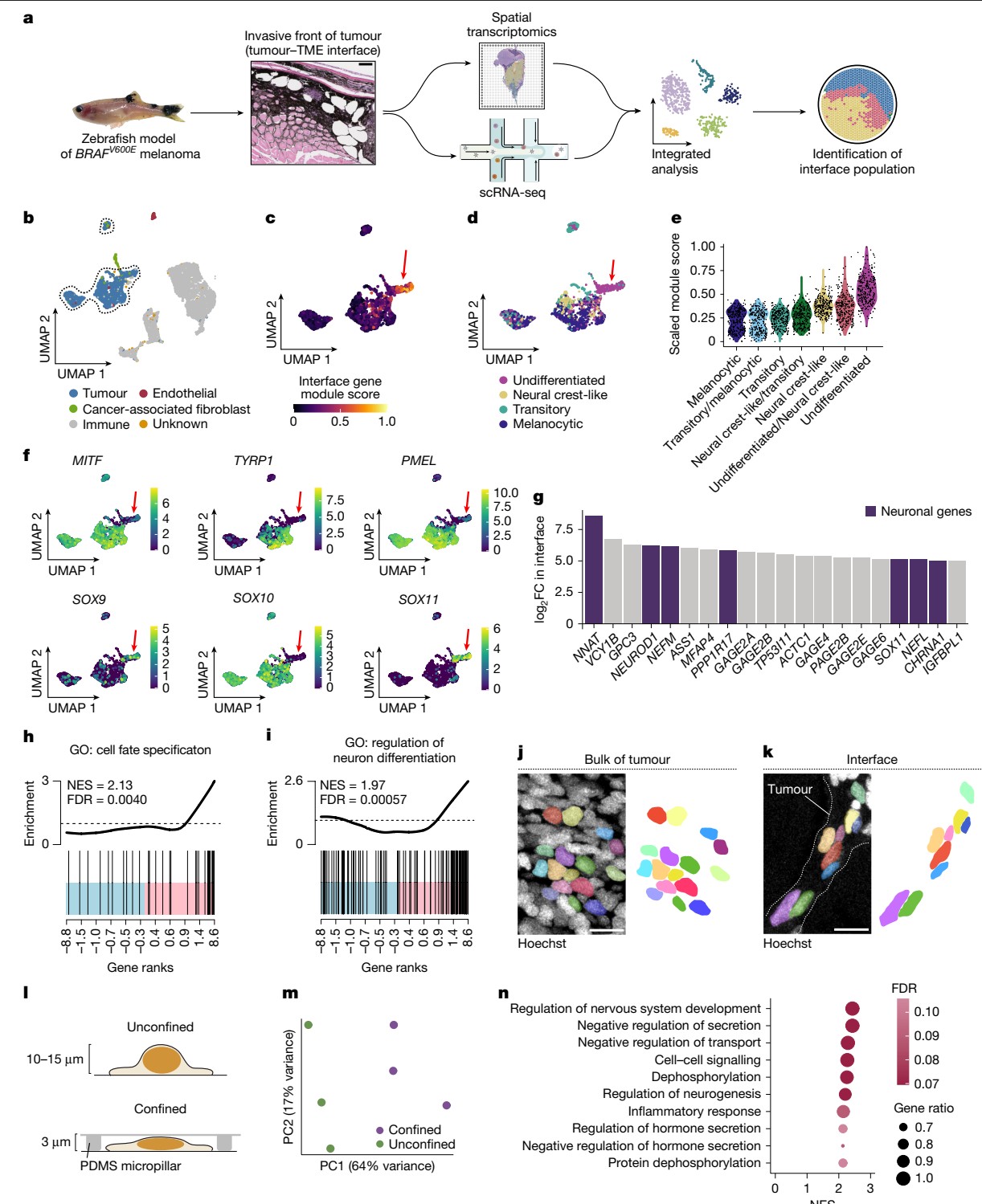

**Fig. 1 | Confinement induces an undifferentiated neuronal gene program.**
**a**, Schematic detailing the workflow of spatial transcriptomics and scRNA-seq experiments performed on zebrafish melanomas. **b**, Uniform manifold approximation and projection (UMAP) of human melanoma scRNA-seq dataset from Jerby-Arnon et al.[7]. Cluster annotations from the original paper are labelled. Tumour cell clusters are outlined. **c**, Gene module scoring for interface genes extracted from zebrafish spatial transcriptomics and scRNA-seq data, projected onto tumour cells outlined in **b**. The red arrow denotes the subpopulation with the highest expression of interface genes. **d**, Cell state classification for melanoma differentiation states identified by Tsoi et al.[2]. Cells were classified on the basis of the highest expression of the gene modules indicated. **e**, Module scores for melanoma cell state genes from Tsoi et al.[2] in interface cells. **f**, Normalized expression per cell in UMAP space for the indicated genes. The red arrow indicates

the interface cluster identified in **b**. **g**, Top 20 most highly upregulated genes in the human interface cluster. Neuronal genes are labelled in purple. **h,i**, GSEA barcode plot for the Gene Ontology (GO) pathways 'cell fate specification' (**h**) and 'regulation of neuron differentiation' (**i**). Normalized enrichment score (NES) and false discovery rate (FDR) are labelled. **j,k**, Immunofluorescence of adult zebrafish tissue sections highlighting the centre of the tumour (**j**) and tumour–TME interface (**k**). Individual nuclei are pseudocoloured and displayed without image overlay at right. **l**, Schematic of in vitro confinement workflow using a polydimethylsiloxane (PDMS) piston. **m**, Principal component analysis plot for each RNA sequencing (RNA-seq) replicate. Percentage variance for each principal component (PC) is labelled. $n = 3$ biological replicates for each condition. **n**, Top 10 most highly upregulated pathways from GSEA of confined cells relative to unconfined cells. NES and FDR are indicated. Scale bars, 100 μm (**a**), 10 μm (**j**,**k**).

our previous observations[6], a subpopulation of human tumour cells highly upregulated interface markers (12.3% of tumour cells; Fig. 1c and Extended Data Fig. 1b), mostly from immunotherapy-resistant patients (Extended Data Fig. 1d–f). To better understand the nature of these cells, we compared our interface population to human melanoma cell states. A previous study defined at least four cell states that encompass the melanoma differentiation trajectory (melanocytic, transitory, neural crest-like and undifferentiated)[2]. We scored tumour and interface cells for the relative expression of gene modules encompassing the entire melanoma cell differentiation trajectory and classified cells into each of the four states on the basis of their expression of gene modules annotated for that state[2]. Although tumour cells were relatively evenly distributed between the four cell states, interface cells showed clear upregulation of genes characteristic of the undifferentiated state (Fig. 1d,e and Extended Data Fig. 1c).

Melanoma behaviour is regulated by phenotype switching, in which cells transition between differentiated/proliferative and undifferentiated/invasive states. The transition between the proliferative and invasive states is regulated by the transcription factors *MITF*, *SOX9* and *SOX10*, among others[8]. *MITF* and *SOX10* regulate melanocyte differentiation and proliferation[9,10], whereas *SOX9* is associated with the undifferentiated invasive state[11]. Interface cells downregulated *MITF* and *SOX10* and upregulated *SOX9* (Fig. 1f and Extended Data Fig. 1g). Classical melanocyte pigmentation genes were minimally expressed by interface cells relative to the tumour bulk, including *MITF* ($\log_2$ fold change (FC) = −5.19), *TYRP1* ($\log_2 FC$ = −7.75) and *PMEL* ($\log_2 FC$ = −8.78) (Fig. 1f, Extended Data Fig. 1g and Supplementary Table 1). Unexpectedly, interface cells also upregulated genes involved in neuronal development, including *SOX11* ($\log_2 FC$ = 5.15), *NNAT* ($\log_2 FC$ = 8.61), *NEUROD1* ($\log_2 FC$ = 6.22) and *NEFM* ($\log_2 FC$ = 6.19) (Fig. 1g and Supplementary Table 1). Gene set enrichment analysis (GSEA) revealed that transcriptional programs linked to cell fate specification and neuronal development were highly enriched in interface cells (Fig. 1h,i). These data indicate that interface cells adopt an invasive state with markers of neuronal development.

## Confinement induces a neuronal state

To examine factors within the local TME that may drive the interface state, we performed histology on tissue sections from our transgenic zebrafish melanoma model, focusing on the invasive front. Tumour cells invading into the TME showed elongated nuclei compared with the bulk tumour mass (Fig. 1j,k). A previous study showed that tumour nuclei become highly elongated when squeezing through mechanically restrictive environments[12]. Although numerous factors probably influence tumour invasion, we hypothesized that mechanical forces exerted on the cell/nucleus may cause stable changes in gene expression and tumour cell behaviour.

To test this hypothesis, we adapted a system to confine human melanoma cells (A375 cell line) in vitro at predefined heights (3 μm) using a polydimethylsiloxane piston and micropatterned coverslips[13] (Fig. 1l). Confinement in vitro was not cytotoxic because confined cells did not upregulate apoptosis markers cleaved caspase-3, annexin V or cleaved PARP (Extended Data Fig. 2a–f). Post-confinement, cells recovered typical morphology within 24 h without widespread cell death (Extended Data Fig. 2g). To profile confinement-induced changes in gene expression, we confined A375 cells for approximately 18 h and performed bulk RNA-seq. Principal component analysis revealed considerable transcriptional alterations induced by confinement (Fig. 1m and Supplementary Table 2). Similar to human interface cells, confined cells upregulated several neuronal pathways (Fig. 1n and Supplementary Table 2). We calculated the overlap between the human interface gene signature and the confined melanoma gene signature (Extended Data Fig. 2h and Supplementary Table 3). Pathway analysis showed that many of the co-regulated genes were related to neuronal development

(Extended Data Fig. 2i and Supplementary Table 3). These data indicate that confinement causes interface cells to adopt a neuronal identity.

## Confinement remodels the cytoskeleton

Neuronal development relies on microtubule (MT) architecture, influencing almost every aspect of neuronal structure and function[14]. The cytoskeleton often transmits force within and between cells and remodels in response to mechanical stimuli[15]. Recent reports indicate that the MT cytoskeleton is stabilized by force to protect confined cells from damage[16]. We hypothesized that melanoma cells hijack neuronal mechanisms to allow them to invade into the mechanically confined microenvironment. We used our in vitro confinement system to characterize how force remodels the MT cytoskeleton by imaging A375 cells labelled with the MT vital dye SiR-tubulin. Confinement extensively remodelled the MT cytoskeleton; within 2–4 h, curved MTs began encircling the cell and nuclear periphery (Fig. 2a,b). This was reminiscent of a previous study that showed that MTs bend to prevent buckling or rupture in response to force[16]. We also observed loss of a central MT organizing centre with radial MTs, reminiscent of neurons with centrosome-independent MT organization[14,17]. These results indicate that confined melanoma cells rapidly undergo structural changes in the MT cytoskeleton resembling neurons.

Acetylated tubulin mediates neuronal architecture and function and marks axonal MTs[18]. Tubulin acetylation stabilizes MTs and thus the cell against mechanical pressure[19]. Curved MTs, such as those we observed in confined cells (Fig. 2a,b), are also indicative of long-lived, stabilized MTs[19]. We performed immunofluorescence on sections from adult fish with transgenic *BRAF*[V600E]-driven melanomas and observed enrichment of acetylated tubulin at the tumour border[6] (Fig. 2c–e). Confining A375 cells in vitro also resulted in significant upregulation of acetylated tubulin ($P = 6.86 \times 10^{-12}$; Fig. 2f–h). Acetylated tubulin filaments in unconfined cells were typically short and linear, whereas in confined cells, acetylated tubulin filaments were longer and more curved—again indicative of stabilization (Fig. 2f,g).

We noticed that in confined cells, the hyperacetylated tubulin network was often perinuclear (Fig. 2f,g) and hypothesized that this network may provide structural support to the nucleus. As the stiffest and largest organelle, the nucleus is vulnerable to confinement-induced stress, with confined migration often causing nuclear envelope rupture and DNA damage[20,21]. During migration through confined spaces, neurons assemble a perinuclear network of acetylated tubulin to protect the nucleus[22,23]. We confirmed the stability of the perinuclear MT network by treating confined cells with nocodazole, which induces MT disassembly through an acetylation-independent mechanism[24]. Acetylated MTs are resistant to nocodazole[19]. Nocodazole (1 μM) induced the disassembly of almost all non-modified tubulin filaments in both unconfined and confined cells (Fig. 2i,j). Although unconfined nocodazole-treated cells exhibited minimal acetylated tubulin, many confined nocodazole-treated cells contained a perinuclear acetylated tubulin cage (Fig. 2i,j). We generated a stable A375 cell line, in which the main eukaryotic tubulin acetyltransferase (ATAT1) was inactivated using CRISPR (Extended Data Fig. 3a,b), and found that this almost completely abolished acetylated tubulin, including the perinuclear network (Extended Data Fig. 3c–e). This indicates that ATAT1 is the acetyltransferase that responds to mechanical stress in melanoma cells by stabilizing the tubulin cytoskeleton.

In addition to acetylation, MTs exhibit a variety of post-translational modifications (PTMs), including tyrosination, glutamylation, glycylation and methylation, which influence MT stability[25]. To clarify the composition of the perinuclear tubulin network, we used immunofluorescence to characterize MT PTMs in confined melanoma cells. Detyrosination has been linked to stabilized MTs[26]. However, most MTs in confined and unconfined A375 cells were highly tyrosinated (Extended Data Fig. 4a–d), probably owing to the high concentrations of tyrosine and tyrosinase

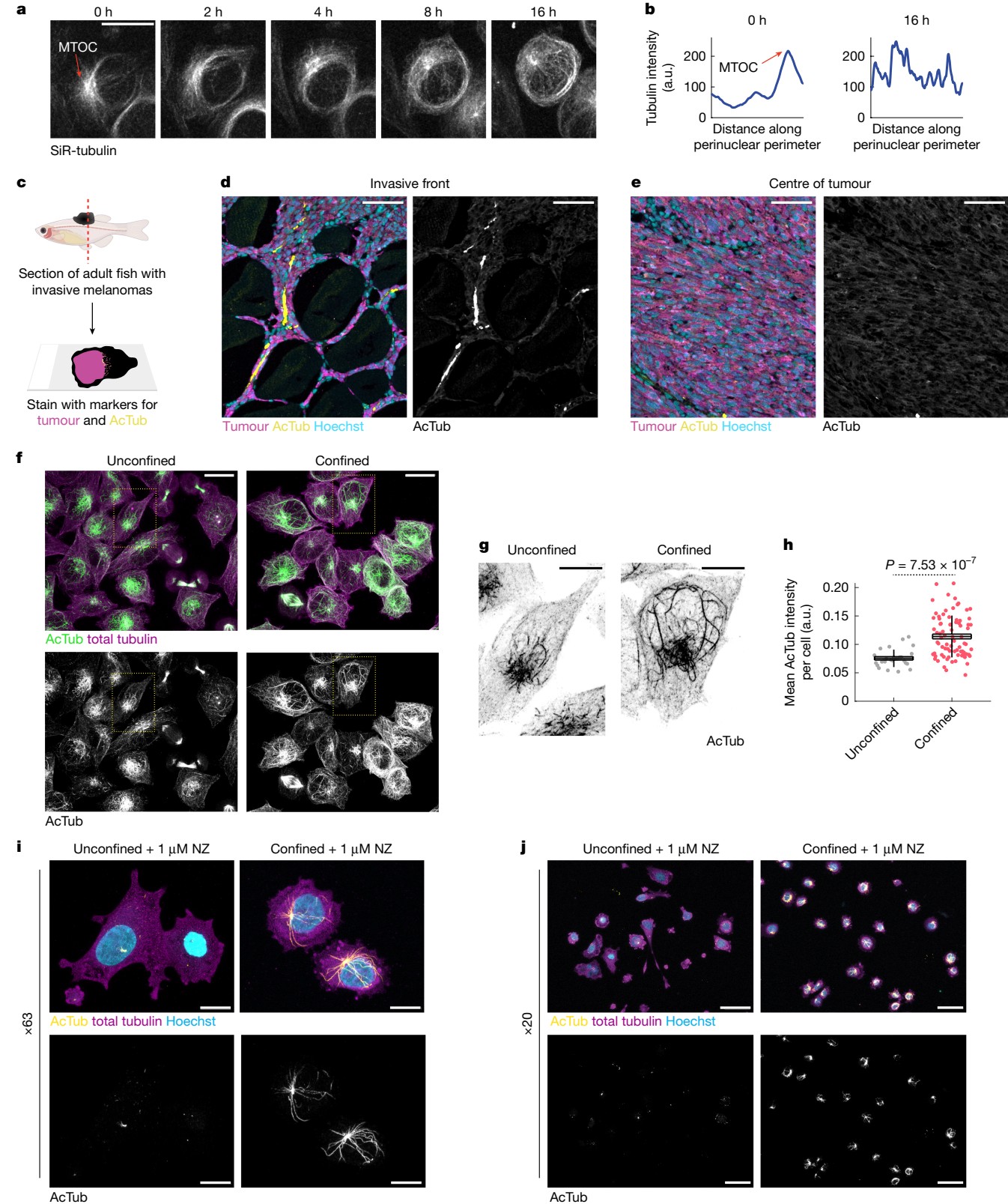

**Fig. 2 | Perinuclear acetylated tubulin cage assembles in response to confinement. a**, Representative stills from confocal imaging of A375 cells stained with SiR-tubulin. MTOC, microtubule-organizing centre. **b**, Line intensity profile of perinuclear tubulin intensity over time from the images shown in **a**. MTOC is highlighted (**a**,**b**). a.u., arbitrary units. **c**, Schematic detailing immunofluorescence staining of sections from adult zebrafish melanomas. **d**,**e**, Immunofluorescence images of acetylated tubulin staining at the invasive front (**d**) compared with the centre of the tumour (**e**). **f**, A375 cells stained with antibodies labelling acetylated tubulin (green) and total tubulin (purple). **g**, Inset of regions labelled in **f**. **h**, Quantification of whole-cell acetylated tubulin intensity. Each point represents one cell. Unconfined, $n = 27$ cells from three images; confined, $n = 80$ cells from nine images. Horizontal lines, mean; box, s.e.m.; vertical lines, s.d. $P$ value is indicated (two-sample $t$-test; two-sided). **i**,**j**, A375 cells treated with 1-µm nocodazole (NZ) for approximately 18 h and stained for acetylated tubulin (yellow, top and bottom), total tubulin (purple) and Hoechst (blue) at ×63 (**i**) and ×20 (**j**) magnification. Scale bars, 50 µm (**d**,**e**,**j**), 20 µm (**a**,**f**), 10 µm (**g**,**i**). Illustrations in **c** were created using BioRender (https://biorender.com).

in melanoma[27]. Although confined perinuclear MTs were occasionally tyrosinated and rarely detyrosinated (Extended Data Fig. 4a–d), there was no specific enrichment of these PTMs in the perinuclear network, as observed for acetylated MTs. Similar results were observed for polyglutamylated MTs (Extended Data Fig. 4e,f). No polyglycylated MTs were found in either condition (Extended Data Fig. 4g,h). This indicates that acetylation is the primary PTM mediating the assembly and/or stability of the perinuclear tubulin network. These data indicate that confined melanoma cells assemble a stable perinuclear tubulin network to reinforce the nucleus against mechanical stress, similar to neurons.

## Confined invasive cells upregulate HMGB2

Our results indicate that confinement induces a neuronal identity in interface melanoma cells, characterized by changes in cytoskeletal architecture and gene expression. We examined our zebrafish and human transcriptomic datasets to identify potential confinement-induced mediators of this state. Interface cells consistently upregulated high mobility group (HMG)-family proteins (Fig. 3a), which regulate chromatin architecture by binding and bending DNA to relieve mechanical strain[28]. Although HMG-enriched transcriptional programs are upregulated across many tumour types, including melanoma[29], their contribution to tumour progression remains poorly understood. We focused on *HMGB2* (zebrafish *hmgb2a* and *hmgb2b*) because it was the most upregulated HMG family member in interface cells from zebrafish (Fig. 3a,b), and because our previous study identified HMGB2 as a signalling factor enriched in interface cells[6]. *HMGB2* was also highly upregulated by human interface cells ($P = 3.24 \times 10^{-37}$; Fig. 3c,d and Extended Data Fig. 1h).

To validate our transcriptomic results indicating that HMGB2 is upregulated at the invasive front, we examined HMGB2 expression in zebrafish melanoma tissue sections. In invading tumour cells, HMGB2 was only upregulated in elongated or misshaped tumour cells that appeared to be under mechanical pressure from adjacent tissues (Fig. 3e–g), indicating that confinement induces HMGB2 upregulation. HMGB2 concentrations were inversely correlated to nuclear circularity ($R = -0.474$; $P = 1.19 \times 10^{-12}$; Fig. 3g). Similarly, the in vitro confinement of A375 human melanoma cells caused nuclear HMGB2 intensity to approximately double relative to unconfined cells ($P = 1.42 \times 10^{-7}$; Fig. 3h,i). This increase was not solely attributable to changes in nuclear density; although Hoechst intensity also slightly increased upon confinement (probably owing to compaction of chromatin/the nucleus; $P = 0.0017$; Fig. 3j), normalizing HMGB2 to Hoechst concentrations still showed a significant increase ($P = 3.01 \times 10^{-6}$; Fig. 3k). Using time-lapse confocal microscopy of A375 cells stably expressing HMGB2–GFP, we found nuclear HMGB2–GFP concentrations increased linearly over approximately 16 h ($0.269 \pm 0.020$-fold per hour) to a final concentration of $1.76 \pm 0.072$-fold relative to initial concentrations (Extended Data Fig. 5a,b). We also examined the concentrations of HMG family members HMGB1 and HMGA1, which were transcriptionally upregulated in interface cells to a lesser extent than HMGB2, and quantified no change in their expression upon confinement (Extended Data Fig. 5c–f), indicating that confinement-induced upregulation of HMGB2 is not a general property of all HMG family members.

We examined a human melanoma tissue microarray to look for evidence of interface cells with elongated nuclei, high HMGB2 concentrations and perinuclear acetylated tubulin. As in the zebrafish, the invasive front in human samples often contained elongated nuclei with high HMGB2 and acetylated tubulin expression, although with expected interpatient variability (Extended Data Fig. 6a–e). Of the 40 patient samples analysed, 20 (50%) contained putative interface cells (elongated nuclei, HMGB2[+] and AcTub[+]), nine (22.5%) contained cells exhibiting perinuclear acetylated tubulin enrichment only (AcTub[+] and HMGB2[−]), three (7.5%) contained cells with elongated HMGB2[+] nuclei but no acetylated tubulin (AcTub[−] and HMGB2[+]) and the remaining

eight (20%) showed no enrichment of HMGB2 or acetylated tubulin (Extended Data Fig. 6d,e). This validates our scRNA-seq analyses, indicating that interface cells are present in human samples (Fig. 1c).

We then investigated whether the upregulation of HMGB2 and acetylated tubulin by confined cells is melanoma-specific or exhibited across other cancer types. We focused on two cancer types probably influenced by mechanical stress in vivo: pancreatic ductal adenocarcinoma and bladder cancer. In both, confinement induced significant upregulation of HMGB2 and acetylated tubulin (Extended Data Fig. 6f–q).

Mechanical stress can be associated with nuclear translocation of transcription factors, such as YAP, as well as epithelial–mesenchymal transition inducers, including Twist, Snail and SMAD3 (ref. 30). To rule out a potential pro-invasive role for these transcription factors upon mechanical stress in melanoma, we quantified their expression in confined A375 cells. None of the factors examined (YAP, Twist, Snail and SMAD3) exhibited nuclear translocation upon confinement, and the nuclear expression of YAP, Twist and SMAD3 decreased (Extended Data Fig. 7).

## HMGB2 upregulation requires nesprin 2

To investigate how mechanical confinement induces HMGB2 upregulation, we focused on the MT cytoskeleton, hypothesizing that the perinuclear acetylated tubulin network (Fig. 2f,g) may propagate confinement-induced force to the nucleus. To confirm that the perinuclear network is upstream of HMGB2 enrichment in the confinement response, we generated stable A375 cell lines in which HMGB2 was inactivated using CRISPR (Extended Data Fig. 8a–c). Upon confining two different A375–HMGB2[KO] cell lines, we confirmed that the perinuclear acetylated tubulin network was unaffected (Extended Data Fig. 8d,e).

We modulated tubulin dynamics and acetylation state in confined cells to determine whether the acetylated tubulin network influences HMGB2 upregulation. Tubulin acetylation is mediated by the acetyltransferase ATAT1 (ref. 31) and deacetylation by HDAC6 (ref. 32). We first treated A375 cells with the HDAC6 inhibitor tubacin[33], which increased tubulin acetylation without affecting histone acetylation (Extended Data Fig. 9a–f). In confined cells, tubacin treatment significantly increased both total nuclear HMGB2 accumulation ($P = 1.1646 \times 10^{-6}$; Extended Data Fig. 9g,h) and the rate of accumulation, which approximately doubled versus controls ($0.353 \pm 0.0364$-fold per hour versus $0.145 \pm 0.0171$-fold per hour; $P = 1.1589 \times 10^{-6}$; Extended Data Fig. 9i). To stabilize MTs independent of acetylation, we treated confined cells with paclitaxel (Taxol), which binds β-tubulin and prevents tubulin monomer incorporation into MT filaments and thus should not influence the acetylation state of HMGB2 or other proteins[34]. Taxol (100 nM) significantly increased HMGB2 accumulation in confined cells ($P = 0.00230$; Extended Data Fig. 9g,h), with an accumulation rate remarkably similar to tubacin-treated cells ($0.338 \pm 0.0425$-fold per hour versus $0.353 \pm 0.0364$-fold per hour; $P = 0.961$; Extended Data Fig. 9i). Although we cannot rule out off-target effects of tubacin on HMGB2, similar results in Taxol-treated and tubacin-treated cells indicate that confinement-induced HMGB2 enrichment is linked to MT stability, rather than acetylation state or other HDAC6 functions.

To confirm the role of perinuclear acetylated tubulin in mediating HMGB2 accumulation, we treated A375[HMGB2–GFP] cells with nocodazole before applying confinement. Nocodazole abolishes most MT networks, with the exception of stabilized acetylated tubulin filaments[24]. Because the perinuclear acetylated tubulin cage was resistant to nocodazole (Fig. 2i,j), nocodazole treatment allowed us to specifically interrogate the contribution of acetylated MTs to HMGB2 upregulation in confined cells. We observed near total loss of visible SiR-tubulin signal in confined nocodazole-treated cells (Extended Data Fig. 9j), whereas the perinuclear acetylated tubulin network remained intact (Fig. 2i,j) and HMGB2–GFP accumulation was unaffected ($P = 0.714$; Extended Data Fig. 9k,l). We quantified HMGB2 accumulation in confined A375–ATAT1[KO] cells lacking acetylated tubulin (Extended Data Fig. 3c–e).

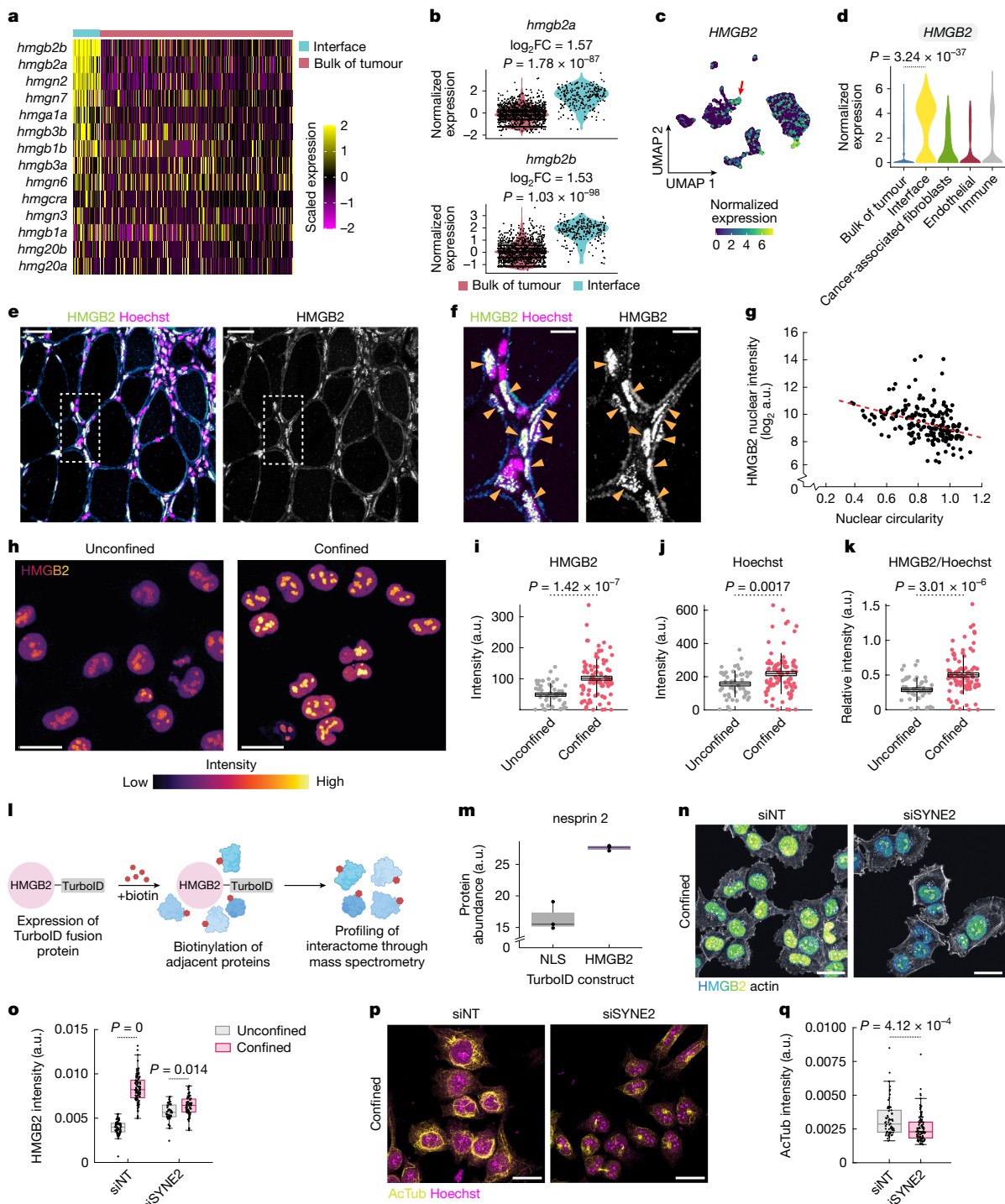

**Fig. 3 | HMGB2 is a confinement-induced marker of invasion. a**, HMG family expression in interface cells from zebrafish melanoma scRNA-seq. **b**, Normalized *hmgb2a/hmgb2b* expression. *P* values are noted (Wilcoxon rank-sum test; two-sided). **c**, *HMGB2* expression per cell in human melanoma scRNA-seq data from Jerby-Arnon et al.[7]. The arrow indicates interface cluster. **d**, Mean *HMGB2* expression per cluster. *P* value calculated using Wilcoxon rank-sum test with Bonferroni's correction; two-sided. **e**, Zebrafish melanoma stained for HMGB2 and Hoechst. **f**, Inset of region indicated in **d**. Elongated HMGB2-high cells are labelled. **g**, Correlation between nuclear circularity and HMGB2 intensity. Red dashed line, line of best fit by linear regression. **h**, Immunofluorescence targeting HMGB2 in confined A375 cells. **i–k**, HMGB2 intensity (**i**), Hoechst intensity (**j**) and HMGB2 intensity normalized to Hoechst (**k**) per cell. Unconfined, *n* = 49 cells from three images; confined, *n* = 97 cells from nine images. Horizontal lines, mean; box, s.e.m.; vertical lines, s.d. *P* value is indicated (two-sample *t*-test; two-sided). **l**, TurboID workflow. **m**, Nesprin 2 protein abundance; *n* = 3 replicates per condition. Horizontal line, median; hinges, first and third quartiles; whiskers, range. NLS, nuclear localization signal. **n**, HMGB2 expression in confined A375 cells. siNT, non-targeting siRNA; siSYNE2, *SYNE2*-targeting siRNA. **o**, Quantification of HMGB2 intensity. siNT unconfined, *n* = 72 cells from eight images; siNT confined, *n* = 94 cells from eight images; siSYNE2 unconfined, *n* = 48 cells from eight images; siSYNE2 confined, *n* = 64 cells from eight images. *P* value is indicated (analysis of variance with Tukey post hoc test; two-sided). **p**, Images showing acetylated tubulin (yellow) and Hoechst (magenta) in confined A375 cells. **q**, Quantification of acetylated tubulin intensity in confined cells. siNT, *n* = 66 cells from eight images; siSYNE2, *n* = 104 cells from eight images. *P* value is indicated (two-sample *t*-test; two-sided). Horizontal line, median; edges, upper and lower quartiles; whiskers, non-outlier minima and maxima (**o**,**q**). Scale bars, 50 μm (**e**), 10 μm (**f**), 25 μm (**h**,**n**,**p**). Illustrations in **l** were created using BioRender (https://biorender.com).

Unexpectedly, HMGB2 accumulation was not impaired in ATAT1[KO] cells (Extended Data Fig. 3f), suggesting that acetylated tubulin is sufficient but not necessary for the enrichment of HMGB2 in response to confinement.

Thus, we investigated other factors that may cooperate with the MT cytoskeleton to promote upregulation of HMGB2 in confined cells using TurboID[35] to perform proximity labelling proteomics targeting HMGB2 interactors (Fig. 3l and Supplementary Table 4). One highly enriched protein was nesprin 2 (gene name: *SYNE2*), a component of the linker of nucleoskeleton and cytoskeleton (LINC) complex that connects the cytoskeleton, nuclear lamina and chromatin[36] (Fig. 3m and Supplementary Table 4). We hypothesized that nesprin 2 may be required for enrichment of HMGB2 upon confinement. Accordingly, confined A375 cells upregulated nesprin 2 (Extended Data Fig. 10a,b), and targeting *SYNE2* with short interfering RNA (siRNA) (Extended Data Fig. 10c) abolished the confinement-mediated accumulation of HMGB2 (Fig. 3n,o) and perinuclear tubulin network (Fig. 3p,q). This indicates that nesprin 2 and the tubulin cytoskeleton interact to upregulate HMGB2.

The LINC complex cooperates with the nuclear lamina to tune nuclear stiffness[36]. Lamin A/C increased by approximately 3-fold in confined cells ($P = 1.28 \times 10^{-162}$; Extended Data Fig. 10d,e), suggesting that the nuclear lamina was remodelled in response to confinement, which we validated by means of atomic force microscopy (AFM) showing increased nuclear stiffness in confined cells (Extended Data Fig. 10f,g). Together, our results demonstrate that confined melanoma cells remodel cytoskeletal and nuclear structures to reinforce the cell against mechanical force, resulting in LINC complex-mediated HMGB2 upregulation and nuclear stiffening.

## Force affects HMGB2–chromatin dynamics

Our results indicate that the LINC complex and tubulin cytoskeleton cooperate to stabilize the nucleus against mechanical stress and upregulate HMGB2. HMGB2 typically binds chromatin without sequence specificity and bends DNA to relieve mechanical strain[37]. Confinement-induced nuclear shape changes could thus increase the strain on chromatin and create a densely packed nuclear environment that affects HMGB2 dynamics. We used fluorescence recovery after photobleaching (FRAP) to characterize nuclear HMGB2 dynamics upon confinement. As previously reported[38], HMGB2–GFP was highly dynamic even in unconfined cells (Fig. 4a–d), with only approximately 25% stably bound within the nucleus (mobile fraction = 75.21 ± 0.91%). In both conditions, FRAP recovery curves fitted well to a two-component exponential equation (average coefficient of determination ($R^2$) = 0.986 ± 0.00054; Fig. 4d,e and Methods), indicating two pools of nuclear HMGB2: fast-diffusing (stochastic interactions with chromatin) and slower-diffusing (more specific, stable interactions)[39]. Although more than 90% of HMGB2 remained fast-diffusing in both conditions, confinement significantly increased the proportion of slower-diffusing HMGB2 (6.76 ± 0.39% versus 4.21 ± 0.39%; $P = 1.134 \times 10^{-5}$; Fig. 4f), suggesting more specific, stable HMGB2–chromatin interactions in confined cells.

## HMGB2 targets plasticity-associated loci

Our FRAP analysis indicates that confinement upregulates HMGB2 and stabilizes its interactions with chromatin; therefore, we propose that this could affect chromatin accessibility. Assay for transposase-accessible chromatin (ATAC) sequencing (ATAC-seq) of A375 cells overexpressing *HMGB2* showed broadly increased chromatin accessibility (Fig. 4g). Peak-gene mapping and pathway analysis revealed that open chromatin regions were highly enriched for neuronal genes, with increased accessibility at loci linked to the neural crest and neuronal development (Fig. 4h–j and Supplementary Table 5). Homer de novo motif analysis identified conserved transcription factor binding motifs enriched

within the promoter of open chromatin regions upon *HMGB2* overexpression (HMGB2[OE]). The top-ranked motif was an *AP-1* motif (Fig. 4k; $P = 1 \times 10^{-49}$), implicated in melanoma and melanocyte plasticity[40,41]. Other highly ranked motifs included *PRDM4*, which functions in neural development[42] (Fig. 4k; $P = 1 \times 10^{-12}$), and *SOX9*, a critical regulator of the pro-invasive phenotype switch in melanoma[43] (Fig. 4k; $P = 1 \times 10^{-11}$). These data indicate that HMGB2 upregulation increases chromatin accessibility at loci associated with plasticity and a neuronal phenotype, promoting a confinement-induced dedifferentiation program.

To clarify potential HMGB2 targets, we performed chromatin immunoprecipitation (ChIP) sequencing (ChIP–seq) targeting HMGB2 in A375 cells using a double-crosslinking approach[44] (Extended Data Fig. 11a). We generated a peak atlas by comparing peaks present in cells expressing baseline concentrations of HMGB2 to those not present in HMGB2[KO] cells, yielding 843 peaks, consistent with previous studies[44–46] (Supplementary Table 6). After manual filtering to remove intergenic, low-quality or non-specific peaks, we identified a final high-confidence set of 96 targets (Extended Data Fig. 11b and Supplementary Table 6). We observed HMGB2 binding at promoter regions of several AP-1 signalling genes, including *FOSL1*, *JUNB* and *JUND* (Extended Data Fig. 11c and Supplementary Table 6), validating our ATAC-seq results and supporting the pro-invasive role of HMGB2. Several neuronal genes were also identified as HMGB2 targets, including *NOTCH2*, *NOTCH2NLC*, *TBX6*, *GBA1* and *ZNF335* (Extended Data Fig. 11d and Supplementary Table 6), as well as pro-tumorigenic genes such as *KMT2A* (Extended Data Fig. 11e and Supplementary Table 6).

To examine the transcriptional effects, we performed bulk RNA-seq on A375 cells stably overexpressing *HMGB2* (Extended Data Fig. 11f). HMGB2[OE] cells adopted a mesenchymal-like and invasive morphology relative to empty vector controls (Extended Data Fig. 11g). The most highly upregulated gene was *UNC5D*, a netrin receptor promoting neuronal survival and migration[47] (Extended Data Fig. 11h and Supplementary Table 7). Other upregulated neuronal genes included *DCC*, *SH3GL2* and *NPX2* (Extended Data Fig. 11h,i and Supplementary Table 7). GSEA revealed an overrepresentation of neuronal genes within enriched pathways (Extended Data Fig. 11j,k and Supplementary Table 7). Supporting a pro-invasive role for HMGB2, several invasive genes were also upregulated (*MAGEA1*, *SPP1*, *CTAG2* and *GDF6*; Extended Data Fig. 11i and Supplementary Table 7). These data indicate that HMGB2 alters chromatin architecture to promote an invasive neuronal state.

## HMGB2 drives invasion through Notch and BRN2

Phenotypic plasticity in melanoma is controlled by a regulatory axis of melanocytic transcription factors (*MITF*) driving proliferation versus invasive factors (*BRN2*, *SOX9* and *AP-1*) that also slow proliferation. We investigated whether HMGB2 drives invasion through these factors. Using the SKMEL5 melanocytic cell line, we created a stable line expressing V5-tagged HMGB2 and performed ChIP–seq targeting V5 and HMGB2 (Extended Data Fig. 12a). We generated a peak atlas containing 1,361 peaks corresponding to 1,286 unique genes (Extended Data Fig. 12b and Supplementary Table 8). We identified robust HMGB2 binding to the *MITF* promoter (Extended Data Fig. 12c). However, loss of HMGB2 did not meaningfully alter *MITF* expression ($\log_2$FC = 0.157; Extended Data Fig. 12e–g and Supplementary Table 9), indicating that HMGB2 may regulate pro-invasive factors instead. HMGB2 robustly bound the promoter of Notch signalling genes in both A375 (Extended Data Fig. 11d and Supplementary Table 6) and SKMEL5 (Extended Data Fig. 12d and Supplementary Table 8) cells. Notch family genes were also upregulated by human interface cells (*DLL1*, *DLL3* and *DLK2*; Supplementary Table 1) and confined human melanoma cells (*NOTCH2NLA*, *DLK2* and *DLL4*; Supplementary Table 2). Notch signalling promotes melanoma invasion through brain-2 (*BRN2*, also known as *POU3F2*) (ref. 48), a well-characterized pro-invasive transcription factor in melanoma[49] that also functions in neuronal development[50]. This indicates

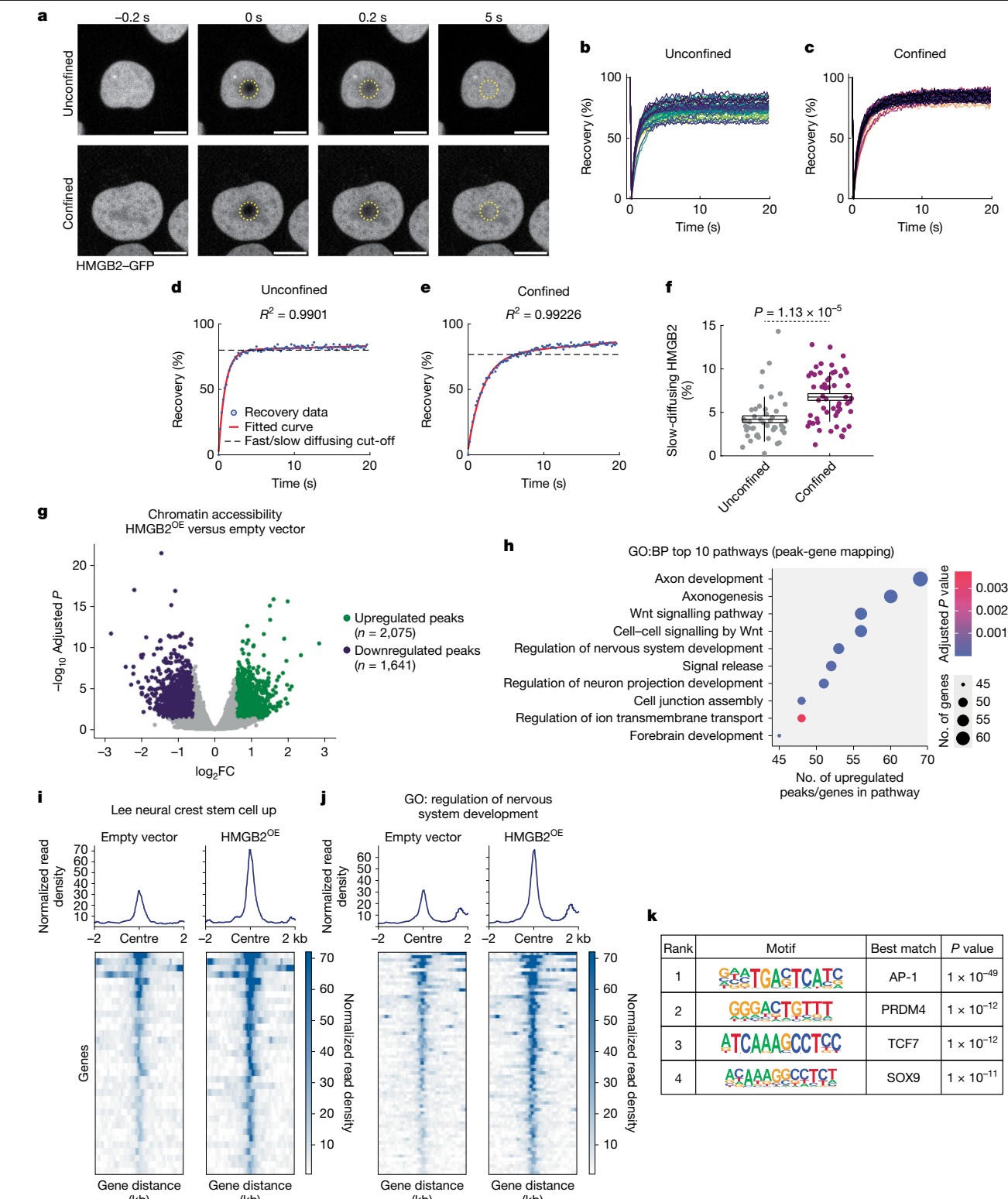

**Fig. 4 | Confinement-mediated stabilization of HMGB2 increases chromatin accessibility at neuronal loci. a**, Representative stills from time-lapse imaging of A375 cells expressing HMGB2–GFP and subjected to FRAP. The yellow dashed region indicates the photobleached area. Time is relative to photobleaching. Scale bars, 10 μm. **b,c**, FRAP recovery curves for HMGB2–GFP in unconfined (**b**) and confined (**c**) cells. Each curve represents fluorescence recovery within the area photobleached on a single cell. **d,e**, Representative plots showing a two-component exponential equation fit to HMGB2–GFP fluorescence recovery curves in unconfined (**d**) and confined (**e**) cells. **f**, Relative proportion of slow-diffusing HMGB2–GFP. Horizontal lines, mean; box, s.e.m.; vertical lines, s.d. $P$ value is indicated (two-sample $t$-test;

two-sided). Unconfined, $n = 45$ cells; confined, $n = 54$ cells (**b**–**f**). **g**, Volcano plot of differentially expressed peaks upon HMGB2[OE]. $P$-value cut-off, 0.05; fold change cut-off, $\log_2(1.25)$ and $\log_2(-1.25)$. **h**, Top 10 enriched GO: Biological Process (BP) pathways from genes mapped to open chromatin loci upon HMGB2[OE]. $P$ values are indicated (two-sided hypergeometric test with Benjamini–Hochberg correction) (**g,h**). **i,j**, Tornado plots showing chromatin accessibility at loci linked to genes from the 'Lee neural crest stem cell up' (**i**) and 'regulation of nervous system development' (**j**) pathways. **k**, Homer de novo motif analysis of transcription factor motifs in open chromatin regions in promoter regions of genes upon HMGB2[OE]. $P$ values are indicated (binomial test).

that invasive and neuronal phenotypes in confined interface cells are induced by HMGB2-mediated upregulation of Notch/BRN2 signalling. Supporting this model, a *BRN2/POU3F2* motif was enriched in the promoter region of HMGB2 target genes (Extended Data Fig. 12h). Although *MITF* and *BRN2* expressions are often inversely correlated[51], previous reports[52,53] and our data indicate that these factors can also be co-expressed. Characterizing the mechanisms that control the expression of these factors and the regulation of the proliferative/invasive trade-off will be an important area for future investigation.

## HMGB2 mediates phenotype switching

One prediction of the melanoma phenotype switching model is the trade-off between opposing phenotypes; highly proliferative cells are less invasive, whereas highly invasive cells are less proliferative (Fig. 5a). Our data indicate that confinement-induced upregulation of HMGB2 could mediate this phenotypic plasticity. Confined A375 cells downregulated many proliferation-related pathways (Fig. 5b). To visualize the influence of confinement on this trade-off, we generated an A375 cell line stably expressing the FastFUCCI cell cycle sensor[54]. Upon confinement, all initially mitotic cells (mAG+ and S/G2–M phase) rapidly lost mAG fluorescence, indicating cell cycle exit (Fig. 5c,d).

To directly test the role of HMGB2 in melanoma invasion, we used in vitro invasion assays (Fig. 5e) and found that targeting *HMGB2* with siRNA significantly impaired invasion ($P = 0.0133$; Fig. 5f,g), whereas overexpression of *HMGB2* increased invasion ($P = 0.0180$; Fig. 5h,i). Previous studies identified a fast amoeboid mode of migration induced by confinement and low adhesion[13,55–57]. We quantified minimal migration of confined A375 cells (mean velocity = $0.0838 \pm 0.0025$ µm min$^{-1}$; Extended Data Fig. 13a–c), which indicates that fast amoeboid migration is not a major contributor to confinement-induced phenotypic plasticity in our system. To test the role of HMGB2 in phenotype switching in vivo, we generated *BRAF*$^{V600E}$ melanomas in zebrafish, in which zebrafish *hmgb2a* and *hmgb2b* were inactivated by CRISPR. Loss of *hmgb2* and *hmgb2b* markedly increased melanoma growth, with tumours growing almost 2-fold larger by 10 weeks ($P = 0.0193$; Fig. 5j–l) while also appearing less invasive than non-targeting controls (Fig. 5m). These data indicate that HMGB2 is required for the invasive state, and that its loss pushes melanoma cells towards a hyperproliferative state.

HMGB family proteins contain three functional domains: A-box and B-box DNA-binding domains and an acidic tail region[58] (Extended Data Fig. 14a). We investigated the functional domains that control HMGB2 localization and activity in melanoma. We assembled constructs with each functional domain removed (Extended Data Fig. 14b) and generated stable A375 cell lines expressing GFP-tagged versions of each (Extended Data Fig. 14c). We were unable to express a truncated construct lacking both DNA-binding domains, probably because of its small size (24 amino acids). In in vitro proliferation and invasion assays, cells expressing either ΔA-box or ΔB-box constructs were significantly less proliferative (Extended Data Fig. 14d) but more invasive (Extended Data Fig. 14e), even compared to full-length HMGB2; this was probably because of higher expression of these smaller constructs (Extended Data Fig. 14f). Because we were unable to generate constructs lacking both DNA-binding domains, when we transplanted cells expressing each deletion construct into mice (Extended Data Fig. 14g), as expected, there was no significant difference in tumour growth rates (Extended Data Fig. 14h,i) or acetylated tubulin at the invasive front (Extended Data Fig. 14j), demonstrating that either HMGB2 DNA-binding domain is sufficient to maintain protein function.

## HMGB2 is associated with drug tolerance

Phenotype switching, particularly the undifferentiated/invasive state, has been linked to drug resistance[2]. This has important clinical relevance because drug-tolerant persister cells are major contributors to relapse[59]. We examined how the confined invasive state affects therapeutic response, hypothesizing that the HMGB2-high neuronal state induced by confinement may promote drug tolerance. To test this, we treated confined cells with Taxol to stabilize MTs. Unconfined cells rapidly displayed fragmented nuclear morphology and underwent apoptosis, whereas confined cells were almost completely resistant to Taxol-induced cell death (Extended Data Fig. 13d). Taxol induces cancer cell death through cell cycle arrest[60], indicating that confinement may cause melanoma cells to exit the cell cycle, in accordance with a pro-invasive phenotype switch associated with downregulation of proliferation[8].

We then tested the role of HMGB2 in drug tolerance in vivo by transplanting human melanoma cells overexpressing *HMGB2* into athymic mice and treating the resulting tumours with dabrafenib and trametinib, widely used targeted therapies in melanoma[61] (Fig. 5n). HMGB2 overexpression significantly impaired melanoma response to dabrafenib/trametinib in vivo ($P = 0.043$; Fig. 5o, Extended Data Fig. 13e and Supplementary Table 10). Together, our results support a role for confinement-mediated upregulation of HMGB2 inducing a pro-invasive and drug-tolerant state in melanoma (Extended Data Fig. 15).

## Discussion

Single-cell profiling has uncovered reproducible transcriptional states that correspond to distinct tumour phenotypes[62]. Melanoma cells exist along a phenotypic axis spanning proliferative and invasive states, with a trade-off in which the invasive and proliferative states are mutually exclusive[1,2]. This is reminiscent of the 'go or grow' hypothesis, in which a cell is optimized for one phenotype over another[63]. The reproducibility of these states in the absence of genetic lesions indicates they may be epigenetically encoded. We found that the transition from proliferation to invasion is in part mediated by the chromatin-associated protein HMGB2. This family of proteins bends DNA to facilitate the action of transcription factors and other chromatin-associated proteins, making them ideal candidates for enforcing changes in chromatin configuration in response to external cues. Our data indicate that HMGB2 responds to mechanical forces exerted on the cell, in which upregulation of HMGB2 increases chromatin accessibility at invasive loci. This may explain the persistence of invasive behaviours once a melanoma cell is exposed to microenvironmental forces. Although our study primarily focused on DNA-centric roles for HMGB2, HMGB family proteins also bind RNA to regulate transcriptional and protein–protein interactions, among other functions[64]. Although we cannot rule out a role for HMGB2–RNA binding in melanoma on the basis of our current data, characterizing this interaction will be an important area for future study.

The role of the mechanical microenvironment on tumour cell phenotypes is still emerging. A previous study showed that increased pressure from a stiff collagen matrix could increase cancer invasion[65]. Cancer-associated fibroblasts can also alter the mechanical microenvironment by secreting matrix metalloproteinases that remodel collagen matrix stiffness[66]. Our study augments these findings by showing that force can also epigenetically remodel tumour cells through reorganization of cytoskeletal, nuclear and chromatin architectures.

Our results indicate that confined melanoma cells enact an invasion program reminiscent of developing neurons. During development, neurons assemble a perinuclear tubulin network to protect the nucleus from high levels of force during confined migration[22,23], similar to what we observed in invasive melanoma cells. It is unclear whether melanoma cells hijack other neuronal behaviours to promote invasion. A previous study from our laboratory indicated that melanoma cells exploit neuronal-like mechanisms during tumour initiation[67], and future studies should aim to explore this phenomenon in later stages of melanoma progression.

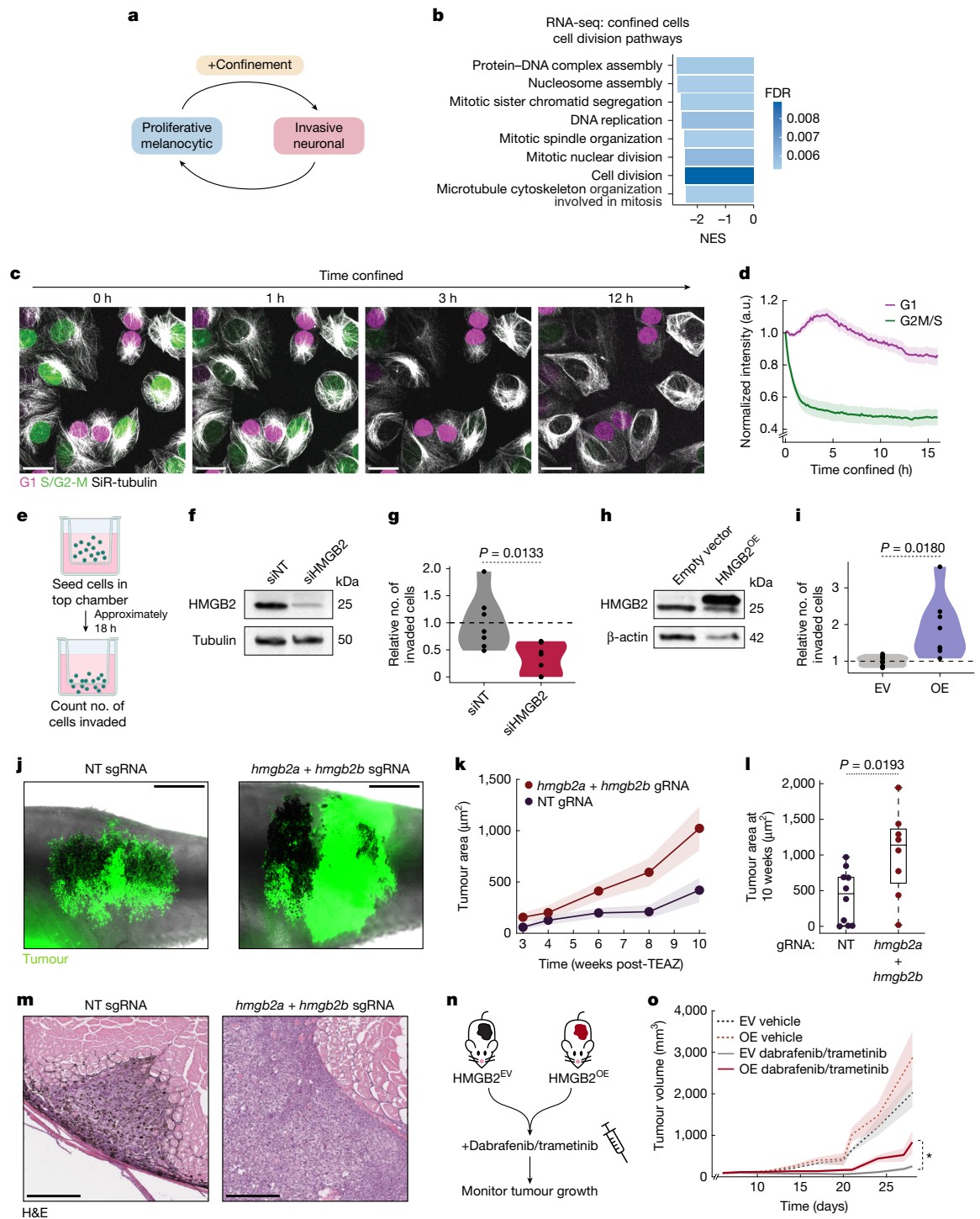

**Fig. 5 | Confinement promotes drug tolerance by downregulating proliferation. a**, Schematic detailing the influence of confinement on melanoma phenotype switching. **b**, Enrichment scores for GO:BP pathways related to cell division from RNA-seq of confined cells relative to unconfined cells. FDR is indicated. **c**, Stills from confocal imaging of A375 cells stably expressing the FastFUCCI reporter and stained with SiR-tubulin. **d**, Quantification of FUCCI signal over time in confined cells; *n* = 37 cells from four videos. Error bars, s.e.m. **e**, Schematic of in vitro invasion assay workflow. **f,h**, Western blot for HMGB2 (top) and tubulin (**f**; loading control; bottom) or β-actin (**h**; loading control; bottom) for A375 cells transfected with the indicated siRNAs (**f**) or plasmids (**h**). siHMGB2, *HMGB2*-targeting siRNA. **g,i**, Quantification of in vitro invasion assay results for A375 transfected with the indicated siRNAs (**g**) or plasmids (**i**); *n* = 3 biological replicates. *P* value is indicated (two-sample *t*-test; two-sided). EV, empty vector; OE, overexpression (of *HMGB2*). **j**, Representative images of adult zebrafish with melanomas generated by means of Transgene

Electroporation in Adult Zebrafish (TEAZ), 12 weeks after electroporation. NT, non-targeting; sgRNA, single-guide RNA. **k**, Tumour surface area over time for melanomas induced in zebrafish using TEAZ. Error bars, s.e.m. **l**, Tumour surface area at 12 weeks after TEAZ. *P* value is indicated (two-sample *t*-test; two-sided). Horizontal line, median; box edges, 25th and 75th percentiles; whiskers, data range. NT sgRNA, *n* = 10 fish; sgRNA targeting *HMGB2* (sgHMGB2), *n* = 8 fish (**j**–**l**). **m**, Representative haematoxylin and eosin (H&E) images of tumour invasion in the indicated conditions at 12 weeks after TEAZ. **n**, Schematic of the drug treatment experiment workflow. **o**, Tumour volume over time for the indicated conditions; *n* = 11 mice per condition from two biological replicates. Error bars, s.e.m. *\*P* < 0.05. *P* value calculated using likelihood ratio tests with a fitted exponential; two-sided (*P* = 0.043). Scale bars, 25 μm (**c**), 2 mm (**j**), 250 μm (**m**). Illustrations in **e** were created using BioRender (https://biorender.com). Mouse cartoons in **n** were adapted from Wikimedia, under a Creative Commons licence CC BY-SA 3.0.

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

## Methods

### Zebrafish husbandry

Stable transgenic zebrafish lines were kept at 28.5 °C in a dedicated aquatics facility with a 14 h on/10 h off light cycle. *Casper* fish with the following genotype were used for all experiments: *mitfa-BRAF^{V600E}*;*p53^{-/-}*; *mitfa^{-/-}*. Fish were anaesthetized using Tricaine (MS-222; stock concentration of 4 g l^{-1}), diluted until the fish were immobilized. All animal procedures were approved by the Memorial Sloan Kettering Cancer Center Institutional Animal Care and Use Committee (protocol no. 12-05-008).

### Cloning of zebrafish CRISPR constructs

To generate *hmgb2a* and *hmgb2b* CRISPR guide RNA (gRNA) plasmids for use in vivo, three gRNAs for each gene were subcloned into Gateway entry vectors containing zebrafish-optimized U6 gRNA promoters. The resulting 3× gRNA plasmid was assembled through Gateway LR cloning. Validation of gRNA/Cas9 activity in vivo was performed using the Alt-R CRISPR-Cas9 system (Integrated DNA Technologies (IDT)) by injecting single-guide RNAs (sgRNAs) and purified Cas9 protein into one-cell-stage zebrafish embryos. Genomic DNA was isolated from five to ten embryos 24 h later, and mutation detection was performed using the Alt-R Genome Editing Detection Kit (IDT).

Zebrafish gRNA sequences:
*hmgb2a* sgRNA1: 5′-GAAAAGTTCACCGAGGTCCC-3′
*hmgb2a* sgRNA2: 5′-AAGGTGAAGGGCGACAACCC-3′
*hmgb2a* sgRNA3: 5′-GACAACCCGGGCATCTCTAT-3′
*hmgb2b* sgRNA1: 5′-CAAACCCAAGGGGAAGACGT-3′
*hmgb2b* sgRNA2: 5′-CTCAAACTTGACCTTGTCGG-3′
*hmgb2b* sgRNA3: 5′-AGAGAAGTTGACGGGCACGT-3′
NT sgRNA: 5′-AACCTACGGGCTACGATACG-3′

### Zebrafish in vivo electroporation

Tumours were generated by means of TEAZ[68]. To generate *hmgb2a/hmgb2b* knockout melanomas, adult 3-month-old to 6-month-old fish were randomly assigned to groups and injected with the following plasmids: miniCoopR–GFP, mitfa:Cas9, Tol2, U6–sgptena, U6–sgptenb and either 394-zU6–3XsgRNA[hmgb2a] and 394-zU6–3XsgRNA[hmgb2b] or 394-zU6–3XsgRNA[NT]. Adult fish were anaesthetized using tricaine and injected with 1 μl of plasmid mixture below the dorsal fin, immediately electroporated and moved to fresh water to recover. Tumour growth was imaged every 1–2 weeks using a ZEISS Axio Zoom V16 fluorescence microscope. Male and female animals were used in equal proportions. No sample size calculation or blinding was performed.

### Cell culture

The following cell lines were obtained from the American Type Culture Collection: A375 (CRL-1619), SKMEL5 (HTB-70), MIA-PaCa-2 (CRM-CRL-1420), Panc-1 (CRL-1469), HTB-4 (T24), HTB-9 (5637) and HEK293T. The cells were maintained in a 37 °C and 5% $CO_2$ humidified incubator. The cell lines were authenticated by the American Type Culture Collection and routinely checked to be free from *Mycoplasma*. The cells were cultured in DMEM (Gibco; 11965) supplemented with 10% fetal bovine serum (GeminiBio; 100-500).

### Transfection of siRNAs

SiRNAs targeting the following genes were obtained from Horizon Discovery: *HMGB2* (L-011689-00-0005), *SYNE2* (L-019259-01-0005) and non-targeting control (D-001810-10-05). DharmaFECT 1 Transfection Reagent (Horizon Discovery; T-2001) was used to transfect 250,000 A375 cells per condition. The medium was changed after 24 h, and experiments were performed 72 h after changing the medium. *HMGB2* knockdown was validated by western blot with an antibody targeting HMGB2 (MilliporeSigma; HPA053314). For gel source data, see Supplementary Fig. 1. Downregulation of *SYNE2* was validated by

means of quantitative polymerase chain reaction (qPCR) with the following primers:
*SYNE2* F: 5′-CAAAGCACAGGAAACTGAGGCAG-3′
*SYNE2* R: 5′-AGACAGTGGCAACGAGGACATG-3′
β-Actin F: 5′-CACCAACTGGGACGACAT-3′
β-Actin R: 5′-ACAGCCTGGATAGCAACG-3′

### Cloning of human CRISPR constructs

The lentiCRISPRv2 system[69] was used to generate stable human knockout cell lines, and gRNAs targeting *HMGB2* or *ATAT1* were selected from the GeCKO2 library. Oligonucleotides containing each gRNA were obtained from IDT and cloned into the lentiCRISPRv2 backbone through restriction digest with *BsmB1* and ligation with Quick Ligase (New England Biolabs). Ligated plasmids were transformed into Stbl3 bacteria (New England Biolabs) and sequenced to verify gRNA insertion. The final plasmids were used to create stable A375 and SK-MEL-5 lines using lentiviral transduction, as described below.

Human gRNA sequences:
*HMGB2* sgRNA1: 5′-CTGCACGAAGAAGGCGTACG-3′
*HMGB2* sgRNA2: 5′-AAGATCAAAAGTGAACACCC-3′
*ATAT1* sgRNA1: 5′-CCAGAAGAACATCTACAGTG-3′
*ATAT1* sgRNA2: 5′-CCTCACTGTAGATGTTCTTC-3′
NT sgRNA: 5′-AACCTACGGGCTACGATACG-3′

### Cloning of HMGB2 overexpression and deletion constructs

To generate the HMGB2–GFP plasmid, the human HMGB2 coding sequence in a pENTR backbone (Horizon Discovery; OHS5898-202621565) was combined with a C terminus EGFP tag using In-Fusion Cloning. The HMGB2–GFP insert was then transferred into a lentiviral expression vector containing the cytomegalovirus promoter (pLX304; Addgene 25890) by means of Gateway cloning using LR Clonase II Plus (Thermo Fisher Scientific). Deletion constructs were generated through In-Fusion Cloning (Takeda Bioscience) using the HMGB2 open reading frame in the pENTR backbone as a template and were subsequently cloned into pLX304 by means of Gateway cloning, as described above. The primers were:
HMGB2-ΔA-box F: 5′-CCAACAAGCCTCCCAAAGGTGATAAGAAGGG-3′
HMGB2-ΔA-box R: 5′-TGGGAGGCTTGTTGGGGTCTCCTTTACC-3′
HMGB2-ΔB-box F: 5′-CCAATGCTGCCAAGGGCAAAAGTGAAGC-3′
HMGB2-ΔB-box R: 5′-CCTTGGCAGCATTGGGGTCCTTTTTCTTCCC-3′
HMGB2-Δacidic tail F: 5′-CATATCGTGACCCAGCTTTCTTGTACAAAG-3′
HMGB2-Δacidic tail R: 5′-CTGGGTCACGATATGCAGCAATATCCT TTTC-3′

### Generation of stable cell lines

HMGB2^{OE}, HMGB2–GFP, HMGB2^{KO}, HMGB2^{del}, ATAT1^{KO} and FastFUCCI stable cell lines were generated by means of lentiviral transduction. The FastFUCCI reporter plasmid was obtained from Addgene (86849). The HMGB2–GFP reporter plasmid, HMGB2 gRNA+Cas9 plasmids, ATAT1 gRNA+Cas9 plasmids and non-targeting gRNA+Cas9 plasmids were assembled, as described above. The HMGB2^{OE} plasmid was obtained from Horizon Discovery (OHS5897-202616132). Eight million HEK293T cells per condition were transfected with 1,200-ng lentiviral vector, 600-ng PAX2 plasmid and 300-ng MD2 plasmid using Effectene Transfection Reagent (QIAGEN). Virus was collected starting 24 h after transfection. Viral supernatant was filtered (0.45-μm filter) before adding to A375 cells at a 1:1 ratio with medium and 10 μg ml^{-1} of polybrene. Cells were infected for 72 h, allowed to recover for 24 h and then selected using blasticidin (5 μg ml^{-1}; 7–10 days) or puromycin (1 μg ml^{-1}; 3 days). For cell lines expressing a fluorescent reporter, cells were sorted using FACSAria III or FACSymphony S6 cell sorters (BD Biosciences). For HMGB2 overexpression and CRISPR lines, successful transduction was validated through western blot with an antibody targeting HMGB2 (MilliporeSigma; HPA053314). For ATAT1^{KO} lines, knockdown was validated through qPCR. For gel source data, see Supplementary Fig. 1.

QPCR primer sequences:
*ATAT1* F: 5′-CACAGTCCCACAGGTGAACA-3′
*ATAT1* R: 5′-CTCCCTGCTTGGAGTCTTGG-3′
β-Actin F: 5′-CACCAACTGGGACGACAT-3′
β-Actin R: 5′-ACAGCCTGGATAGCAACG-3′

## In vitro confinement and imaging

A375, HTB-4 and HTB-9 cells were subjected to overnight (approximately 16 h) confinement at a height of 3 μm using a static cell confiner (4Dcell). Pancreatic ductal adenocarcinoma cell lines (MIA-PaCa-2 and Panc-1) were confined at a 5-μm height owing to their larger size. The cells were plated 6 h before imaging in fibronectin-coated glass-bottom 35-mm dishes (FluoroDish) or glass-bottom six-well plates (MatTek). The cells were allowed to attach before confinement was applied. Confined cells were incubated at 37 °C and 5% $CO_2$ overnight. For live imaging, dyes plus 10 μM verapamil were added to the plated cells 2–3 h before imaging. The dyes used for live imaging were SiR-tubulin (Spirochrome; 100 nM) and SiR-DNA (Spirochrome; 250 nM). Pharmacological inhibitors were added immediately before applying confinement. The inhibitors used were Taxol (Tocris; 1097), tubacin (Selleck Chemicals; S2239), nocodazole (Tocris; 1228) and trichostatin A (MilliporeSigma; T8552). Live imaging was performed on an LSM 880 (ZEISS) confocal microscope at 37 °C and 5% $CO_2$, at ×63 magnification and 5–10 min of temporal resolution, using ZEN Black v.2.3 SP1 software (ZEISS). For immunofluorescence, cells were fixed with 4% paraformaldehyde for 15 min at room temperature before proceeding with staining and imaging, as described below.

## In vitro proliferation and invasion assays

The CyQUANT Direct Red Cell Proliferation Assay (Thermo Fisher Scientific; C35013) was used to assay cell proliferation. Cells were plated at a density of 500 cells per well in 96-well plates and allowed to grow for 72 h. The cell number was quantified using the CyQUANT Direct Red Nuclei Acid Stain and Background Suppressor added at a 1:1 ratio to the cell culture medium, and the intensity was read out at 622 nm on a plate reader. For invasion assays, VitroGel Cell Invasion Assay Kit (TheWell Bioscience; IA-VHM01-1P) and Cultrex Collagen I Cell Invasion Assay kit (Bio-Techne; 3457-096-K) were used. The cells were serum-starved overnight in DMEM, plated in the upper chamber of the invasion assay insert and allowed to migrate for 18 h. The cell number was quantified using crystal violet or calcein staining.

## Immunofluorescence staining and imaging

Cells were plated on glass CC2-coated chamber slides (Thermo Fisher Scientific) or fibronectin-coated glass-bottom dishes (FluoroDish) and allowed to attach for approximately 24 h. The cells were fixed with 4% paraformaldehyde for 15 min, permeabilized with 0.1% Triton in PBS and blocked in 10% goat serum (Thermo Fisher Scientific) for 1 h at room temperature. The primary antibodies used were rabbit anti-HMGB2 (Abcam; ab124670), rabbit anti-HMGB1 (Abcam; ab18256), rabbit anti-HMGA1 (Abcam; ab129153), mouse anti-α-tubulin (MilliporeSigma; CP06), chick anti-β-tubulin (Novus Biologicals; NB100-1612), mouse anti-acetylated tubulin (MilliporeSigma; 6793), rabbit anti-acetylated tubulin (Cell Signaling Technology (CST); 5335), rat anti-tyrosinated tubulin (MilliporeSigma; MAB1864-I), mouse anti-polyglutamylated tubulin (MilliporeSigma; T9822), mouse anti-GFP (Abcam; ab1218), rabbit anti-H3Ac (MilliporeSigma; 06-599), mouse anti-Annexin V (Santa Cruz Biotechnology; sc-74438), rabbit anti-cleaved caspase-3 (CST; 9661), rabbit anti-cleaved PARP (CST; 5625), rabbit anti-YAP (CST; 14074), mouse anti-Twist (Abcam; ab50887), rabbit anti-Snail (CST; 3879), rabbit anti-SMAD3 (Abcam; ab40854) and rabbit anti-SYNE2 (Abcam; ab204308). All primary antibodies were used at 1:200. The cells were incubated with primary antibodies overnight at 4 °C, washed in PBS and incubated with the appropriate fluorescently labelled secondary antibody (1:250). Alexa Fluor 488 conjugated phalloidin

(CST; 8878S), when used, was added at 1:50, and Hoechst was added at 1:1,000. The cells were mounted in VECTASHIELD (Vector Laboratories) and allowed to cure overnight. Stained cells were imaged on a ZEISS LSM 880 confocal at ×40 or ×63 resolution using ZEN Black v.2.3 SP1 software (ZEISS). A Gaussian blur with a radius of 0.5–0.75 pixels was occasionally applied to images to reduce noise for visualization purposes only.

## Staining of human tumour samples

Human melanoma tissue microarrays were obtained from TissueArray. Com (Me481f). Slides were baked at 60 °C for 20 min and deparaffinized in consecutive xylene and ethanol washes. Antigen retrieval was performed using 1X IHC Antigen Retrieval Solution (Thermo Fisher Scientific; 00-4955-58) heated at 95 °C for 20 min in a pressure cooker. After washing in PBS, the samples were blocked in 10% goat serum (Thermo Fisher Scientific) for 1 h at room temperature before incubation overnight at 4 °C in the following primary antibodies, all diluted in blocking buffer at 1:200: rabbit anti-HMGB2 (Abcam; ab124670) and mouse anti-acetylated tubulin (MilliporeSigma; 6793). After washing in PBS, the slides were incubated with the appropriate fluorescently labelled secondary antibody (1:250) and Hoechst (1:1,000). After washing in PBS, a final incubation was performed with a fluorescently conjugated rabbit anti-S100a6 antibody (Abcam; ab204028; 1:250) to label tumour cells before mounting the slides in VECTASHIELD. The slides were imaged on a Pannoramic slide scanner (3DHISTECH) using a ×20/0.8 numerical aperture objective, with higher-resolution images acquired on an LSM 880 confocal (ZEISS), as described above.

## Image analysis

Images were analysed using CellProfiler[70], TrackMate[71] and MATLAB v.R2021b and R2023b (MathWorks). For images of fixed cells, the cells were segmented in CellProfiler using Hoechst staining to generate a nuclei mask and phalloidin or other cytoskeletal staining to generate a whole-cell mask. The mean intensity per cell/nucleus was quantified, and expression of nuclear-localized proteins was normalized to Hoechst intensity per nucleus. For quantification of live imaging data, HMGB2–GFP intensity per cell over time was quantified using TrackMate. The resulting intensity data were analysed in MATLAB by fitting a line to each curve and automatically removing curves in which more than four data points differed from the line of best fit by more than 0.2 a.u. In all cases, plotting and statistics were done in MATLAB. The images were assembled for figure preparation using Fiji (v.2.14).

## Fluorescence recovery after photobleaching

A375 cells expressing HMGB2–GFP were confined for approximately 18 h before FRAP measurements. FRAP was done on an LSM 880 confocal at 37 °C with 5% $CO_2$ using a ×63 oil immersion lens and ZEN Black v.2.3 SP1 software (ZEISS). A 5-μm circular diameter region of interest was defined within the nucleus of each cell before photobleaching at 405 nm and 488 nm wavelengths for ten pulses. One time point was acquired before photobleaching. Fluorescence recovery was imaged at 0.2-s intervals for a total of 20 s. All analyses were performed in MATLAB. For analysis, fluorescence within the region of interest was normalized to the fluorescence at the initial time point (before photobleaching). Samples in which the fluorescence within the region of interest was not bleached to at least 25% of the pre-bleaching value were automatically removed from the analysis. Each recovery curve was fitted with a two-component exponential using the function 'fit' with the 'exp2' parameter: $F(t) = y0 + A_1(1 - e^{-t/\tau 1}) + A_2(1 - e^{-t/\tau 2})$, where $y0$ represents the fluorescence immediately after photobleaching, $A_1$ represents the amplitude of the fast-diffusing population, $A_2$ represents the amplitude of the slow-diffusing population, $t$ is time and $\tau 1$ and $\tau 2$ correspond to the time constants for the fast-diffusing and slow-diffusing populations, respectively.

## Atomic force microscopy

Cells were plated on glass-bottom Petri dishes (FluoroDish FD35) and confined for 18 h, as described above. Immediately after removing the dish from the confiner, cell stiffness was measured using a NanoWizard V microscope (JPK Bruker) in QI Advanced mode. The samples were maintained at 37 °C during imaging using the PetriDishHeater (Bruker). For cell stiffness mapping, 1-μm-diameter spherical AFM probe (silicon nitride cantilever; nominal spring constant $k = 0.2$ N m$^{-1}$; SAA-SPH-1UM; Bruker) was used. Each spring constant of the AFM probe was measured using the thermal noise method in liquid at 37 °C. For the stiffness mapping, a 2 nN set point was used (60 μm × 60 μm image size with 32 × 32 pixels of resolution) to ensure up to 10–20% sample indentation to avoid glass surface influence. The data were processed with JPK Data Processing software using the Hertz model with 0.5 Poisson ratio as a fit parameter. To calculate nuclear stiffness, force maps were segmented on the basis of the corresponding cell height measurements to extract the nuclear region.

## TurboID experiments

**Generation of TurboID constructs.** Cloning of TurboID constructs and validation was performed, as described in a previous study[35]. The cyto-TurboID plasmid was obtained from Addgene. For TurboID–HMGB2, the TurboID cassette was amplified by polymerase chain reaction and cloned into pENTR–HMGB2 at the N terminus of HMGB2 using In-Fusion Cloning. For nuclear localization signal–TurboID, an entry vector was assembled using In-Fusion Cloning, containing the TurboID cassette, followed by three consecutive nuclear localization signal sequences. The pENTR–TurboID–HMGB2 and pENTR–TurboID–3XNLS constructs were subcloned into the pLX304 backbone by Gateway cloning using LR Clonase II Plus (Thermo Fisher Scientific). Stable A375 cell lines were generated, as described above, and expression and localization of the TurboID fusion protein were confirmed by immunofluorescence targeting haemagglutinin (found at the N terminus of the TurboID cassette). The TurboID activity was validated by pulsing the cells with 10 mM biotin (MilliporeSigma; 1071508), followed by both western blotting and immunofluorescence using fluorescently labelled streptavidin. IRDye 800CW Streptavidin (LI-COR; 926-32230) was used for western blotting, and streptavidin conjugated to Alexa Fluor 488 (Thermo Fisher Scientific; S11223) or Alexa Fluor 555 (Thermo Fisher Scientific; S21381) was used for immunofluorescence.

**Sample preparation.** For mass spectrometry experiments, ten million cells per condition and replicate were plated in 15-cm dishes. The medium was removed from the dishes and replaced with 10 mM biotin for 1 h. The labelling reaction was stopped by placing the dishes on ice and washing the cells five times with ice-cold PBS. The cells were then detached by scraping in ice-cold PBS and then pelleted and resuspended in radio-immunoprecipitation assay buffer + protease inhibitors. The cells were lysed by means of sonication (10% amplitude; 2 s per cycle; six cycles), and Bradford assay was used to measure the protein concentration. For each sample, 1-mg protein was incubated with streptavidin magnetic beads (Thermo Fisher Scientific; 88817) in radio-immunoprecipitation assay buffer overnight with rotation at 4 °C. The next day, the beads were pelleted using a magnetic rack, the supernatant was removed and the beads were washed once in 50 mM Tris–HCl (pH 7.5) and twice in 2 M urea in 50 mM Tris–HCl (pH 7.5).

**Protein digestion.** The beads were resuspended in 80 ml of 2 M urea and 50 mM EPPS (pH 8.5) and treated with DL-dithiothreitol (1 mM final concentration) for 30 min at 37 °C with shaking (1,100 rpm) on a Thermomixer (Thermo Fisher Scientific). Free cysteine residues were alkylated with 2-iodoacetamide (3.67 mM final concentration) for 45 min at 25 °C at 1,100 rpm in the dark. The reaction was quenched using 3.67 mM dithiothreitol, and LysC (750 ng) was added, followed

by incubation for 1 h at 37 °C at 1,150 rpm. Finally, trypsin (750 ng) was added, followed by incubation for 16 h at 37 °C at 1,150 rpm. After incubation, the digest was acidified to pH less than 3 with the addition of 50% of trifluoroacetic acid (TFA), and the peptides were desalted on Sep-Pak C18 cartridges (Waters). Briefly, the cartridges were conditioned by sequential addition of (1) 100% methanol; (2) 70% acetonitrile (ACN)/0.1% TFA; and (3) 5% ACN/0.1% TFA twice. After conditioning, the acidified peptide digest was loaded onto the cartridge. The stationary phase was washed with 5% ACN/0.1% formic acid twice. Finally, peptides were eluted using 70% ACN/0.1% formic acid twice. Eluted peptides were dried under vacuum in a SpeedVac centrifuge followed by reconstitution in 12 μl of 0.1% formic acid, sonication and transfer to an autosampler vial. Peptide yield was quantified using NanoDrop (Thermo Fisher Scientific).

**Mass spectrometry.** Peptides were separated on a 25-cm column with a 75-mm diameter and 1.7-mm particle size composed of C18 stationary phase (IonOpticks; Aurora 3 1801220) using a gradient from 2% to 35% B over 90 min and then to 95% B for 7 min (buffer A, 0.1% formic acid in high-performance liquid chromatography-grade water; buffer B, 99.9% ACN and 0.1% formic acid) with a flow rate of 300 nl min$^{-1}$ using a nanoElute 2 system (Bruker). Mass spectrometry data were acquired on a timsTOF HT (Bruker) with a CaptiveSpray source (Bruker) using a data-independent acquisition parallel accumulation–serial fragmentation (PASEF) method (dia-PASEF). The mass range was set from 100 to 1700 $m/z$, and the ion mobility range was set from 0.60 V s cm$^{-2}$ (collision energy of 20 eV) to 1.6 V s cm$^{-2}$ (collision energy of 59 eV) with a ramp time of 100 ms and an accumulation time of 100 ms. The dia-PASEF settings included a mass range of 400.0–1,201.0 Da, mobility range of 0.60–1.60 and a cycle time estimate of 1.80 s. The dia-PASEF windows were set with a mass width of 26.00 Da, mass overlap of 1.00 Da and 32 mass steps per cycle.

**Data analysis.** Raw data files were processed using Spectronaut v.18.5 (Biognosys) and searched with the Pulsar search engine with a human UniProt protein database downloaded on 15 August 2023 (226,261 entries). Cysteine carbamidomethylation was specified as a fixed modification, whereas methionine oxidation, acetylation of the protein N terminus and deamidation (NQ) were set as variable modifications. A maximum of two trypsin missed cleavages were permitted. Searches used a reversed sequence decoy strategy to control peptide FDR, and 1% FDR was set as the threshold for identification. Unpaired $t$-test was used to calculate $P$ value in differential analysis, and volcano plot was generated on the basis of log$_2$FC and $q$ value (multiple testing corrected $P$ value using Benjamini–Hochberg method). A $q$ value ≤ 0.05 was considered the statistically significant cut-off.

## Mouse experiments

Mouse in vivo studies were performed in accordance with the guidelines approved by the Memorial Sloan Kettering Cancer Center Institutional Animal Care and Use Committee and Research Animal Resource Center. The mice were housed under pathogen-free conditions, in an environment with controlled temperature (21.5 °C ± 1.5 °C) and humidity (55% ± 10%) and under 12 h light/dark cycles. For the drug efficacy studies, 6-week-old to 8-week-old athymic female mice (The Jackson Laboratory) were injected subcutaneously with five million A375 cells in a 50:50 mix with Matrigel (Corning). Once tumours reached an average volume of 100 mm$^3$, the mice were randomized into two treatment groups ($n = 4$–6 mice per group) to receive either a vehicle control or trametinib (1 mg kg$^{-1}$) in combination with dabrafenib (30 mg kg$^{-1}$). Both drugs were delivered through oral gavage daily five times for 3 weeks. The mice were observed daily throughout the treatment period for signs of morbidity/mortality. Tumours were measured twice weekly using calipers, and volume was calculated using the following formula: length × width$^2$ × 0.52. Body weight was also assessed twice weekly.

For the HMGB2 deletion construct growth curve studies, tumour cells were implanted, as described above, and tumour volume was measured twice weekly. The animals were monitored until their tumour size reached 1,500 mm³, at which point tumours were collected, fixed in 10% formalin for 24 h, transferred to 70% ethanol and processed for histology. In accordance with limits established by the Memorial Sloan Kettering Cancer Center (MSKCC) Institutional Animal Care and Use Committee, the mice were euthanized when tumour burden exceeded 1,500 mm³. These limits were not exceeded in any of the experiments. Histology was performed by HistoWiz using the following antibodies for immunohistochemistry: mouse anti-BRAF[V600E] (Abcam; ab228461) and rabbit anti-acetylated tubulin (Abcam; ab179484). Two biological replicates consisting of four to six mice per condition (for a total of 10–11 mice per group) were performed for both experiments. No sample size calculation or blinding was performed.

### Bulk RNA-seq and analysis

For bulk RNA-seq of A375 and SKMEL5 cells overexpressing HMGB2, three replicates of approximately one million cells each were pelleted and resuspended in TRIzol before snap freezing. For bulk RNA-seq of confined A375 cells, 200,000 cells were plated in each well of a six-well plate. Three wells were confined for approximately 18 h using a six-well static confiner (4Dcell) at 3-µm height, whereas the remaining three wells were left unconfined. The cells were then collected in TRIzol, pooling the three wells for each condition to generate samples of approximately 600,000 cells each. This process was repeated for a total of three independent biological replicates per condition. Library preparation and sequencing were done by Azenta Life Sciences. Raw sequencing reads were processed using FastQC (Babraham Bioinformatics) and Trimmomatic[72] before alignment to the human genome hg38. All downstream analyses were performed in R (v.4.3.1). Differential gene expression was analysed using DESeq2 (ref. 73) with the default parameters. GSEA was performed using the fgsea[74] R package (v.1.26) with Gene Ontology biological process pathway sets from MSigDB[75].

### Bulk ATAC-seq and analysis

Samples containing approximately 100,000 cells each were centrifuged at 700g for 5 min at 4 °C before being resuspended in 500-µl growth medium supplemented with 10% DMSO. The cells were frozen at −80 °C overnight before library preparation and sequencing were performed by Azenta Life Sciences. Sequencing reads were trimmed and filtered for quality control using TrimGalore (v.0.6.7) with a quality setting of 15, Cutadapt[76] (v.4.0) and FastQC v.0.12.1. Reads were aligned to the human genome assembly hg38 using Bowtie 2 (ref. 77) (v.2.3.5.1) and were deduplicated using MarkDuplicates from Picard (Broad Institute; v.2.16). Peaks were identified using MACS2 (ref. 78) with a P-value setting of 0.001 using a publicly available melanocyte dataset (GSM3191792) as control. To generate a global peak atlas, blacklisted regions were removed before merging all peaks within a 500-bp region and quantifying reads using featureCounts. Differentially enriched peaks were identified using DESeq2 (ref. 73). Peak gene mapping was done by assigning all intergenic peaks to that gene and, in other cases, by genomic distance to the transcription start site. Pathway was analysed using clusterProfiler[79]. Tornado plots were generated with deepTools[80] (v.3.5.1) functions (computeMatrix and plotHeatmap), with genes annotated from the indicated pathway sets. Motif enrichment was analysed using Homer[81] (v.4.11.1) functions (findMotifsGenome and annotatePeaks).

### ChIP sequencing

**Sample preparation and sequencing.** For profiling of HMGB2 binding in A375 and SKMEL5 cells, freshly collected cells (approximately 20 million cells/replicate/condition) were crosslinked first with 1.5 mM of EGS (Thermo Fisher Scientific; 21565) for 20 min at room temperature and subsequently with 1% formaldehyde (Thermo Fisher Scientific; 28906) for 40 min at 4 °C. The reaction was quenched by the addition of glycine to the final concentration of 0.125 M. Fixed cells were washed twice with PBS and resuspended in SDS buffer (100 mM NaCl, 50 mM Tris–HCl (pH 8.0), 5 mM EDTA, 0.5% SDS and 1× protease inhibitor cocktail; Roche). The resulting nuclei were spun down, resuspended in the immunoprecipitation buffer (100 mM NaCl, 100 mM Tris–HCl (pH 8.0), 5 mM EDTA and 5% Triton X-100) at 1 ml per 0.5 million cells mixed in 2:1 ratio, with the addition of 1× protease inhibitor cocktail (MilliporeSigma; 11836170001). The nuclei were processed on a Covaris E220 Focused-ultrasonicator to achieve an average fragment length of 200–300 bp with the following parameters: peak incident power = 140, duty factor = 5, cycles per burst/burst per second = 200 and time = 20 min (for A375 cells) or 45 min (for SKMEL5 cells). Chromatin concentrations were estimated using the Pierce BCA Protein Assay Kit (Thermo Fisher Scientific; 23227) according to the manufacturer's instructions. The immunoprecipitation reactions were set up in 500 µl of the immunoprecipitation buffer in Protein LoBind Tubes (Eppendorf; 22431081) and pre-cleared with 50 µl of Dynabeads Protein G (Thermo Fisher Scientific; 10004D) for 2 h at 4 °C. After pre-clearing, the samples were transferred into new Protein LoBind Tubes and incubated overnight at 4 °C with 5 µg of HMGB2 (Abcam; ab67282), V5 (Abcam; ab9116; used for SKMEL5 ChIP only) and H3K4me3 (Epicypher; 13-0041) antibodies. For normalization purposes, 5 µl of *Drosophila* spike-in chromatin (Active Motif; 53083) and 2 µl of spike-in antibody (Active Motif; 61686) were added to each reaction. The next day, 50 µl of BSA-blocked Dynabeads Protein G was added to each reaction and incubated for 2 h at 4 °C. The beads were then washed twice with low-salt washing buffer (150 mM NaCl, 1% Triton X-100, 0.1% SDS, 2 mM EDTA and 20 mM Tris–HCl (pH 8.0)), twice with high-salt washing buffer (500 mM NaCl, 1% Triton X-100, 0.1% SDS, 2 mM EDTA and 20 mM Tris–HCl (pH 8.0)), twice with LiCL wash buffer (250 mM LiCl, 10 mM Tris–HCl (pH 8.0), 1 mM EDTA, 1% Na deoxycholate and 1% IGEPAL CA-630) and once with TE buffer (10 mM Tris–HCl (pH 8.0) and 1 mM EDTA). The samples were then reverse-crosslinked overnight in the elution buffer (1% SDS and 0.1 M NaHCO₃) and purified using the ChIP DNA Clean & Concentrator kit (Zymo Research; D5205) following the manufacturer's instructions. After quantification of the recovered DNA fragments, libraries were prepared using the ThruPLEX DNA-Seq Kit (Takara Bio; R400676) following the manufacturer's instructions, purified with SPRIselect magnetic beads (Beckman Coulter; B23318) and quantified using a Qubit Flex fluorometer (Thermo Fisher Scientific) and profiled using TapeStation (Agilent). The libraries were sent to MSKCC Integrated Genomics Operation core facility for sequencing on an Illumina NovaSeq 6000 (approximately 30–40 million 100-bp paired-end reads per library).

**Data analysis.** ChIP–seq reads were trimmed and filtered for quality and library adaptors using TrimGalore (v.0.4.5) with a quality setting of 15 and running cutadapt[76] (v.1.15) and FastQC (v.0.11.5). Reads were aligned to human assembly hg38 using Bowtie 2 (v.2.3.4.1) (ref. 77) and were deduplicated using MarkDuplicates in Picard Tools (v.2.16.0). To ascertain enriched regions, MACS2 (ref. 78) was used with a P-value setting of 0.001 and run against a matched control for each condition. A peak atlas was created by combining the superset of all peaks using the 'merge' function in the BEDTools suite (v.2.29.2). Read density profiles were created using deepTools 'bamCoverage' (v.3.3.0), normalized to ten million uniquely mapped reads and with read pile-ups extended to 200 bp. The tool featureCounts (v.1.6.1) was used to build a raw count matrix, and DESeq2 was used to calculate the differential enrichment for all pairwise contrasts for experiments with replicates. For single-sample data, MACS2 was run by swapping bams of different conditions to find differential regions. Peak gene associations were created by assigning all intragenic peaks to that gene, whereas intergenic peaks were assigned using the linear genomic distance to the

transcription start site. GSEA[82] was performed with the pre-ranked option and default parameters, in which each gene was assigned the single peak with the largest (in magnitude) $\log_2FC$ associated with it. Composite and tornado plots were created using deepTools[80] (v.3.3.0) by running computeMatrix and plotHeatmap on normalized bigwigs with average signal sampled in 25-bp windows and flanking region defined by the surrounding 2 kb. Motif signatures were obtained using Homer[81] (v.4.5).

### Reanalysis of human melanoma scRNA-seq data

Human melanoma scRNA-seq data from a previous study[7] were downloaded from Gene Expression Omnibus (GEO) (GSE115978). All analyses were performed in R using Seurat[83] (v.4.4.0 and v.5.0.1). The count matrix was normalized using sctransform. Clustering was done using Seurat functions (FindNeighbors and FindClusters) with a resolution of 0.8. Cell types and treatment status were annotated using metadata from the original publication[7]. Cell types were classified using gene lists from a previous study[2] and the Seurat function AddModuleScore with default parameters. Module scores were scaled between 0 and 1. Cells were classified by differentiation state on the basis of the highest expression score for the given gene modules. Differentially expressed genes were calculated using the Seurat function FindMarkers with default parameters. GSEA was performed using fgsea, as described above.

### Statistics and reproducibility

All statistical analyses and plotting were performed in either R (for RNA-seq and ATAC-seq data; v.4.3.1) or MATLAB (for imaging data; v.R2021b). For scRNA-seq data, $P$ values were calculated using the Wilcoxon rank-sum test with Bonferroni's correction for multiple groups (R function pairwise.wilcox.test). Pearson correlation coefficients and corresponding $P$ values were calculated using the R function cor.test. For differential expression analyses of bulk RNA-seq and bulk ATAC-seq data, $P$ values were calculated in DESeq2 using the Wald test. For image analysis, $P$ values were calculated using MATLAB functions (anova1 and multcompare) using the Tukey post hoc test. To calculate the cell migration velocity, the Euclidean distance travelled by individual cells across time points was measured using the MATLAB function pdist, and the velocity was calculated by dividing by the time step. For mouse experiments, we performed a series of likelihood ratio tests to investigate growth rate differences. A biexponential model was fit to the growth curve using maximum likelihood estimation to obtain estimates for the early-time and late-time growth rates, with a single exponential fit to vehicle data. For all representative images shown, the images represent at least three independent replicates.

### Reporting summary

Further information on research design is available in the Nature Portfolio Reporting Summary linked to this article.

### Data availability

Raw and processed RNA-seq, ATAC-seq and ChIP–seq data generated in this study were deposited in GEO (accession no. GSE253803). Human melanoma scRNA-seq data were obtained from GEO (accession no. GSE115978). The TurboID proteomics data were deposited in the ProteomeXchange Consortium through the Proteomics Identifications Database partner repository with the dataset identifier PXD060265. All other relevant data supporting the key findings of this study are available within the Article and its Supplementary Information or from the corresponding authors upon request. Source data are provided with this paper.

### Code availability

All codes used for analysis and plotting are available at GitHub (github. org/mvhunter1/Hunter_2024).

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

**Acknowledgements** We thank M. Sherman and the members of White and Sherman laboratories for their useful discussions. We thank the MSKCC Molecular Cytology Core Facility for assistance with imaging, the MSKCC Aquatics Core Facility for zebrafish care and maintenance and the MSKCC Flow Cytometry Core Facility for assistance with FACS. M.V.H. was funded by the K99/R00 Pathway to Independence Award from the National Cancer Institute (1K99CA266931), Scholarship for the Next Generation of Scientists from the Cancer Research Society and a postdoctoral fellowship from the Canadian Institutes of Health Research. Y.M. was supported by a Medical Scientist Training Program grant from the NIH under award number T32GM007739 to the Weill Cornell/Rockefeller/Sloan Kettering Tri-Institutional MD-PhD Program and Kirschstein National Research Service Award predoctoral fellowship under award number F30CA265124. I.Y. and R.M.W. were funded by NIH Research Program Grant 1U01CA260432. R.M.W. was funded by the Melanoma Research Alliance, Debra and Leon Black Family Foundation, NIH Research Program Grants R01CA229215 and R01CA238317, NIH Director's New Innovator Award DP2CA186572, Pershing Square Sohn Foundation, The Mark Foundation for Cancer Research, The Alan and Sandra Gerry Metastasis Research Initiative at MSKCC, The Harry J. Lloyd Foundation, Consano, Starr Cancer Consortium and American Cancer Society RSG-19-024-01-DDC. This study was supported by the NCI Cancer Center Support Grant PO CA008748 to Memorial Sloan Kettering Cancer Center.

**Author contributions** M.V.H. and R.M.W. conceived the study. M.V.H. performed all experiments and analyses, unless otherwise noted. S.B. and E.dS. performed mouse experiments. E.J., Z.Y. and P.-J.H. performed ChIP–seq experiments. L.T., Z.L. and M.M. performed TurboID experiments and analysis. Y.H.K. performed AFM. A.B. and H.B. performed statistical analyses. E.R. imaged the human tumour samples. E.M. and Y.M. performed TEAZ experiments. R.M. and I.Y. assisted with the transcriptomic experiments. R.P.K. performed ChIP–seq analysis and assisted with ATAC-seq analysis. M.V.H. and R.M.W. wrote the paper. All authors provided feedback before submission.

**Competing interests** R.M.W. is a paid consultant to N-of-One Therapeutics, a subsidiary of QIAGEN. R.M.W. is on the scientific advisory board of Consano but receives no income for this. R.M.W. receives royalty payments for the use of the Casper zebrafish line from Carolina Biological Supply. The other authors declare no competing interests.

**Additional information**
**Correspondence and requests for materials** should be addressed to Miranda V. Hunter or Richard M. White.

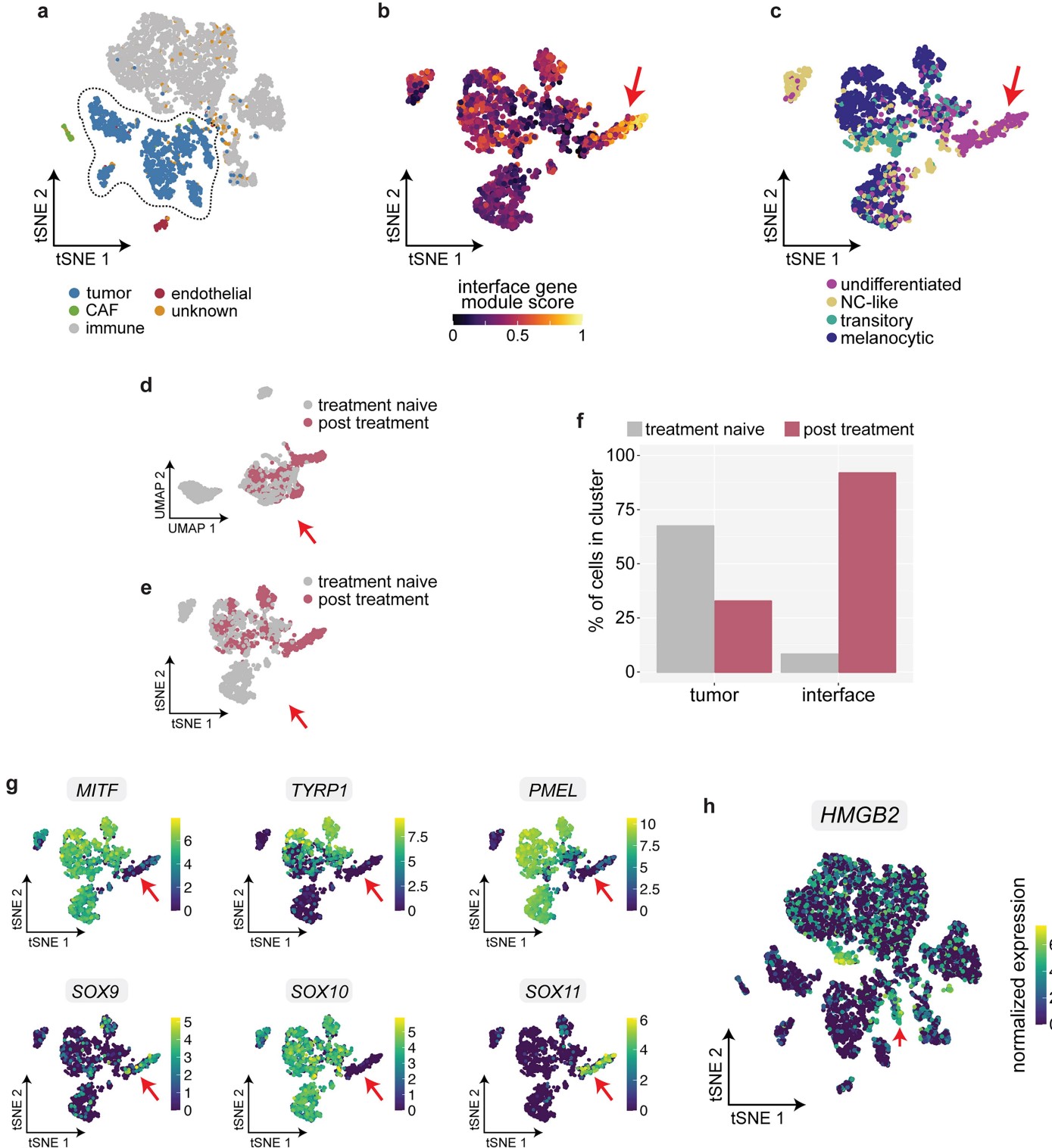

**Extended Data Fig. 1 | Interface cells are found in human patient samples.**
**a**. tSNE of human melanoma scRNA-seq dataset from Jerby-Arnon et al. Cluster annotations from the original manuscript are labeled. Tumor cell clusters are outlined. **b**. Gene module scoring for interface genes extracted from zebrafish spatial transcriptomics and scRNA-seq data, projected onto tumor cells outlined in a. Red arrow denotes the subpopulation with highest expression of interface genes. **c**. Cell state classification for melanoma differentiation states identified by Tsoi et al. Cells were classified based on highest expression of the gene modules indicated. **d-f**. Tumor and interface cells classified by treatment status. **g-h**. Normalized expression per cell for the indicated genes. Red arrow indicates the interface cluster identified in b.

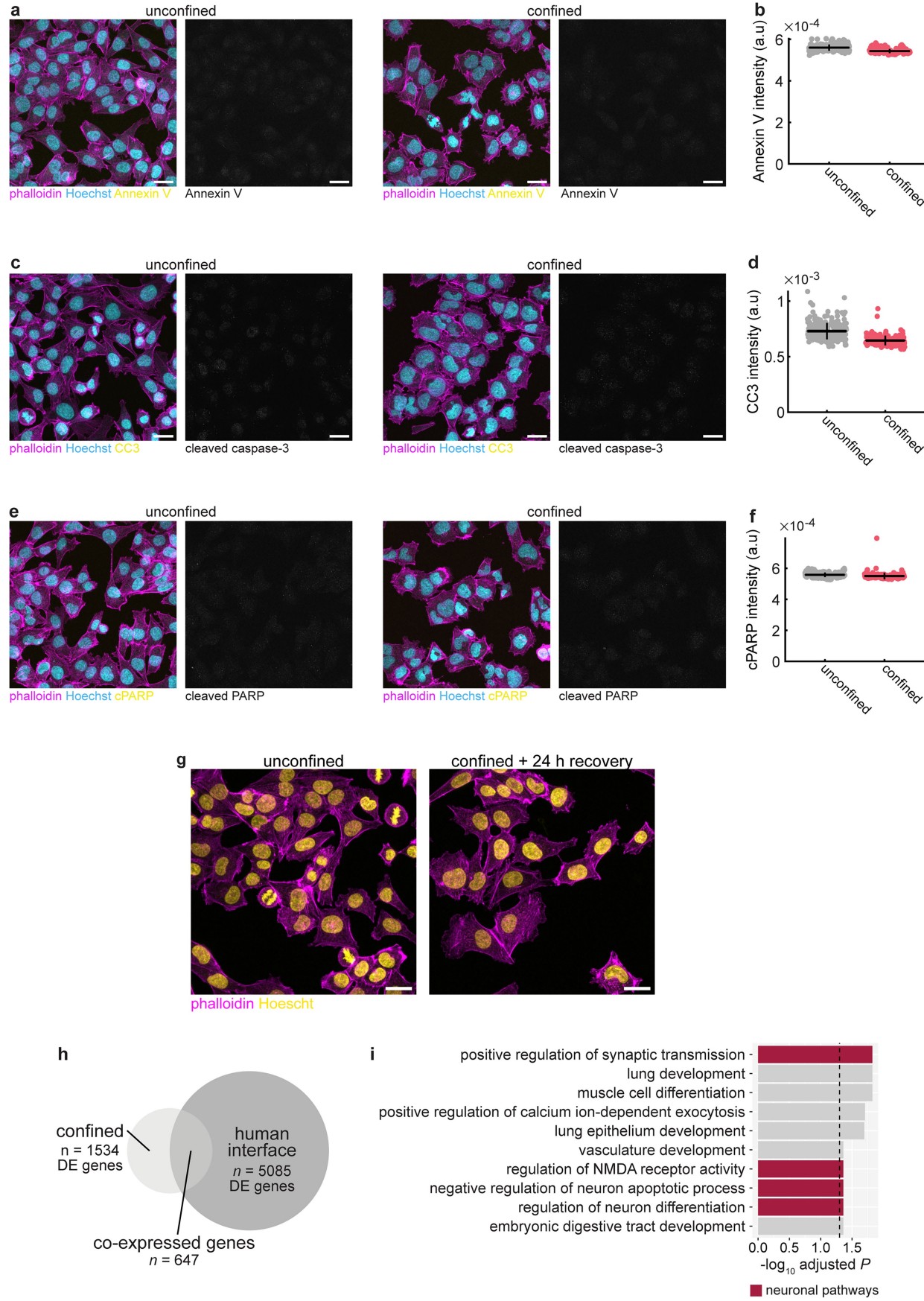

**Extended Data Fig. 2** | See next page for caption.

**Extended Data Fig. 2 | Confinement does not cause apoptosis and induces a neuronal gene program. a,c,e**. IF of confined A375 cells labelled with antibodies against the apoptosis markers Annexin V (a), cleaved caspase-3 (c), and cleaved PARP (e). **b,d,f**. Intensity per cell for the indicated markers. **b**. Unconfined: $n = 250$ cells from 6 images. Confined: $n = 225$ cells from 6 images. **d**. Unconfined: $n = 261$ cells from 6 images. Confined: $n = 185$ cells from 6 images. **f**. Unconfined: $n = 245$ cells from 6 images. Confined: $n = 174$ cells from 6 images. **g**. IF of A375 cells confined for ~18 h and then left to recover for 24 h (right) or unconfined cells (right). **a,c,e,g**. Scale bars, 25 μm. **h**. Euler diagram depicting co-expression of differentially expressed (DE) genes with an adjusted P-value < 0.05. **i**. Results from pathway analysis on the co-upregulated genes between both datasets using the GO:BP pathway set. Pathway analysis performed with Enrichr. Neuronal pathways are highlighted.

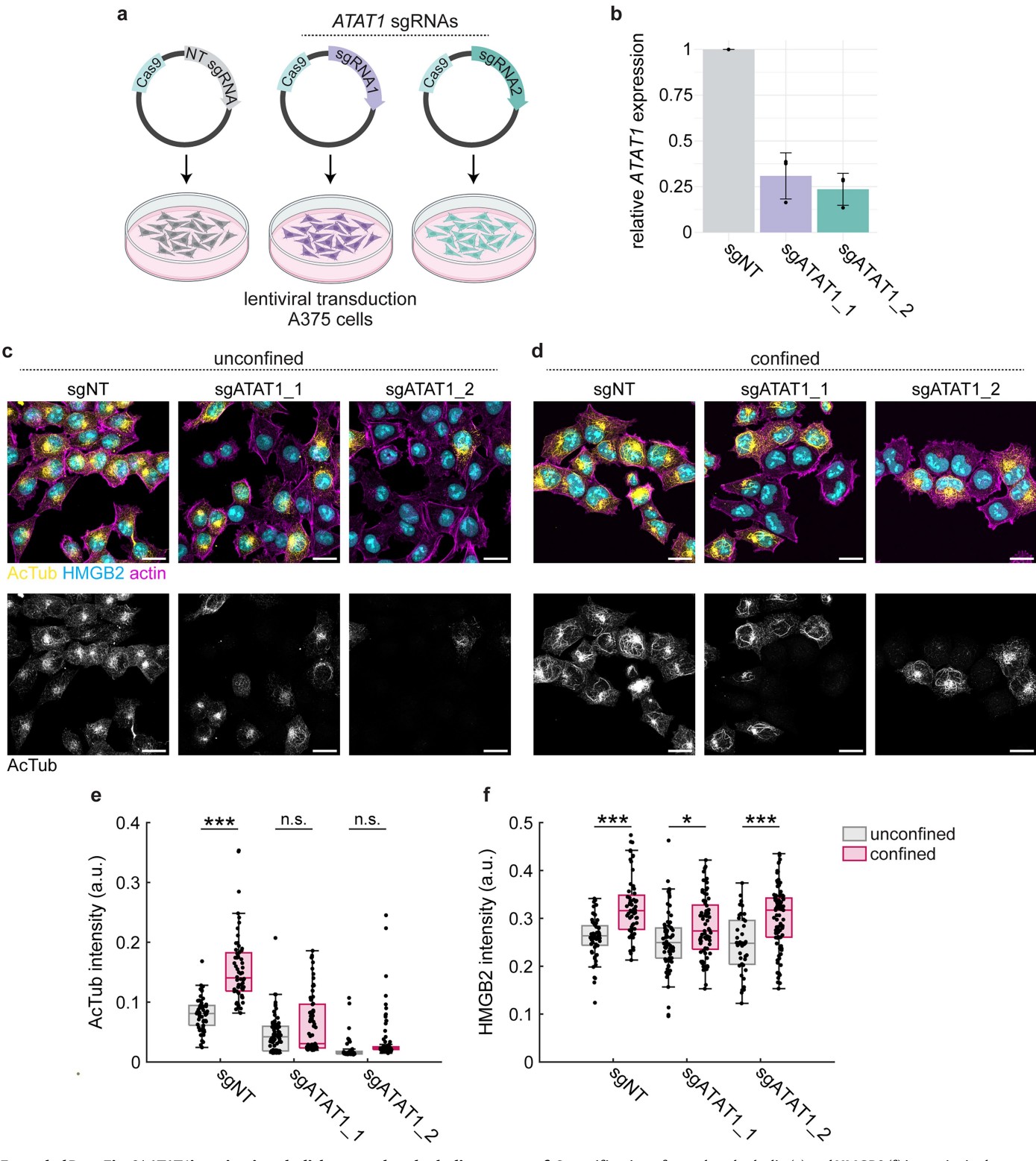

**Extended Data Fig. 3 | *ATAT1* inactivation abolishes acetylated tubulin.**
**a**. Schematic detailing generation of ATAT1 knockdown cell lines. **b**. Expression of *ATAT1* mRNA in the noted cell lines. Expression is normalized to β-actin and the non-targeting control condition. For each cell line, *n* = 3 biological replicates for a total of *n* = 12 technical replicates. **c-d**. IF images showing expression of acetylated tubulin (yellow, top; bottom), HMGB2 (cyan, top) and actin (magenta, top) in unconfined (c) and confined (d) cells. Scale bars, 20 μm.

**e-f**. Quantification of acetylated tubulin (e) and HMGB2 (f) intensity in the noted conditions. sgNT unconfined: *n* = 55 cells. sgNT confined: *n* = 60 cells. sgATAT1_1 unconfined: *n* = 75 cells. sgATAT1_1 confined: *n* = 73 cells. sgATAT1_2 unconfined: *n* = 41 cells. sgATAT1_2 confined: *n* = 94 cells. *n* = 8 images were used for quantification for each condition. ***, $P < 0.001$; *, $P < 0.05$; n.s., $P > 0.05$. Illustrations in **a** were created using BioRender (https://biorender.com).

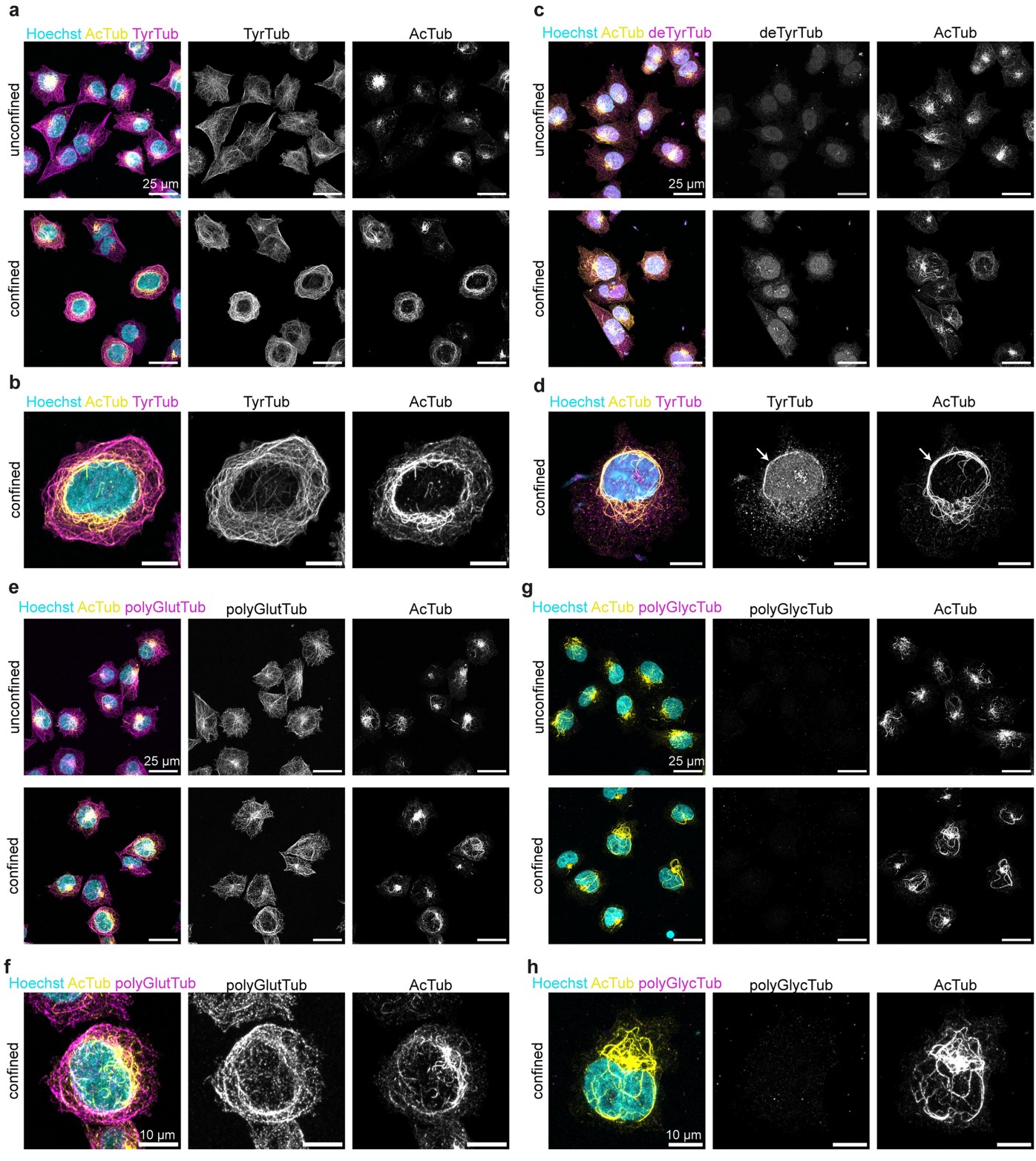

**Extended Data Fig. 4 | Localization of tubulin post-translational modifications in confined cells. a-h**. IF of A375 cells labeled with antibodies targeting acetylated tubulin, in addition to tyrosinated tubulin (a-b), detyrosinated tubulin (c-d), polyglutamylated tubulin (e-f), and polyglycylated tubulin (g-h). **a,c,e,g**. Scale bars, 25 μm. **b,d,f,h**. Scale bars, 10 μm.

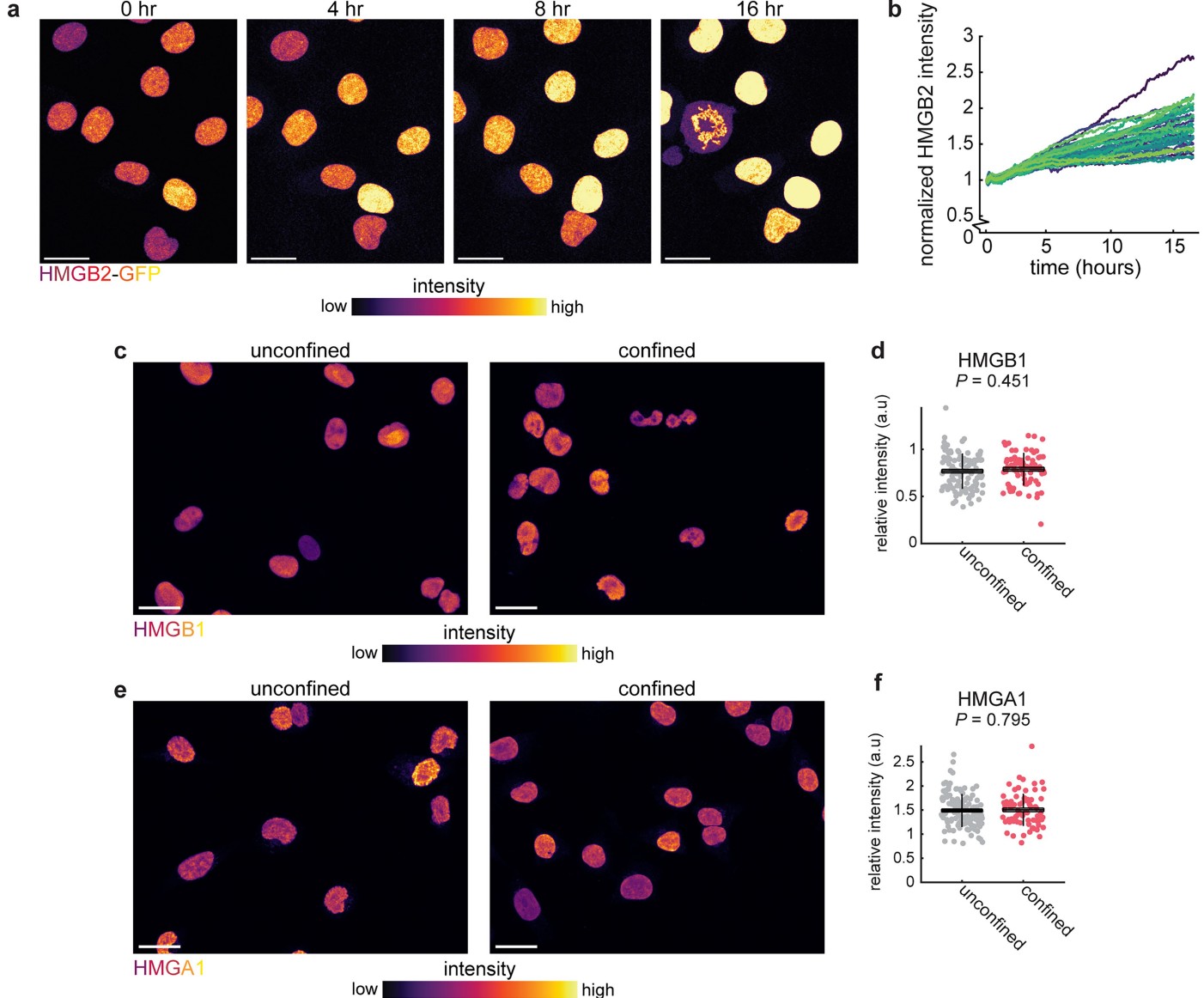

**Extended Data Fig. 5 | Confinement specifically upregulates HMGB2.**
**a**. Stills from confocal imaging of A375 cells expressing HMGB2-GFP. Cells are pseudocolored by HMGB2 intensity. **b**. HMGB2-GFP intensity per cell over time, normalized to intensity at the first time point acquired. $n = 37$ cells from 6 movies. **c,e**. Immunofluorescence images of A375 cells stained with antibodies targeting HMGB1 (c) and HMGA1 (e). Scale bars, 25 μm.

**d,f**. Quantification of intensity in confined/unconfined cells for the indicated markers. Each point represents 1 cell. Horizontal lines, mean; box, SEM; vertical lines, SD. **d**. Unconfined: $n = 105$ cells from 9 images. Confined: $n = 76$ cells from 9 images. **f**. Unconfined: $n = 125$ cells from 9 images. Confined: $n = 80$ cells from 9 images.

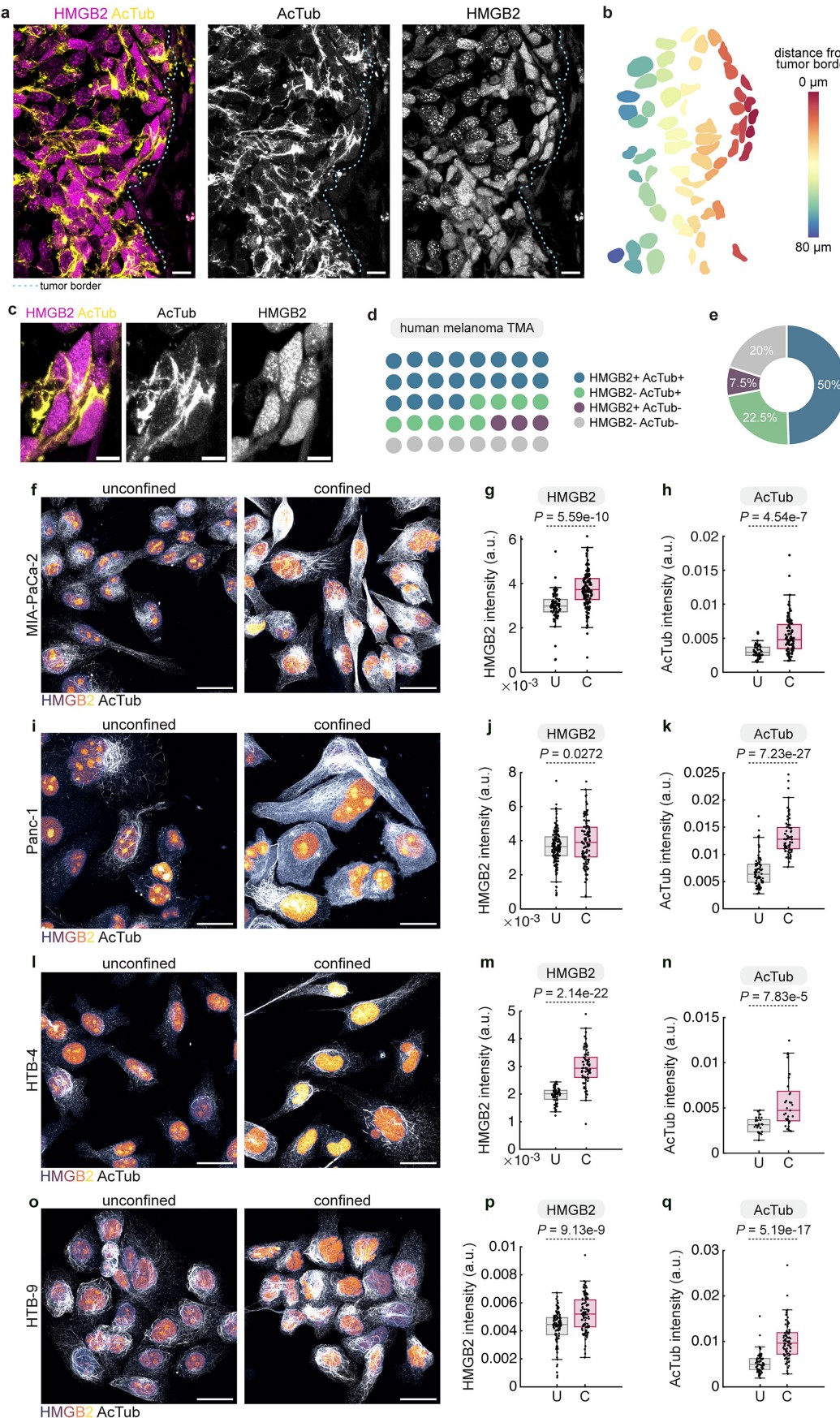

**Extended Data Fig. 6** | See next page for caption.

**Extended Data Fig. 6 | Interface cells are present in human samples and other cancers. a**. Immunofluorescence performed on a human melanoma tissue sample showing enrichment of HMGB2 and acetylated in elongated nuclei at the tumor border. **b**. Visualization of nuclear shape relative to distance from the tumor border. **c**. Inset of a. **d**. Schematic of a human melanoma tissue microarray colored by the presence of HMGB2+ and/or acetylated tubulin+ cells in each sample. **e**. Quantification of d. **f,i,l,o**. Immunofluorescence images of confined MIA-PaCa-1 (f), Panc-1 (i), HTB-4 (l), and HTB-9 (o) cells showing HMGB2 (red/orange) and acetylated tubulin (white). Scale bars, 25 μm. **g-h,j-k,m-n,p-q**. Quantification of HMGB2 (g,j,m,p) and acetylated tubulin (h,k,n,q) intensity. U = unconfined, C = confined. *P*-values are indicated. For quantification of nuclear HMGB2 intensity: MIA-PaCa-2 unconfined: $n$ = 74 cells from 8 images. MIA-PaCa-2 confined: $n$ = 128 cells from 8 images. Panc-1 unconfined: $n$ = 159 cells from 14 images. Panc-1 confined: $n$ = 119 cells from 9 images. HTB-4 unconfined: $n$ = 63 cells from 8 images. HTB-4 confined: $n$ = 78 cells from 9 images. HTB-9 unconfined: n = 121 cells from 8 images. HTB-9 confined: $n$ = 103 cells from 8 images. For quantification of whole-cell acetylated tubulin intensity: MIA-PaCa-2 unconfined: $n$ = 41 cells from 8 images. MIA-PaCa-2 confined: $n$ = 84 cells from 8 images. Panc-1 unconfined: $n$ = 82 cells from 14 images. Panc-1 confined: $n$ = 77 cells from 9 images. HTB-4 unconfined: $n$ = 26 cells from 8 images. HTB-4 confined: $n$ = 33 cells from 9 images. HTB-9 unconfined: $n$ = 75 cells from 8 images. HTB-9 confined: $n$ = 88 cells from 8 images.

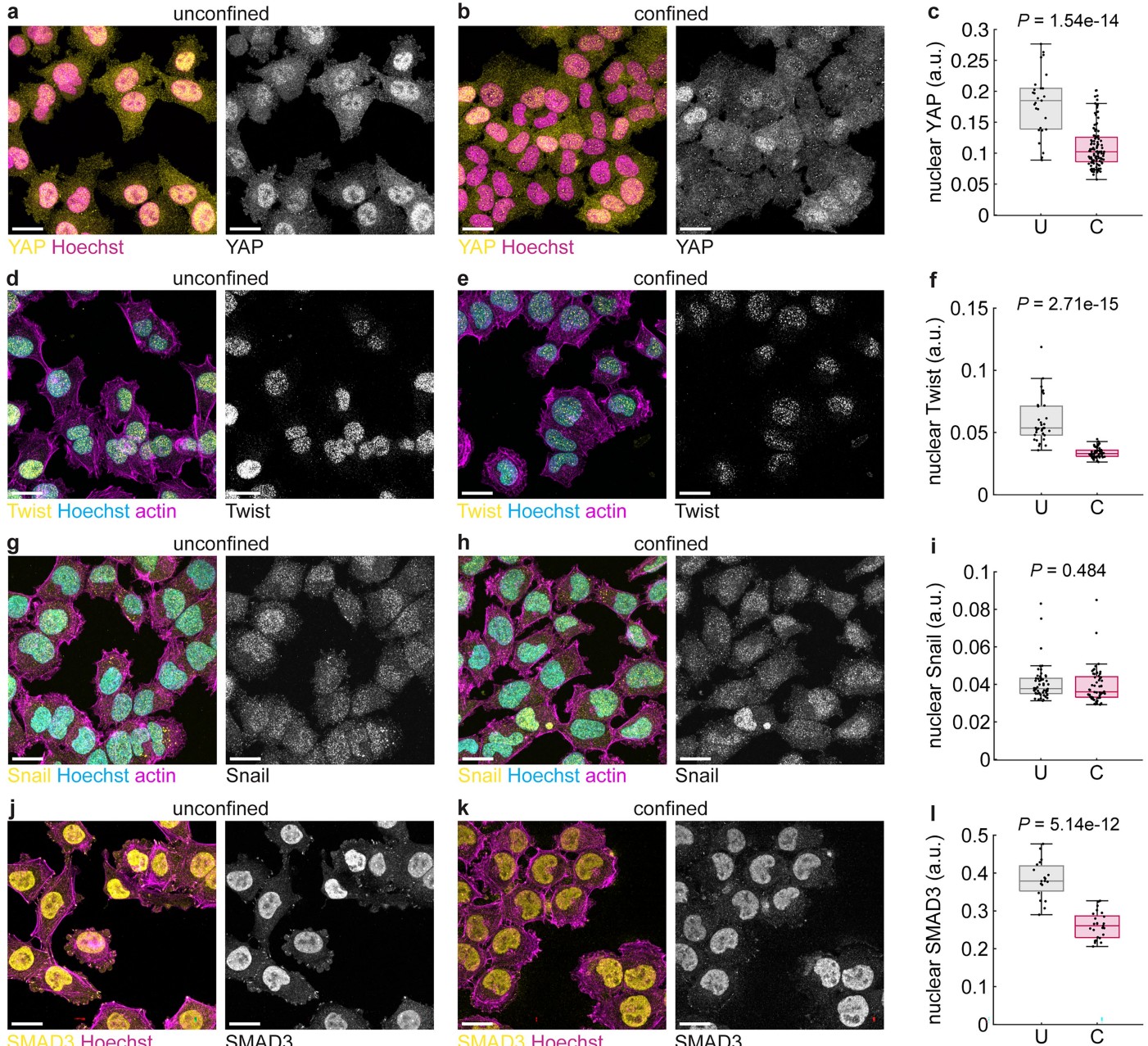

**Extended Data Fig. 7 | Expression of commonly mechanosensitive transcription factors in confined melanoma cells. a-b, d-e, g-h, j-k**. IF images of A375 cells stained with antibodies against YAP (a-b), Twist (d-e), Snail (g-h) or SMAD3 (j-k). Scale bars, 20 μm. **c,f,i,l**. Quantification of intensity in confined/unconfined cells for the indicated markers. Each point represents one cell. Horizontal line, median; box edges, lower and upper quartiles; whiskers, upper and lower limits of data without outliers. U = unconfined; C = confined. **c**. Unconfined: $n$ = 27 cells from 6 images. Confined: $n$ = 111 cells from 6 images. **f**. Unconfined: $n$ = 33 cells from 6 images. Confined: $n$ = 52 cells from 6 images. **i**. Unconfined: $n$ = 56 cells from 6 images. Confined: $n$ = 54 cells from 6 images. **l**. Unconfined: $n$ = 19 cells from 4 images. Confined: $n$ = 25 cells from 6 images.

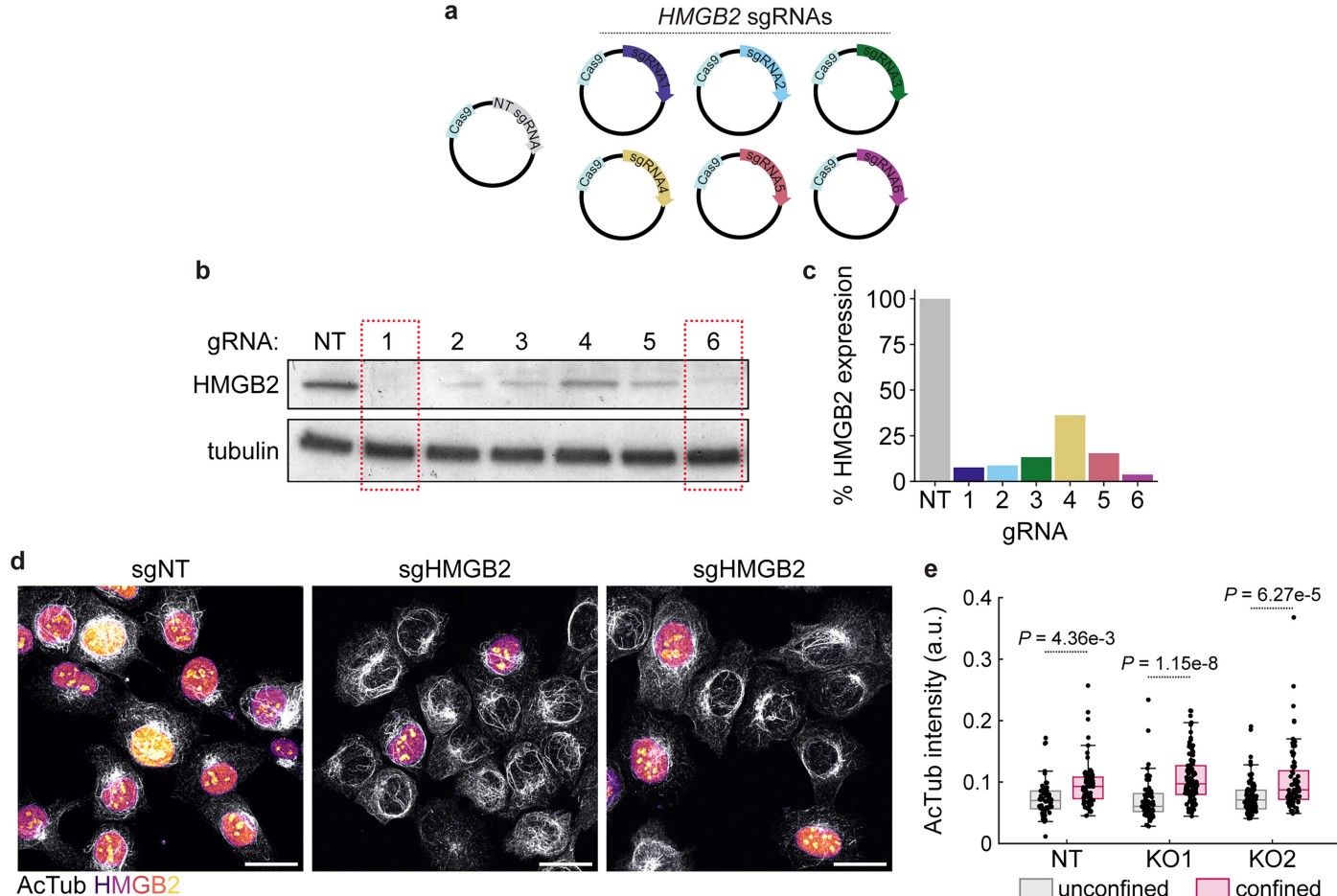

AcTub HMGB2

**Extended Data Fig. 8 | Loss of HMGB2 does not affect perinuclear acetylated tubulin assembly. a**. Schematic showing generation of HMGB2$^{KO}$ A375 cell lines using the LentiCRISPR approach. **b**. Western blot showing expression of HMGB2 in A375 cells stably expressing each of the 6 HMGB2 gRNAs. Red boxes indicate KO cell lines selected for further analyses. **c**. Quantification of HMGB2 expression in b relative to tubulin (loading control). **d**. Immunofluorescence of acetylated tubulin (white) and HMGB2 (red/orange/yellow) intensity in cells expressing the indicated sgRNAs. Scale bars, 25 μm. **e**. Quantification of acetylated tubulin intensity. sgNT unconfined: $n = 58$ cells from 8 images. sgNT confined: $n = 69$ cells from 8 images. sgHMGB2_1 unconfined: $n = 71$ cells from 8 images. sgHMGB2_1 confined: $n = 104$ cells from 8 images. sgHMGB2_2 unconfined: $n = 82$ cells from 8 images. sgHMGB2_2 confined: $n = 77$ cells from 8 images.

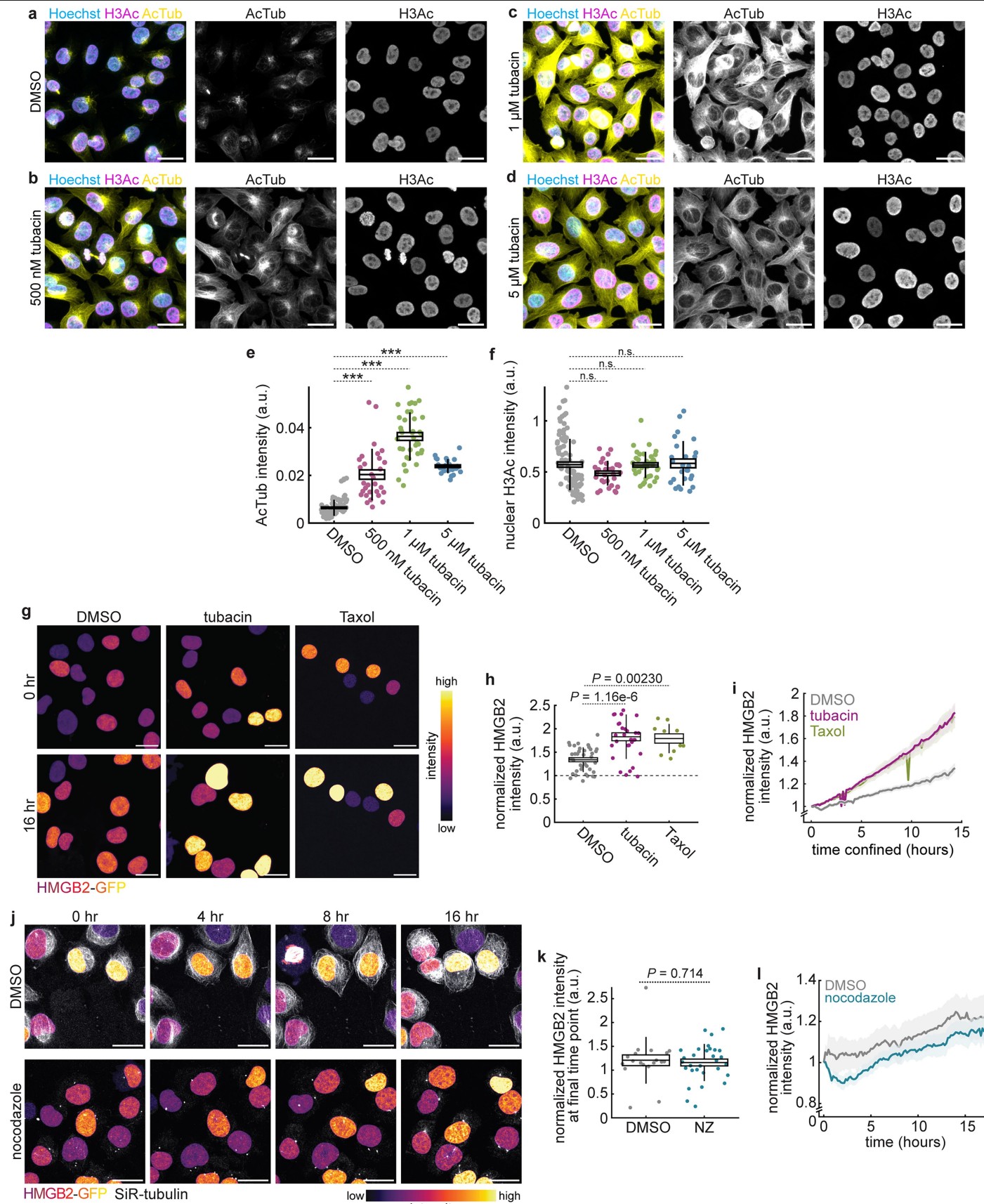

**Extended Data Fig. 9** | See next page for caption.

**Extended Data Fig. 9 | Tubulin stabilization upregulates nuclear HMGB2.**
**a-d**. A375 cells treated with the indicated tubacin concentrations (b-d) or DMSO as a vehicle control (a) and stained with antibodies labelling acetylated histone H3 and acetylated tubulin. Scale bars, 25 μm. **e-f**. Quantification of whole-cell acetylated tubulin intensity (e) and nuclear histone H3 acetylation (g). ***, P < 0.001. n.s., not significant. **g**. HMGB2-GFP accumulation in A375 cells treated with DMSO, tubacin or Taxol. Scale bars, 25 μm. Time after applying confinement is indicated. **h**. HMGB2-GFP intensity in confined cells over time. **i**. HMGB2-GFP intensity per cell at the final time point imaged (-16 h). **h-i**. DMSO: $n$ = 38 cells from 7 movies. Tubacin: $n$ = 31 cells from 7 movies. Taxol: $n$ = 10 cells from 3 movies. **j**. HMGB2-GFP and SiR-tubulin intensity over time for confined cells treated with DMSO or 1 μM nocodazole. Scale bars, 25 μm. **k**. HMGB2-GFP intensity per cell at the final time point imaged (- 16 h). NZ = nocodazole. Horizontal lines, mean; box, SEM; vertical lines, SD. **l**. HMGB2-GFP intensity over time. Error bars, SEM. **k-l**. DMSO: $n$ = 19 cells from 3 movies. Nocodazole: $n$ = 30 cells from 3 movies.

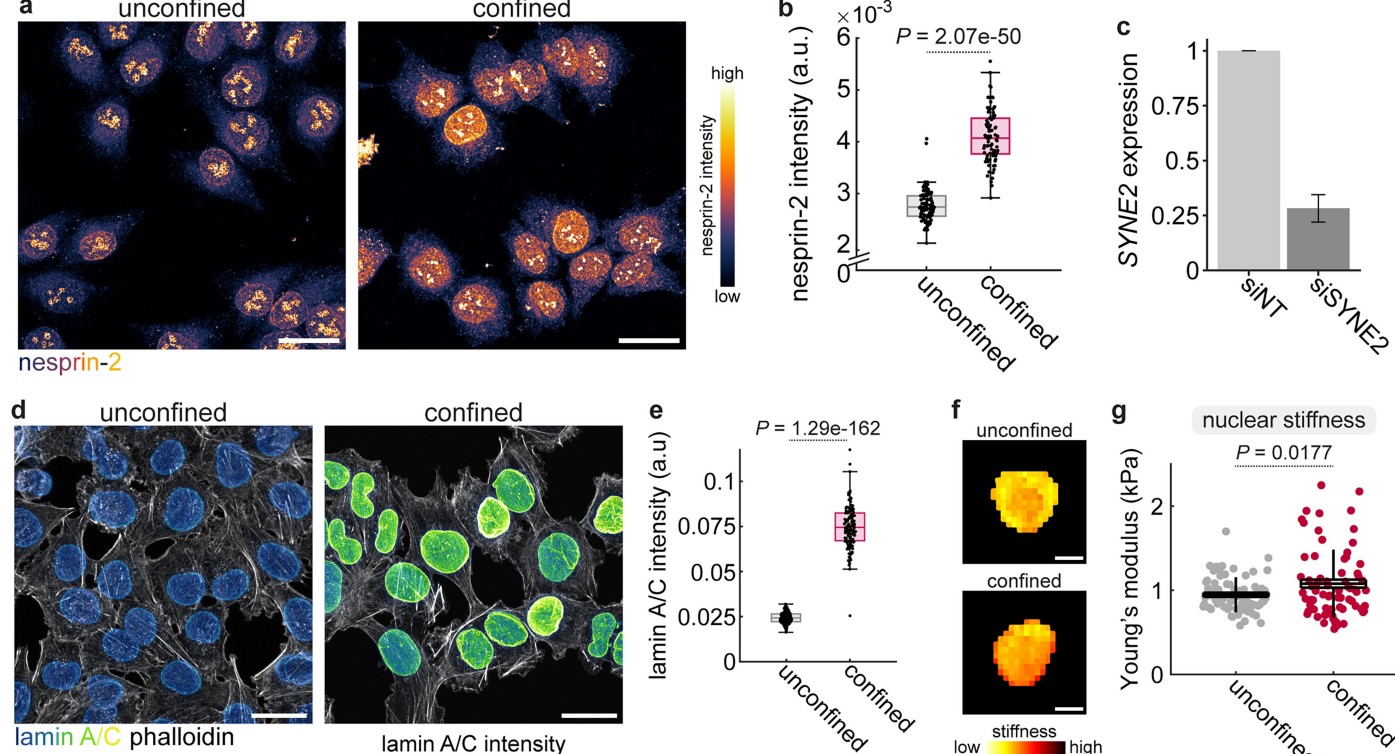

**Extended Data Fig. 10 | The LINC complex is required for HMGB2 enrichment upon confinement. a**. Immunofluorescence images showing expression of nesprin-2 in A375 cells. Scale bars, 25 µm. **b**. Quantification of nesprin-2 intensity in A375 cells. Unconfined: $n = 93$ cells from 8 images. Confined: $n = 85$ cells from 8 images. **c**. Expression of *SYNE2* mRNA in the noted conditions. Expression is normalized to β-actin and the non-targeting control condition. For each cell line, $n = 3$ biological replicates for a total of $n = 12$ technical replicates. **d**. Immunofluorescence images showing expression of lamin A/C (blue/green) and phalloidin in A375 cells. Scale bars, 25 µm. **e**. Quantification of lamin A/C intensity. Unconfined: $n = 180$ cells from 8 images. Confined: $n = 135$ cells from 8 images. **f**. Representative AFM force maps of the nuclear region of A375 cells. Scale bars, 10 µm. **g**. Quantification of nuclear stiffness. Unconfined: $n = 70$ cells from 3 biological replicates. Confined: $n = 71$ cells from 3 biological replicates.

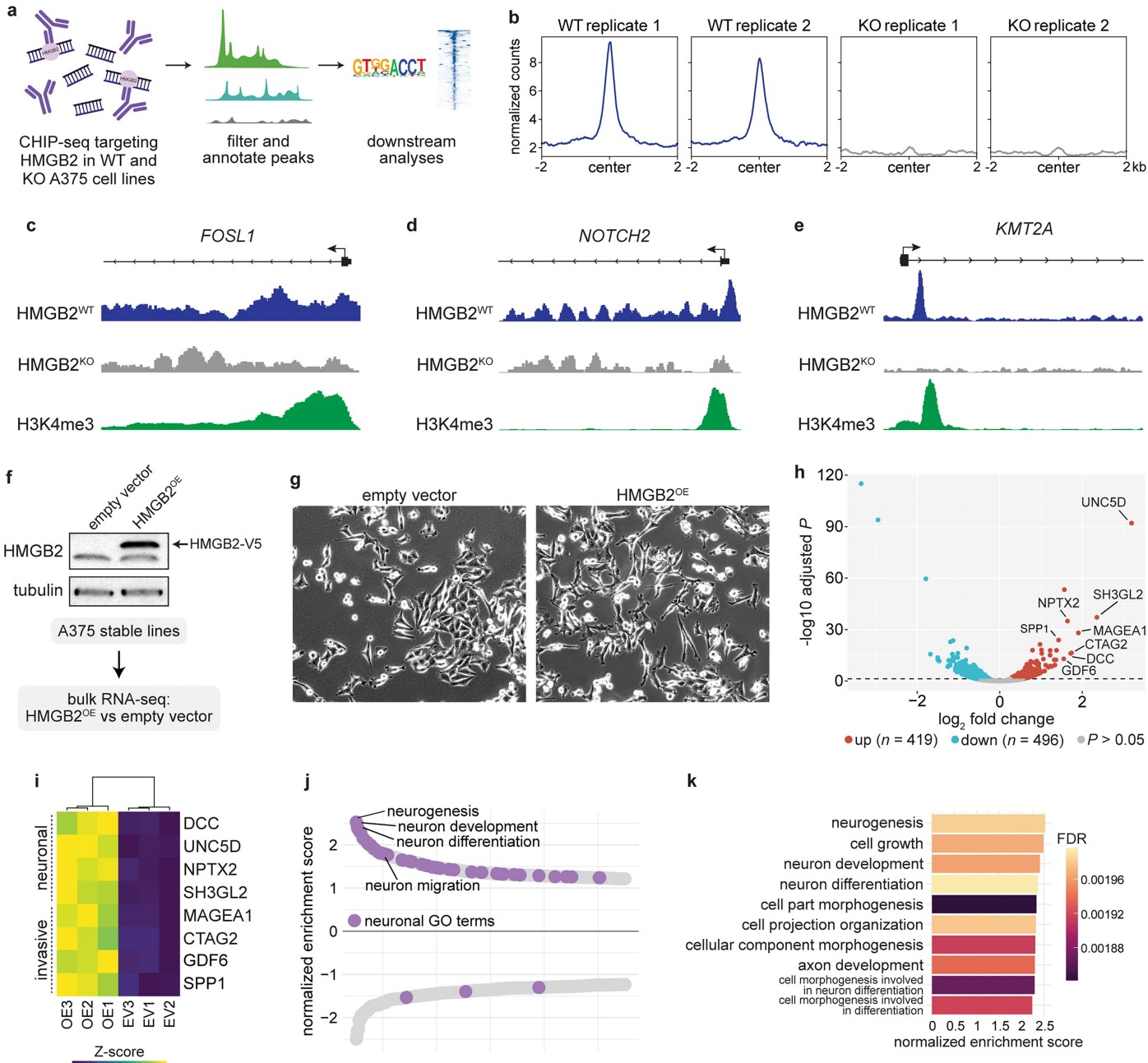

**Extended Data Fig. 11 | ChIP-sequencing and RNA-sequencing identify HMGB2 targets in A375 cells. a**. Schematic detailing ChIP-seq experiment workflow. **b**. Composite plots showing read density at target gene TSS in HMGB2 WT and KO cells. **c-e**. Integrated genome browser tracks representing HMGB2 binding and H3K4me3 signal near the TSS of *FOSL1* (c), *NOTCH2* (d), and *KMT2A* (e). One representative replicate per condition is shown (of two replicates per condition performed). A full list of HMGB2 targets can be found in Supplementary Table 6. **f**. Western blot for HMGB2 (top) and tubulin (loading control, bottom) in stable A375 lines infected with lentivirus encoding HMGB2 or an empty vector control. **g**. Representative images of cells from cell lines indicated in f. **h**. Volcano plot of differentially expressed genes upon HMGB2^OE. **i**. Heatmap of Z-scored expression of selected neuronal and invasive genes across replicates. **j**. Double waterfall plot of top GO biological processes pathways by normalized enrichment score (NES). Neuronal pathways are labeled in purple. **k**. Top 10 GO biological processes pathways by NES upregulated upon HMGB2^OE. Schematic in **a** was created using BioRender (https://biorender.com).

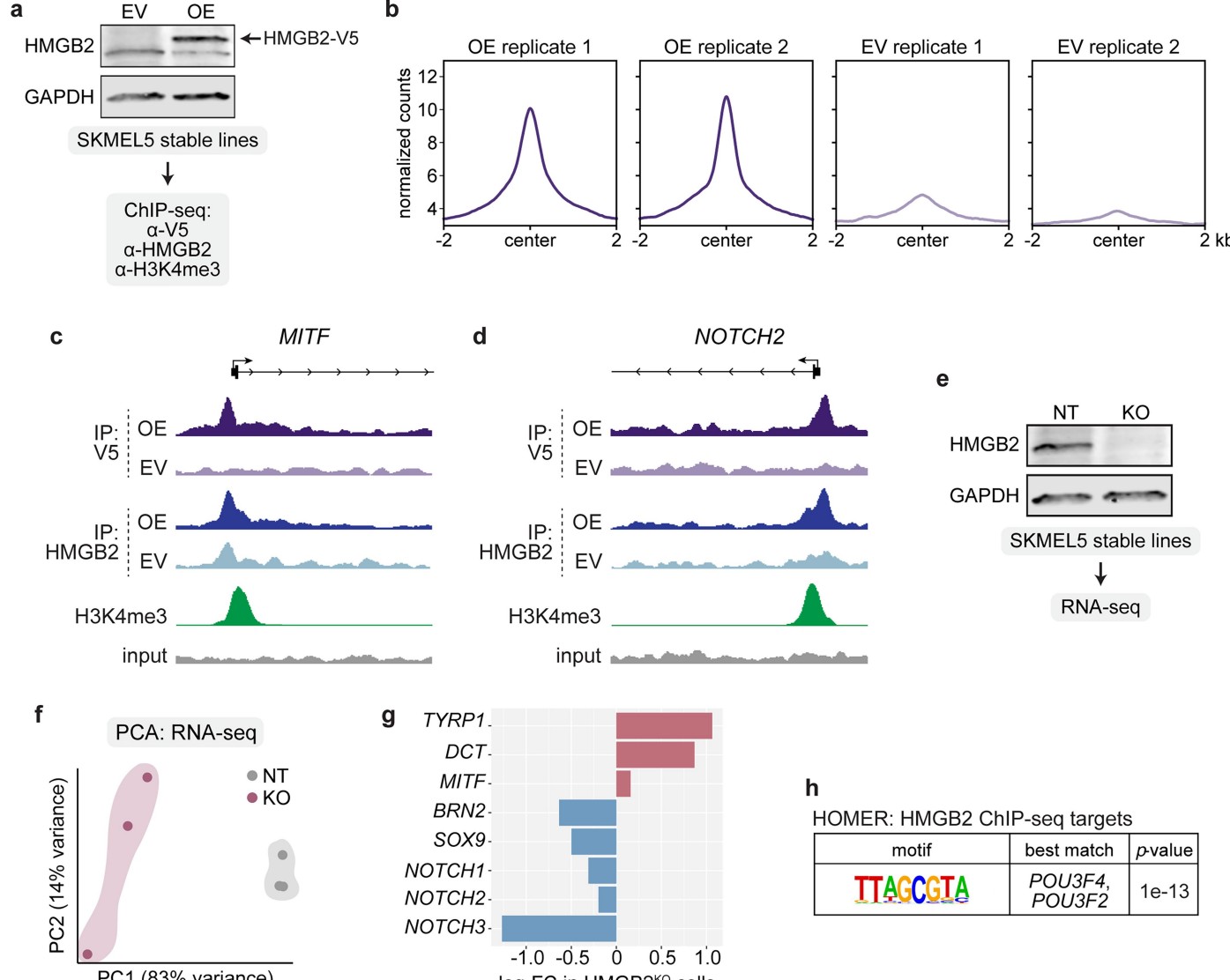

**Extended Data Fig. 12 | Characterization of HMGB2 targets in SKMEL5 cells.**
**a**. Western blot showing expression of V5-tagged HMGB2 in SKMEL5 cells, and ChIP-seq experimental workflow. **b**. Composite plots of read density at target gene TSS in HMGB2 EV and OE cells. **c-d**. Integrated genome browser tracks representing HMGB2-V5 (purple) and HMGB2 (blue) binding and H3K4me3 (green) and input (gray) signal around the TSS of *MITF* (c), and *NOTCH2* (d). One representative replicate per condition is shown (of two replicates per condition performed). A full list of HMGB2 targets in SKMEL5 cells can be found in Supplementary Table S8. **e**. Western blot showing expression of HMGB2 in SKMEL5$^{KO}$ cells and RNA-seq experiment workflow. **f**. Principal component analysis of SKMEL5 RNA-seq samples. **g**. Expression of selected genes in SKMEL5-HMGB2$^{KO}$ cells relative to SKMEL5-NT cells. **h**. HOMER de novo motif analysis showing enrichment of a BRN2/POU3F2 motif in the promoter region of HMGB2 targets identified from ChIP-seq.

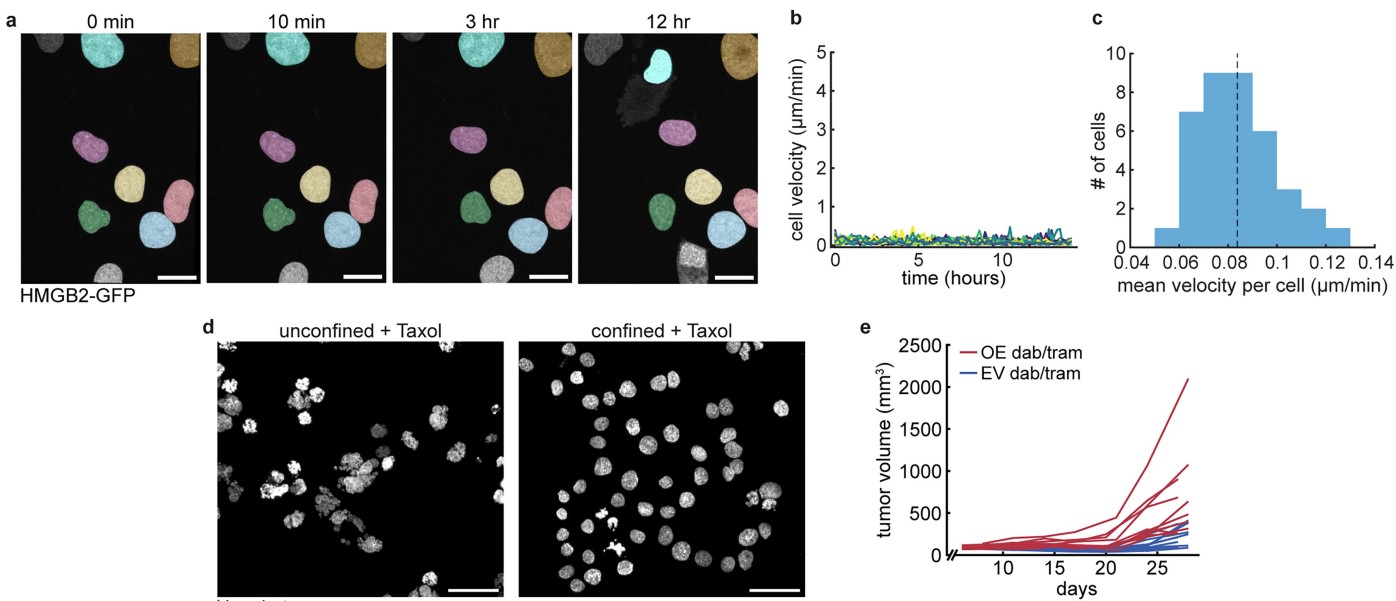

**Extended Data Fig. 13 | Confined cells do not display fast amoeboid migration and are drug tolerant. a**. Stills from time-lapse imaging of confined A375 cells expressing HMGB2-GFP. Nuclei are pseudocolored. Scale bars, 20 μm. **b**. X-Y velocity over time per cell. **c**. Histogram showing mean velocity per cell. Dotted line indicates mean velocity across all cells. **b-c**. n = 38 cells from 7 movies. **d**. Hoechst (DNA) staining of cells treated with Taxol and confined (right) or unconfined control (left). Scale bars, 50 μm. **e**. Spaghetti plot showing tumor volume over time in A375-HMGB2EV (blue) and A375-HMGB2OE (red) mouse xenografts treated with dabrafenib/trametinib. Raw data can be found in Supplementary Table S10.

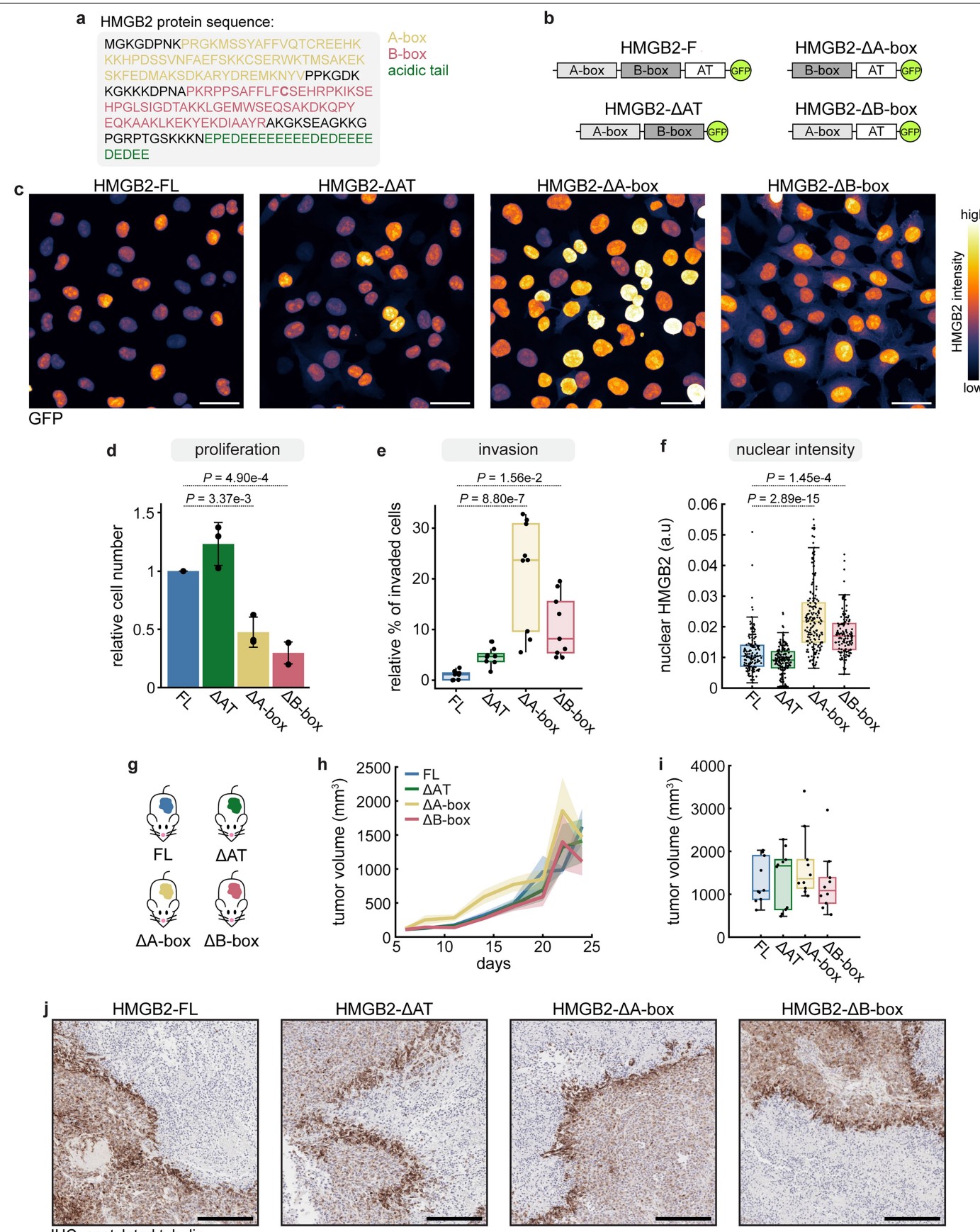

**Extended Data Fig. 14** | See next page for caption.

**Extended Data Fig. 14 | Structure-function analysis of HMGB2. a.** HMGB2 protein sequence (Uniprot). Functional domains are labeled. **b.** Schematics illustrating each deletion construct. **c.** Immunofluorescence images showing expression of GFP-tagged constructs outlined in a. Scale bars, 25 μm. **d-e.** Quantification of in vitro proliferation (d) and invasion (e) assay results for A375 cells expressing the indicated constructs ($n$ = 3 biological replicates). **f.** Quantification of nuclear HMGB2-GFP intensity for the noted constructs. FL: n = 94 cells from 3 images. ΔAT: n = 105 cells from 3 images. ΔA-box: n = 111 cells from 3 images. ΔB-box: n = 82 cells from 3 images. **g.** Schematic detailing in vivo tumor growth assay. **h-i.** Tumor volume over time (h) and at endpoint (i; 22–24 days post transplant). $n$ = 10 mice per condition from 2 biological replicates. **j.** Immunohistochemistry targeting acetylated tubulin in cells at the tumor-microenvironment interface. Mouse cartoons in **g** adapted from Wikimedia, under a Creative Commons licence CC BY-SA 3.0.

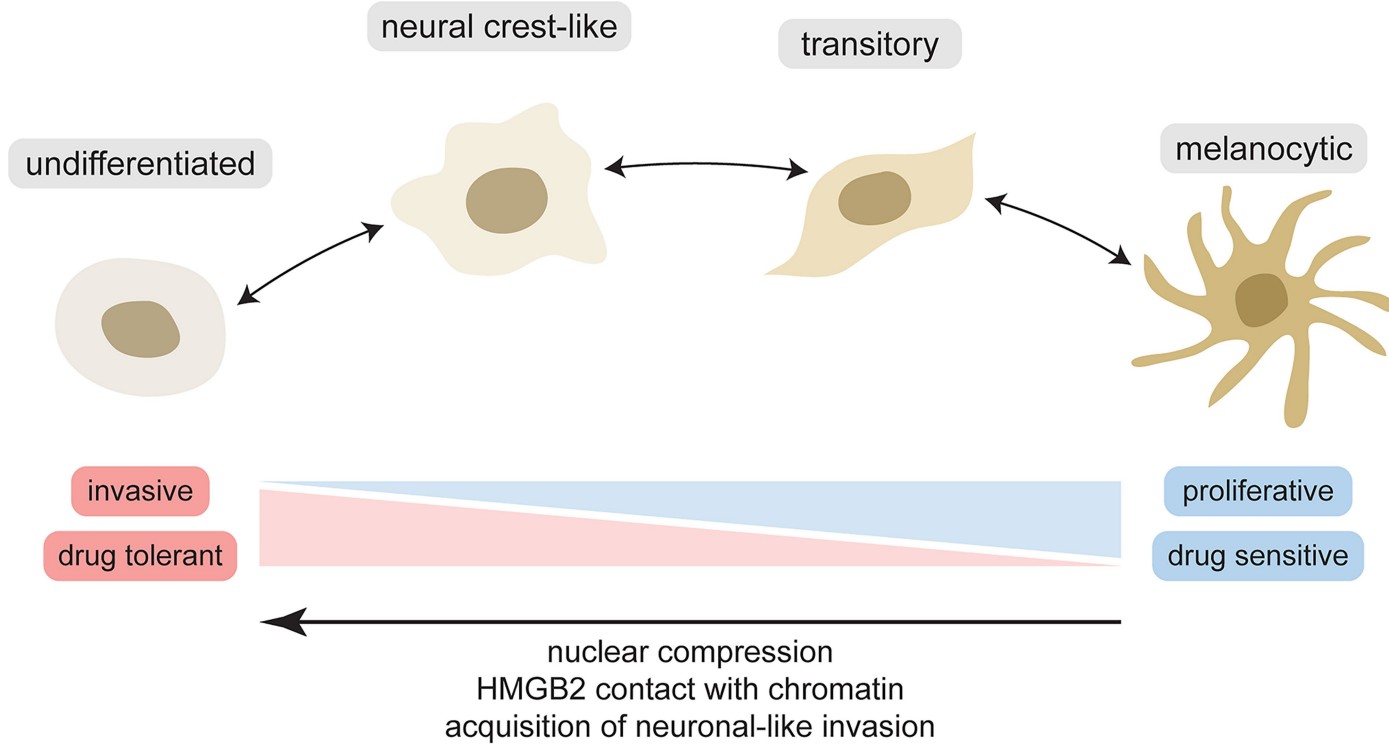

**Extended Data Fig. 15 | Confinement governs phenotypic plasticity in melanoma.** Model for the role of confinement in melanoma phenotype switching: nuclear compression induces HMGB2 contact with chromatin, increasing chromatin accessibility and gene expression at neuronal loci.

# Reporting Summary

## Statistics

For all statistical analyses, confirm that the following items are present in the figure legend, table legend, main text, or Methods section.

| n/a | Confirmed | |
|---|---|---|
| ☐ | ☒ | The exact sample size (*n*) for each experimental group/condition, given as a discrete number and unit of measurement |
| ☐ | ☒ | A statement on whether measurements were taken from distinct samples or whether the same sample was measured repeatedly |
| ☐ | ☒ | The statistical test(s) used AND whether they are one- or two-sided *Only common tests should be described solely by name; describe more complex techniques in the Methods section.* |
| ☒ | ☐ | A description of all covariates tested |
| ☐ | ☒ | A description of any assumptions or corrections, such as tests of normality and adjustment for multiple comparisons |
| ☐ | ☒ | A full description of the statistical parameters including central tendency (e.g. means) or other basic estimates (e.g. regression coefficient) AND variation (e.g. standard deviation) or associated estimates of uncertainty (e.g. confidence intervals) |
| ☐ | ☒ | For null hypothesis testing, the test statistic (e.g. *F*, *t*, *r*) with confidence intervals, effect sizes, degrees of freedom and *P* value noted *Give P values as exact values whenever suitable.* |
| ☒ | ☐ | For Bayesian analysis, information on the choice of priors and Markov chain Monte Carlo settings |
| ☒ | ☐ | For hierarchical and complex designs, identification of the appropriate level for tests and full reporting of outcomes |
| ☐ | ☒ | Estimates of effect sizes (e.g. Cohen's *d*, Pearson's *r*), indicating how they were calculated |

*Our web collection on statistics for biologists contains articles on many of the points above.*

## Software and code

Policy information about availability of computer code

| Data collection | Zen Black v2.3 SP1 software was used to acquire imaging data. |
|---|---|
| Data analysis | Custom R and MATLAB code used for analysis. R versions: 4.3.1; MATLAB versions: R2021b and R2023b. Other software/packages used for bioinformatics analyses: Seurat v4.4.0 and 5.0.1; HOMER v4.5 and 4.11, fgsea v1.26, TrimGalore v0.4.5 and 0.6.7, FastQC v0.11.5 and v0.12.1, cutadapt v1.15 and v4.0, bowtie2 v2.3.4.1 and v2.3.5.1, Picard v2.16, deepTools v3.3.0 and v3.5.1, clusterProfiler v4.10.0, DESeq2 v1.42, featureCounts v1.6.1. Additional software/plugins used for image analysis: CellProfiler v4.2.5, TrackMate v7.11.1, Fiji v2.14, Spectronaut v18.5. All code used for analysis and plotting is available at github.org/mvhunter1/Hunter_2024. |

For manuscripts utilizing custom algorithms or software that are central to the research but not yet described in published literature, software must be made available to editors and reviewers. We strongly encourage code deposition in a community repository (e.g. GitHub). See the Nature Portfolio guidelines for submitting code & software for further information.

## Data

Policy information about availability of data

All manuscripts must include a data availability statement. This statement should provide the following information, where applicable:
- Accession codes, unique identifiers, or web links for publicly available datasets
- A description of any restrictions on data availability
- For clinical datasets or third party data, please ensure that the statement adheres to our policy

| Raw and processed RNA-seq, ChIP-seq, and ATAC-seq data generated in this study have been deposited to the Gene Expression Omnibus (GEO) under accession |
|---|

number GSE253803. Human melanoma scRNA-seq data was obtained from GEO accession number GSE115978. The TurboID proteomics data have been deposited to the ProteomeXchange Consortium via the PRIDE partner repository with the dataset identifier PXD060265. All other relevant data supporting the key findings of this study are available within the article and its Supplementary Information files or from the corresponding authors upon request.

# Research involving human participants, their data, or biological material

Policy information about studies with [human participants or human data](). See also policy information about [sex, gender (identity/presentation), and sexual orientation]() and [race, ethnicity and racism]().

| | |
|---|---|
| Reporting on sex and gender | No human data was generated in this study. |
| Reporting on race, ethnicity, or other socially relevant groupings | No human data was generated in this study. |
| Population characteristics | No human data was generated in this study. |
| Recruitment | No human data was generated in this study. |
| Ethics oversight | No human data was generated in this study. |

Note that full information on the approval of the study protocol must also be provided in the manuscript.

# Field-specific reporting

Please select the one below that is the best fit for your research. If you are not sure, read the appropriate sections before making your selection.

[✗] Life sciences    [ ] Behavioural & social sciences    [ ] Ecological, evolutionary & environmental sciences

For a reference copy of the document with all sections, see [nature.com/documents/nr-reporting-summary-flat.pdf]()

# Life sciences study design

All studies must disclose on these points even when the disclosure is negative.

| | |
|---|---|
| Sample size | While no formal power calculation was performed, in all cases we aimed for the sample size to be as large as possible within the technical confines of the experiment and availability of animals. |
| Data exclusions | No data was excluded from the analysis other than quality control filtering. |
| Replication | RNA-seq and ATAC-seq data has not been replicated due to technical/cost limitations, however 3 technical replicates were performed for each condition in all experiments. For all other experiments, at least n=3 biological replicates were assayed. |
| Randomization | No randomization was done as in all cases there was only one experimental group. |
| Blinding | No blinding was done as in all cases there was only one experimental group. |

# Reporting for specific materials, systems and methods

We require information from authors about some types of materials, experimental systems and methods used in many studies. Here, indicate whether each material, system or method listed is relevant to your study. If you are not sure if a list item applies to your research, read the appropriate section before selecting a response.

## Materials & experimental systems

| n/a | Involved in the study |
|---|---|
| [ ] | [✗] Antibodies |
| [ ] | [✗] Eukaryotic cell lines |
| [✗] | [ ] Palaeontology and archaeology |
| [ ] | [✗] Animals and other organisms |
| [✗] | [ ] Clinical data |
| [✗] | [ ] Dual use research of concern |
| [✗] | [ ] Plants |

## Methods

| n/a | Involved in the study |
|---|---|
| [ ] | [✗] ChIP-seq |
| [✗] | [ ] Flow cytometry |
| [✗] | [ ] MRI-based neuroimaging |

## Antibodies

| | |
|---|---|
| Antibodies used | Primary antibodies used were: rabbit anti-HMGB2 (abcam, ab124670 - for HMGB2 IF), rabbit anti-HMGB2 (Millipore Sigma, HPA053314 - for HMGB2 Western blot), rabbit anti-HMGB1 (abcam, ab18256), rabbit anti-HMGA1 (abcam, ab129153), mouse anti-α- |

tubulin (Millipore Sigma, CP06), chick anti-β-tubulin (Novus Biologicals, NB100-1612), mouse anti-acetylated tubulin (Millipore Sigma, 6793), rabbit anti-acetylated tubulin (Cell Signaling Technologies, CST 5335), rat anti-tyrosinated tubulin (Millipore Sigma, MAB1864-I), mouse anti-polyglutamylated tubulin (Millipore Sigma, T9822), mouse anti-GFP (abcam, ab1218), rabbit anti-H3Ac (Millipore Sigma, 06-599), mouse anti-Annexin V (Santa Cruz, sc-74438), rabbit anti-cleaved caspase-3 (CST 9661), rabbit anti-cleaved PARP (CST 5625), rabbit anti-YAP (CST 14074), mouse anti-Twist (abcam 50887), rabbit anti-Snail (CST 3879), rabbit anti-SMAD3 (abcam ab40854), rabbit anti-SYNE2 (abcam ab204308), rabbit anti-S100a6 (abcam ab204028), mouse anti-BRAF[V600E] (abcam ab228461), rabbit anti-acetylated tubulin (abcam ab179484), rabbit anti-HMGB2 (abcam ab67282 - for ChIP-seq only), rabbit anti-V5 (abcam ab9116), rabbit anti-H3K4me3 (Epicypher 13-0041). All primary antibodies were used at 1:200.

| Validation | All antibodies are commonly used in our lab and have been validated by the suppliers. |

# Eukaryotic cell lines

Policy information about cell lines and Sex and Gender in Research

| Cell line source(s) | A375, SK-MEL-5, Panc-1, MiA-PaCa-2, HTB-4, HTB-9 and HEK 293T cells were obtained from ATCC. |
| Authentication | A375, SK-MEL-5, Panc-1, MiA-PaCa-2, HTB-4, HTB-9 and HEK 293T cells were authenticated using Short Tandem Repeat profiling at ATCC. |
| Mycoplasma contamination | Cells were routinely tested to be free of mycoplasma. |
| Commonly misidentified lines (See ICLAC register) | None of the cell lines used in this study are commonly misidentified. |

# Animals and other research organisms

Policy information about studies involving animals; ARRIVE guidelines recommended for reporting animal research, and Sex and Gender in Research

| Laboratory animals | Zebrafish (Danio rerio) - Genotype: casper; mitfa-BRAFV600E; p53-/-; mitfa-/-. Age: 6-12 months. Mouse (Mus musculus): athymic, 6-8 week old females. |
| Wild animals | No wild animals were used in the study. |
| Reporting on sex | Sex was not a variable in the zebrafish studies and thus was not controlled for. Female mice were used for mouse experiments. |
| Field-collected samples | No field-collected samples were used in the study. |
| Ethics oversight | All animal procedures were approved by the Memorial Sloan Kettering Cancer Center Institutional Animal Care and Use Committee (protocol #12-05-008). |

Note that full information on the approval of the study protocol must also be provided in the manuscript.

# Plants

| Seed stocks | No plant material was used in the study. |
| Novel plant genotypes | No plant material was used in the study. |
| Authentication | No plant material was used in the study. |

# ChIP-seq

## Data deposition

[x] Confirm that both raw and final processed data have been deposited in a public database such as GEO.

[x] Confirm that you have deposited or provided access to graph files (e.g. BED files) for the called peaks.

| Data access links *May remain private before publication.* | The ChIP data has been added to our existing GEO repository GSE253803. |
| Files in database submission | FASTQ and bed files. |

| Genome browser session (e.g. UCSC) | No longer applicable. |

## Methodology

| Replicates | 2 replicates were performed for each experimental group and negative control. |

| Sequencing depth | The libraries were sequenced on an Illumina NovaSeq 6000, with ~30-40 million 100 bp paired-end reads per library. |

| Antibodies | anti-HMGB2: abcam 67282<br>anti-V5: abcam 9116<br>rabbit anti-H3K4me3: Epicypher 13-0041 |

| Peak calling parameters | To ascertain enriched regions, MACS2 was used with a p-value setting of 0.001 and run against a matched control for each condition. A peak atlas was created by combining the superset of all peaks using the 'merge' function in the BEDTools suite v2.29.2. Read density profiles were created using deepTools 'bamCoverage' v3.3.0, normalized to 10 million uniquely mapped reads and with read pileups extended to 200 bp. Version 1.6.1 of featureCounts was used to build a raw counts matrix and DESeq2 was used to calculate differential enrichment for all pairwise contrasts for experiments with replicates. |

| Data quality | For single sample data, MACS2 was run by swapping bams of different conditions to find differential regions. Peak-gene associations were created by assigning all intragenic peaks to that gene, while intergenic peaks were assigned using linear genomic distance to transcription start site. |

| Software | TrimGalore (v 0.4.5), cutadapt (v 1.15), FastQC (v 0.11.5), bowtie2 (v 2.3.4.1), PicardTools (v 2.16.0), MACS2 (v 2.2.9.1), BEDTools (v 2.29.2), deepTools (v 3.3.0), featureCounts (v 1.6.1), HOMER (v 4.5). |

