## [Peer Review File · Nature]

Mechanical confinement governs phenotypic plasticity in melanoma

Corresponding Author: Professor Richard White

Version 1:

Reviewer comments:

Referee #1

(Remarks to the Author)

Mechanical confinement governs phenotypic plasticity in melanoma.
Hunter et al, 2023. Nature.

Summary: The manuscript by Hunter et al identifies HMGB2 as a mediator of the neuronal drug tolerant melanoma state using zebrafish models of melanoma and human samples. The numerous novel approaches developed in this manuscript is impressive and provides solid evidence for the role of mechanical confinement in driving a neuronal invasive phenotype. Thorough characterization of how compaction leads to phenotype switching is both biologically and technologically innovative. However, the manuscript could be improved by more data supporting the role of nuclear confinement leading to the invasive neuronal phenotype, how HMGB2 mechanistically acts in driving this process, and how this phenotype leads to drug tolerance. These data would be required to publish in Nature.

Major:

1. The authors utilize an A375 cell line which are in an MITF low state and therefore already represent the less differentiated more invasive phenotype. The manuscript would then benefit from comparing HMGB2 activation in an MITF high cell state using ChIP paralleled with RNA sequencing. This would answer if HMGB2 upregulation can drive MITF repression and allow for a better mechanism of action of HMGB2.
2. More mechanistic work is required to claim nuclear confinement increases drug tolerance and invasion. To address, zebrafish tumors overexpressing HMGB2 should be treated in vivo to quantify delayed therapeutic response (drug tolerance) and/or transplanted to quantify migration (invasion).
3. It remains unclear how HMGB2 mediates this phenotype and how to disentangle its effects on chromatin vs. binding tubulin filaments. As above, it is critical to purify chromatin from cells with and without compression and examine the binding of HMGB2. Further, HMGB2 is known to bind insulating elements on chromatin thereby affecting 3D chromatin structure. ATAC-sequencing from Figure 4 can be analyzed to define chromatin compartments (see: PMID26316348).
4. To further refine the mechanism of action of HMGB2, functional domain studies with HMGB2 mutations in human cell lines should be used to identify which domain on HMGB2 mediate these effects and if those domains are separate from the domains act on chromatin. Relevance of identified mutations should be confirmed in zebrafish primary tumors to quantify the number of cells with a neuronal phenotype at the invasive tumor interface.
5. The role of confinement driving phenotype switching is novel and insightful for the field. The authors should consider exploring if HMGB2 is necessary for confinement or if this compaction-driven neural phenotype can arise independently of HMGB2. This can be achieved by knockout HMGB2 in A375 and SKMEL5 cells to quantify limited phenotype switching upon compression would confirm HMGB2 as a driver of this process.

Minor:

1. Figure 1B is composed of patients that have are naïve or ICB resistant. This figure would benefit from the addition of a supplemental panel dissecting the naïve vs. resistant patients to show that the neuronal phenotype is exhibited during tumor development and retained during treatment resistance.
2. In accordance with this, histology of patient samples showing elongated nuclei at the invasive edge in humans would nicely parallel Figure 1J, 1K.
3. Figure 5 would benefit from a discussion of how hmgb/hmgb knockout in zebrafish tumor altered invasiveness.

Referee #2

(Remarks to the Author)

This manuscript by Hunter and colleagues combines a zebrafish model of melanoma with an in vitro melanoma cell line to propose that mechanical stress to cells in the tumor's periphery induces a particular phenotype based on HMGB2 upregulation. This work is a nice continuation of their previous study (Hunter et al, Nat Comms 2021) where this "interface" melanoma subpopulation was identified and first characterised. Overall, this is an interesting study, taking a mechanobiology approach on a specific subset of melanoma cells that can be of high therapeutic interest. Data are well represented, the text is very clearly written, and the approach to the problem interesting in itself.

However, I find that both the mechanistic and the molecular insights provided by the authors fall short from achieving three important things: First, from identifying the actual molecular components of the pathway that triggers HMGB2 upregulation; second, from identifying the mode-of-action via which HMGB2 gives rise to this particular phenotype (if this is indeed chromatin-bound HMGB2-driven remains unclear on the basis of the data); and third, from identifying a druggable target in any of the above pathways that would render their observation clinically actionable (even at the level of an in vivo model). Nevertheless, I hope that the more detailed comments I am offering below will be of some use to the authors.

Secondary comments:

- UMAP representation of scRNA-seq data can be highly misleading due to how UMAP directionality reduction is performed. I would advise the authors to use tSNE representations instead.
- The elongated nuclei in Fig. 1j,k are not necessarily also more flat, an attribute that can only be shown via a 3D imaging approach.
- The similarity claimed for gene expression changes in melanoma cells lines under mechanical confinement and the melanoma "interface" cells in vivo is not sufficiently back up by GO term analysis in Fig. 1n. The authors should provide 1-to-1 correlation of gene expression changes between the in vitro and the in vivo model (using central melanoma cells as control for the latter comparison). Then we can see which genes are indeed convergently and which divergently regulated.
- It would be important to understand the extent of HMGB2 upregulation due to mechanical stress. The data presented in Fig. 3a,c,f,j do not make this clear. For example, scaled expression in Fig. 3a is not immediately interpretable (can we see counts and log2FC?), while in Fig. 3g the high-low intensity scale seems arbitrary. Then, looking at Fig. 3h-j (where I am not sure what the Hoechst intensity increase really means, cannot be simply compaction), the effect seems to be <2-fold increase. If this is not also reflected in the RNA-seq data, then is it rather protein stabilization? Panel 3f would suggest a less steep increase in the model though. It would also be important to understand how HMGB2 was prioritized in the first place amongst many other (possibly even stronger upregulated) factors?
- Moreover, in Fig. 3c the "interface" levels should be compared not to bulk tumor abut to the central, unconfined part only. Moreover, the UMAP plot in Fig. 3b suggests that expression is not really higher in a per cell level in "interface" cells, but rather that most cells show this increase. The authors should therefore ask whether HMGB2-high cells are actually selected for this spatial destination by looking at the evolution of the tumor from earlier stages until this one here in their fish model.
- Please note that what appears to be nucleolar staining in Fig. 3g (which is the main point of fluorescent intensity increase) is an artefact of formaldehyde fixation of HMGBs in general. It is important that this experiment is repeated using more suitable fixation parameters (see Zirkel et al, Mol Cell 2018 and Mensah et al, Nature 2023).
- I would suggest caution concerning the acetylation modulating experiments. HMGBs in general are known to be post-translationally modified by acetylation at various sites along their length, which can change their DNA binding and localization properties. It would therefore be important to control for this in all these experiments using tubacin/taxol/NZ.
- I am probably misunderstanding something in the interpretation of the data in Fig. S5. Since early NZ treatment does abolish the perinuclear MT mesh, but accumulation persists, wouldn't this suggest that the two events are not coupled? The authors seem to suggest otherwise in the main text.
- The FRAP analysis, at best, shows some mild increase in chromatin association with HMGB2. I am also not sure how the change of nuclear volume/shape during confinement might affect FRAP outcomes? Therefore, to convincingly show such an effect, one would need to perform ChIP-seq (with spike-ins) to quantify binding patterns genome-wide. The HMGB2-OE experiments are welcome, but in the absence of an HMGB2 motif, it cannot be deduced that it is this factor that directly associates with the de novo opened positions. In fact, HMGBs have been shown to also bind RNA pretty specifically and this might explain more of the phenotype than chromatin effects (see Sofiadis et al, Mol Syst Biol 2021 for an example of HMGB1).
- The final result of the manuscript is intriguing, as higher levels of HMGB2 across multiple different types of cancer actually correlate with increased proliferation and anti-senescent gene expression programs. It is therefore surprising to see these anti-proliferative effects, and can only hypothesize that the mechanical stress of confinement (somehow) imposes on the usual HMGB2 effects. However, the authors do not dissect how this might occur and only present us with the observation.

A. Papantonis

Referee #3

(Remarks to the Author)

Hunter et al use a zebrafish model of melanoma to study the mechanobiology of a subset of rare cancer cells localized at the interface between the tumor and its surroundings. These cells display high upregulation of genes characteristic of an invasive state as well as markers of neuronal development, a profile that is also found in published datasets of human melanoma patients. The authors show that these cells display highly elongated nuclei, which leads them to hypothesize that nuclear deformation plays a key role in defining the cellular transcriptional state and function. To test this hypothesis, they subject a micropatterned human melanoma cell line to compressive stress and find that this confinement is sufficient to express the neuronal pathways that were upregulated in interface cells. In addition, they show that both interface cells *in vivo* and compressed cells *in vitro* assemble a network of stable acetylated microtubules at the nuclear periphery. This type of nuclear cage has been observed previously in other cell types in response to mechanical compression and is thought to shield the nucleus from mechanical stresses during migration through narrow pores. The authors then identify that, as in other cancer types, the high mobility group (HMG)-family proteins (mainly HMGB2) is upregulated in interface cells. This protein, which bends DNA in a sequence unspecific manner for transcription factor binding, increases its expression in response to compression in a way that is linked to microtubule stability. Using FRAP experiments, the authors identify a significant increase in the bound fraction of HMGB2 in confined cells. Using ATAC-seq, they show that upregulation of HMGB2 increases chromatin accessibility at neuronal loci, promoting the expression of genes that have been associated with mesenchymal migration. They also show *in vitro* and *in vivo* that HMGB2 contributes to phenotypic switching between proliferative and invasive phenotypes. Finally, they report that the confined cells are resistant to taxol-induced cell death, providing a link between confinement and therapy.

Overall, this is an insightful paper that connects a mechanobiological feature (nuclear deformation) with phenotypic switching in melanoma. Mechanisms are dissected in detail by combining *in vivo* zebrafish work and *in vitro* biophysical assays. The paper is clearly written and the narrative is solid, but some conclusions are insufficiently supported by the experiments. My major points are the following:

- 1) Nuclear deformation has been previously associated with nuclear translocation of transcription factors and regulators. YAP is the best known of them, but others associated with EMT have also been shown to be mechanosensitive (twist, snail, smad3, etc). Translocation of these factors might have a relevant role in phenotypic switching and in driving melanoma invasion. The authors should consider this possibility in their model and test it experimentally.
- 2) Compression of cancer cells is sufficient to drive their migration independently of transcriptional changes, see Lomakin et al (Science, 2020), Venturini et al. (Science, 2020), Conti et al (biorxiv). The authors should rule out that these acute changes do not dominate the phenotypic switching they report. The proposed mechanism is compelling, but how dominant is it? Can the authors measure cell migration during compression as in the papers cited above?
- 3) The relevance of the authors' findings in the advance of cancer cell biology is clear, but its direct impact on human cancer is not. The authors should show that some of the key features of the mechanobiology of interface cells are present in human (or at least mouse) melanoma. For example, they should show that the correlation between nuclear shape, MT perinuclear localization, and HMGB2 overexpression found in zebrafish are also present in human/mouse samples.
- 4) Large nuclear deformations have been identified by pathologists for decades in different types of cancers. Are the authors' results specific to melanoma? This could be easily addressed using cells from other types of cancer and subject them to compression to assess the perinuclear accumulation of MTs and the overexpression of HMGB2.
- 5) How the microtubule cage drives changes in expression of HMGB2 should be clarified. The authors carried out experiments showing that strengthening the microtubule cage leads to HMGB2 over-expression, but the mechanism is unclear. Can they also weaken the cage? Is the microtubule cage needed at all for HMGB2 over-expression?
- 6) The higher invasive capacity of HMGB2OE cells should be tested functionally (the snapshots of Fig S6 are insufficiently conclusive). There are many simple assays *in vitro* that can ascertain whether these cells are indeed more invasive.

Minor

- 1) Line 152: please correct "unconfined and unconfined"
- 2) The authors qualify the interface cells are "rare". Can they provide a quantitative statement of how rare they are?

Version 2:

Reviewer comments:

Referee #1

(Remarks to the Author)

Mechanical confinement governs phenotypic plasticity in melanoma.
Hunter et al, 2023. Nature.

Summary: The revised manuscript by Hunter et al. has identified a transcriptional network driving a pro-invasion phenotype

in melanoma that is mediated by HMGB2. This revision has strengthened the overall manuscript particularly the addition of multiple ChIP-seq experiments and refining the model of how HMGB2 functions. However, some concerns still arise, particularly regarding drug tolerance, and are detailed below. Upon resolution of these issues, this manuscript would be of broad interest to the readers of Nature and should be accepted.

Major:

1. The addition of the mouse experiment to address drug tolerance (Fig 5o), although appreciated, is not sufficient to conclude that HMGB2 leads to sustained drug tolerance. Individual spaghetti plots for each mouse in the cohort should be shown in supplemental to help clarify if it is a single mouse driving the increase in tumor volume or if it's a cohort effect. Further, it appears that at Day 25 both the control and HMGB2 OE mice are gaining tolerance, although HMGB2 OE is occurring at a faster rate.
2. The co-expression of MITF and BRN2 is known to be an artifact of cell culture in comparison to 3D culture or human melanoma (see PMIDs: 19826052, 18829533, 25132268). In addition, BRN2 is a known repressor of MITF expression (see PMIDs: 18829533, 21435193).
 - a. The manuscript would benefit from staining human samples or harvested zebrafish tumors/mouse xenografts for MITF and BRN2 to show distinct localization.
 - b. The discussion would benefit from more explanation of why MITF was not affected upon nuclear confinement with an upregulation of HMGB2 and BRN2 activity.

Minor:

1. Size of the tumors upon treatment (Fig 5o) should be noted in the methods as they appear quite small.
2. If collected, mouse xenografts should be stained with proliferation, pERK, and other known markers to be modulated following HMGB2 OE and drug treatment to show translation of the identified mechanism to the in vivo mouse model.

Referee #2

(Remarks to the Author)

The authors have carefully addressed the majority of the points I raised. Although I still believe that the HMGB2-RNA interactions are equally important in the context of HMGB2 upregulation, I understand how this might be a challenging bit to investigate. Therefore, I am happy to suggest that the manuscript is published in its current form.

One minor note: I find the filtration of HMGB2 peaks to be excessively strict, thereby resulting in <100 targets. I would suggest that the authors consider a broader set in their analysis.

A. Papantonis

Referee #3

(Remarks to the Author)

The authors have addressed each of my comments thoroughly. While some of their responses reinforce their original conclusions, others suggest that the underlying mechanisms may be more complex than initially proposed. For example, the finding that acetylated tubulin is sufficient but not necessary for HMGB2 enrichment in response to confinement, along with the newly discovered involvement of nesprin, raises alternative mechanistic interpretations rather than the original linear argument linking the tubulin cage to HMGB2. Furthermore, the authors now show that transcription factors and regulators leave the nucleus in response to compression, opening the possibility of additional transcriptional mechanisms besides those studied here. I see this not as a weakness but as a strength. For instance, the translocation of transcription factors from the nucleus to the cytoplasm challenges current understanding in mechanobiology and will attract the interest of many research groups. Overall, I find that this article advances the fields of cancer cell biology and mechanobiology significantly and brings them further together.

I recommend acceptance with just one minor comment.

1) In their discussion of Supp Fig. 9 the authors should emphasize that compression causes the translocation of transcription factors and regulators from the nucleus to the cytoplasm.

Version 3:

Reviewer comments:

Referee #1

(Remarks to the Author)

The authors have sufficiently and thoroughly responded to all comments. While the claim of resistance conferral could still be strengthened, that can be the scope of future work. The manuscript should be accepted as it is of high interest to the melanoma field and the readers of Nature.

Reviewer 1

Summary: The manuscript by Hunter et al identifies HMGB2 as a mediator of the neuronal drug tolerant melanoma state using zebrafish models of melanoma and human samples. The numerous novel approaches developed in this manuscript is impressive and provides solid evidence for the role of mechanical confinement in driving a neuronal invasive phenotype. Thorough characterization of how compaction leads to phenotype switching is both biologically and technologically innovative. However, the manuscript could be improved by more data supporting the role of nuclear confinement leading to the invasive neuronal phenotype, how HMGB2 mechanistically acts in driving this process, and how this phenotype leads to drug tolerance. These data would be required to publish in Nature.

Major:

1. The authors utilize an A375 cell line which are in an MITF low state and therefore already represent the less differentiated more invasive phenotype. The manuscript would then benefit from comparing HMGB2 activation in an MITF high cell state using ChIP paralleled with RNA sequencing. This would answer if HMGB2 upregulation can drive MITF repression and allow for a better mechanism of action of HMGB2.

Response: We have now performed both RNA-seq and ChIP-seq in SKMEL5 (MITF-high) cells. We made a stable SKMEL5 cell line in which *HMGB2* was inactivated using CRISPR, and then performed RNA-seq. We found that loss of *HMGB2* causes cells to upregulate melanocytic genes (*DCT*, *TYRP1*) and downregulate invasive genes (*SOX9*) (**new Fig. S16e-g**), supporting our model in which HMGB2 is required for the invasive/neuronal state. We also profiled HMGB2 promoter binding in SKMEL5 cells using ChIP-seq and found that, as you suggested, HMGB2 binds the *MITF* promoter (**new Fig. S16c**); however, loss of HMGB2 did not significantly affect *MITF* expression ($\log_2FC = 0.157$, $P_{adj} = 0.137$; **new Fig. S16g**).

Based on this, we hypothesized that HMGB2 may instead be regulating expression of other transcription factors associated with plasticity and acquisition of the neuronal/invasive state. To complement our ChIP-seq profiling of HMGB2 in SKMEL5 cells, we also performed ChIP-seq targeting HMGB2 in A375 cells (**new Fig. S14 and Table S6**). We found that HMGB2 robustly bound the promoter of genes associated with Notch signaling in both A375 (**new Fig. S14d and Table S6**) and SKMEL5 (**new Fig. S16d and Table S8**) cells. Notch-family genes were also upregulated by human interface cells (*NOTCH1*, *DLL1*, *DLL3*, *DLK2*; **Table S1**) and confined human melanoma cells (*NOTCH2NLA*, *DLK2*, *DLL4*; **Table S2**). Notch signaling was recently found to drive melanoma invasion through a reciprocal relationship with the pro-invasive transcription factor brain-2 (*BRN2*, also known as *POU3F2*; PMID: 34958806), a classical marker of the melanoma invasive state. *BRN2* also has a well-characterized role in neuronal development and has been referred to as a master neural transcription factor (PMID: 27784708). This suggests that the pro-invasive neuronal phenotype we identified in confined interface cells is related to HMGB2-mediated upregulation of Notch/*BRN2* signaling. In support of this model, a *BRN2/POU3F2* motif was enriched in the promoter region of HMGB2 target genes identified from ChIP-seq (**new Fig. S16h**). Together, these results clarify the role of HMGB2 in invasion, showing that HMGB2 regulates Notch/*BRN2* signaling to promote melanoma invasion and a neuronal phenotype.

2. More mechanistic work is required to claim nuclear confinement increases drug tolerance and invasion. To address, zebrafish tumors overexpressing HMGB2 should be treated in vivo to quantify delayed therapeutic response (drug tolerance) and/or transplanted to quantify migration (invasion).

Response: While we could have done this experiment in zebrafish, we instead did this experiment in mice, as they have more direct translational relevance for this question. We transplanted A375 human melanoma cells overexpressing HMGB2 into mice and treated the resulting tumors with dabrafenib and trametinib, widely used targeted therapies in melanoma. Overexpression of HMGB2 significantly impaired the response of the melanoma cell to dabrafenib/trametinib *in vivo*, relative to tumors expressing baseline levels of HMGB2. We added these data as **new Fig. 5n-o**.

3. It remains unclear how HMGB2 mediates this phenotype and how to disentangle its effects on chromatin vs. binding tubulin filaments. As above, it is critical to purify chromatin from cells with and without compression and examine the binding of HMGB2.

Response: HMGB proteins can mediate their effects either through interactions with chromatin/DNA, or through interactions with RNA as recently described (PMIDs: 30575817, 25937287, 19890330). Our initial FRAP and ATAC-seq data support a role for increased chromatin accessibility in response to HMGB2 upregulation, but we agree that more direct evidence for this mechanism was needed. To address this, we first performed ChIP-seq targeting HMGB2 in confined and unconfined cells, as you suggested. This was a technically very challenging experiment. The confinement device is small, limiting the number of cells we could prepare for ChIP-seq, in addition to the inherent limitations of performing ChIP on a protein that binds DNA significantly less strongly than a typical transcription factor (discussed in PMID: 29706538). While there was some variability across replicates, likely for technical reasons, overall we find that confined cells exhibit increased binding of HMGB2 to chromatin genome-wide (**Fig. R1**).

Figure R1: Differentially enriched HMGB2 targets in confined A375 cells. Number of HMGB2 target peaks quantified in confined cells relative to unconfined cells.

As mentioned above in point 1, we have now also comprehensively performed ChIP-seq for HMGB2 in both A375 (MITF^{Lo}) and SKMEL5 (MITF^{Hi}) cells. What is consistent across both of these datasets is the observation that HMGB2 regulates various members of the Notch/BRN2 axis, well-described regulators of both the invasive and neuronal phenotypes we describe in the manuscript. These data suggest that HMGB2 is regulating phenotype plasticity, at least in part, by shifting the balance to favor transcription factors associated with the invasive cell state.

While these data confirm HMGB2's interaction with chromatin, this still leaves open the question of how mechanical force leads to the increase in HMGB2 to begin with. While it is possible that HMGB2 directly binds tubulin, we do not know of specific data that indicate this. Thus, to identify factors that interact with HMGB2 in melanoma cells that may mediate this mechanically-induced upregulation, we utilized the TurboID proximity labeling system to identify protein interactors of HMGB2 in melanoma cells in an unbiased manner (**new Fig. 3I**). We generated stable A375 cell lines expressing either HMGB2-TurboID or NLS-TurboID fusion proteins, pulsed the cells with 10 mM biotin for 1 hour to induce biotinylation of proteins proximal to the TurboID cassette, and quantified enrichment of biotinylated proteins using mass spectrometry. To identify proteins that

specifically interact with HMGB2 above the ‘background’ of the overall nuclear proteome, for each protein detected we calculated fold enrichment in HMGB2-TurboID relative to NLS-TurboID. One highly enriched protein was nesprin-2 (gene name: *SYNE2*), a component of the LINC (Linker of Nucleoskeleton and Cytoskeleton) complex that physically connects the cytoskeleton, nuclear lamina and chromatin (**new Fig. 3m**). We hypothesized that nesprin-2 cooperates with HMGB2 to respond to mechanical force and stabilize the nucleus against mechanical stress. Accordingly, confined A375 cells upregulated nesprin-2 (**new Fig. S13a-b**), and knockdown of *SYNE2* with siRNA (**new Fig. S13c**) abolished confinement-mediated accumulation of HMGB2 (**new Fig. 3n-o**) and formation of the perinuclear acetylated tubulin network (**new Fig. 3p-q**). While it is likely that there are other potential interactions between HMGB2 and other non-DNA cellular components, these data indicate that the LINC complex component nesprin-2 is one key mediator of HMGB2 upregulation and perinuclear tubulin assembly under mechanical confinement.

Further, HMGB2 is known to bind insulating elements on chromatin thereby affecting 3D chromatin structure. ATAC-seq from Figure 4 can be analyzed to define chromatin compartments (see: PMID26316348).

Response: We agree that knowing the effect of HMGB2 on 3D chromatin structure would be relevant to our model. However, the type of analysis described in the referenced paper is only possible to do on single-cell ATAC-seq data, whereas our data in Fig. 4 was generated from bulk ATAC-seq. While it would certainly be interesting to perform this type of analysis using Hi-C, single-cell ATAC-seq, or similar approaches, we feel this would be best explored in a follow-up study, as we estimate it would take at least a year to complete and fully characterize these effects.

4. To further refine the mechanism of action of HMGB2, functional domain studies with HMGB2 mutations in human cell lines should be used to identify which domain on HMGB2 mediate these effects and if those domains are separate from the domains act on chromatin. Relevance of identified mutations should be confirmed in zebrafish primary tumors to quantify the number of cells with a neuronal phenotype at the invasive tumor interface.

Response: HMGB-family proteins contain three functional domains: two DNA-binding domains (the A-box and B-box) and a third acidic tail region (**new Fig. S18a-b**). Thus, the vast majority of HMGB2 controls DNA-binding, making it difficult, if not impossible, to completely abolish HMGB2’s DNA-binding function in cells. We attempted to do this by generating a construct in which both the A-box and B-box domains were deleted, leaving only the acidic tail region (24 amino acids). However, given its small size, we were unable to successfully express this construct in A375 cells.

We instead assembled constructs in which each individual functional domain was deleted (**new Fig. S18b**) and generated stable A375 cell lines expressing a GFP-tagged version of each construct (**new Fig. S18c**). We performed *in vitro* proliferation and invasion assays to measure the effect of each functional domain on melanoma cell behavior. Cells expressing either of the Δ A-box and Δ B-box constructs were significantly more invasive (**new Fig. S18e**) and less proliferative (**new Fig. S18d**), indicating that either DNA binding domain alone is sufficient to induce the invasive phenotype. We also noted that both the Δ A-box and Δ B-box constructs exhibited significantly higher levels of nuclear expression relative to the full-length construct (**new Fig. S18f**), likely due to their smaller size, and they had a corresponding increase in invasive capacity compared to even the full length (**new Fig. S18e**).

As we were not able to generate constructs completely lacking both DNA-binding domains, when we transplanted A375 cells expressing each of the deletion constructs into mice (**new Fig. S18g**), as expected there was no significant difference in growth rates (**new Fig. S18h-i**) or acetylated tubulin at the invasive front (**new Fig. S18j**). This is likely because these constructs still retain DNA-binding activity and are sufficient to induce invasion, as noted above.

5. The role of confinement driving phenotype switching is novel and insightful for the field. The authors should consider exploring if HMGB2 is necessary for confinement or if this compaction-driven neural phenotype can arise independently of HMGB2. This can be achieved by knockout HMGB2 in A375 and SKMEL5 cells to quantify limited phenotype switching upon compression would confirm HMGB2 as a driver of this process.

Response: As described above, the revised manuscript now contains multiple new lines of evidence directly linking HMGB2 to the invasive neuronal phenotype. Here we summarize the key new data in the revised version:

- (1) As you suggested, we generated both A375 and SKMEL5 HMGB2^{KO} cell lines and profiled the effect on phenotype switching using RNA-seq and ChIP-seq. Loss of HMGB2 caused cells to upregulate melanocytic genes (*DCT*, *TYRP1*) and downregulate invasive genes such as *SOX9* (**new Fig. S16e-g**), supporting our model in which HMGB2 induces a less proliferative/differentiated and more invasive/neuronal state.
- (2) As mentioned in major points 1 and 3 above, our new ChIP-seq data from both A375 and SKMEL5 cells indicate that HMGB2 directly targets Notch/BRN2 signaling to promote melanoma invasion and a neuronal phenotype.
- (3) We added new data showing that overexpression of HMGB2 promotes invasion (**new Fig. 5h-i**) and drug tolerance (**new Fig. 5n-o**).
- (4) As noted above in major point 3, we have generated new proximity proteomics data that provides a mechanism linking confinement to upregulation of HMGB2. This experiment identified the LINC complex component nesprin-2 as a key factor linking confinement-induced perinuclear acetylated tubulin assembly to HMGB2 upregulation (**new Figs. 3l-q** and **S13**). Additionally, we confirmed that the assembly of the perinuclear acetylated tubulin network, a hallmark of the neuronal state, is upstream of HMGB2 by quantifying perinuclear acetylated tubulin in HMGB2^{KO} cells. Assembly of the perinuclear acetylated tubulin network was unaffected by loss of HMGB2 (**new Fig. S10d-e**).

Minor:

1. Figure 1B is composed of patients that have are naïve or ICB resistant. This figure would benefit from the addition of a supplemental panel dissecting the naïve vs. resistant patients to show that the neuronal phenotype is exhibited during tumor development and retained during treatment resistance.

Response: We have added **new Fig. S1d-f** showing that the interface cells are largely from ICB resistant patients.

2. In accordance with this, histology of patient samples showing elongated nuclei at the invasive edge in humans would nicely parallel Figure 1J, 1K.

Response: A similar point was raised by Reviewer 3, major point 3. To address this, we stained human melanoma tissue microarrays (TMAs) to look for evidence of interface cells, characterized by elongated nuclei with high levels of HMGB2 and perinuclear enrichment of acetylated tubulin. Similar to our zebrafish results, in human samples the invasive front was characterized by the presence of elongated nuclei with high levels of HMGB2 and acetylated tubulin expression. Of the 40 patient samples we analyzed, 20 samples (50%) contained putative interface cells (elongated nuclei, HMGB2+, AcTub+), 9 samples (22.5%) contained tumor cells exhibiting perinuclear acetylated tubulin enrichment only (AcTub+, HMGB2-), 3 samples (7.5%) contained tumor cells with elongated HMGB2+ nuclei, but no acetylated tubulin signal (AcTub-, HMGB2+), and the remaining 8 samples (20%) did not contain any tumor cells with enrichment of HMGB2 or acetylated tubulin. This result validates our scRNA-seq analyses that also indicated that interface cells are present in human samples (**Fig. 1c**). We added these data as **new Fig. S7**.

3. Figure 5 would benefit from a discussion of how *hmgba/hmbgb* knockout in zebrafish tumor altered invasiveness.

Response: We added a description of the effects of HMGB2 CRISPR on invasion to the text (lines 447-448) and representative images as **new Fig. 5m**.

Reviewer 2

This manuscript by Hunter and colleagues combines a zebrafish model of melanoma with an in vitro melanoma cell line to propose that mechanical stress to cells in the tumor's periphery induces a particular phenotype based on HMGB2 upregulation. This work is a nice continuation of their previous study (Hunter et al, Nat Comms 2021) where this "interface" melanoma subpopulation was identified and first characterised. Overall, this is an interesting study, taking a mechanobiology approach on a specific subset of melanoma cells that can be of high therapeutic interest. Data are well represented, the text is very clearly written, and the approach to the problem interesting in itself.

However, I find that both the mechanistic and the molecular insights provided by the authors fall short from achieving three important things: First, from identifying the actual molecular components of the pathway that triggers HMGB2 upregulation; second, from identifying the mode-of-action via which HMGB2 gives rise to this particular phenotype (if this is indeed chromatin-bound HMGB2-driven remains unclear on the basis of the data); third, from identifying a druggable target in any of the above pathways that would render their observation clinically actionable (even at the level of of an in vivo model).

Response: Thank you for these constructive suggestions. We have extensively revised the manuscript to address these points.

Regarding the first point, we agree that the molecular mechanism connecting mechanical stress to HMGB2 was unclear. To address this, we utilized the TurboID proximity labeling system to identify protein interactors of HMGB2 in melanoma cells (**new Fig. 3I**). We generated stable A375 cell lines expressing either HMGB2-TurboID or NLS-TurboID fusion proteins, pulsed the cells with 10 mM biotin for 1 hour to induce biotinylation of proteins proximal to the TurboID cassette, and quantified enrichment of biotinylated proteins using mass spectrometry. To identify proteins that specifically interact with HMGB2 above the 'background' of the overall nuclear proteome, for each protein detected we calculated fold enrichment in HMGB2-TurboID relative to NLS-TurboID. One highly enriched protein was nesprin-2 (gene name: *SYNE2*), a component of the LINC (Linker of Nucleoskeleton and Cytoskeleton) complex that physically connects the cytoskeleton, nuclear lamina and chromatin (**new Fig. 3m**). We hypothesized that nesprin-2 cooperates with HMGB2 to respond to mechanical force and stabilize the nucleus against mechanical stress. Accordingly, confined A375 cells upregulated nesprin-2 (**new Fig. S13a-b**), and targeting *SYNE2* with siRNA (**new Fig. S13c**) abolished confinement-mediated accumulation of HMGB2 (**new Fig. 3n-o**) and formation of the perinuclear acetylated tubulin network (**new Fig. 3p-q**). Finally, we characterized how the acetylated tubulin/nesprin-2/HMGB2 interaction affects nuclear stiffness in confined cells, as the LINC complex is known to cooperate with the nuclear lamina to tune nuclear stiffness. We quantified a ~3-fold increase in lamin A/C in confined cells ($P = 1.28e-162$; **new Fig. S13d-e**), suggesting that the nuclear lamina is remodeled in response to confinement. To further validate these results, we used atomic force microscopy (AFM) to directly measure the nuclear stiffness of confined melanoma cells. In concordance with the upregulation of acetylated tubulin, nesprin-2, and lamin A/C we observed in confined cells, nuclear stiffness also significantly increased (**new Fig. S13f-g**). Together, our new results suggest that confined melanoma cells remodel their cytoskeletal and nuclear architecture to reinforce the cell against mechanical force, resulting in LINC complex-mediated upregulation of HMGB2 and a stiffer nucleus.

Regarding the second point, we have now further clarified the mechanism by which HMGB2 promotes phenotype switching. As you mention, HMGB proteins can mediate their effects either through interactions with chromatin/DNA, or as more recently described through interactions with RNA. Our initial FRAP and ATAC-seq data suggested a role for increased chromatin accessibility in response to HMGB2 upregulation, but we agree that more direct evidence for this mechanism was needed.

First, to identify HMGB2 target genes in melanoma, we performed ChIP-seq targeting HMGB2 in A375 cells, using your lab's published and established protocol for ChIP-seq of HMGB-family proteins (PMID: 29706538). We first did this in A375 cells expressing wild-type levels of HMGB2, as well as in a HMGB2 knockout cell line (**new Fig. S10a-c**) to confirm target specificity (**new Fig. S14a**). We generated a peak atlas by comparing peaks present in A375 cells expressing baseline levels of HMGB2 to those not present in A375-HMGB2^{KO}

cells. This approach resulted in 843 peaks, in line with previous studies of HMGB2 targets in other cell lines (PMIDs: 29706538, 27799366, 34215724). We then manually filtered the list of peaks to remove intergenic, low-quality, or otherwise non-specific peaks, resulting in a final conservative, high-confidence set of 96 HMGB2 target genes (**new Table S6** and **Fig. S14b**). Within this set, we observed binding of HMGB2 at the promoter of neuronal genes, including *NOTCH2*, *NOTCH2NLC*, *TBX6*, *GBA1*, and *ZNF335* (**new Fig. S14d** and **Table S6**), as well as pro-tumorigenic genes such as *KMT2A* (**new Fig. S14e** and **Table S6**). We also noted binding at the promoter region of several genes associated with AP-1 signaling (i.e. *FOSL1*, *JUNB*, and *JUND* (**new Fig. S14c** and **Table S6**), a transcription factor tightly linked to the melanoma invasive state (PMIDs: 25865119, 35926467). Thus, this ChIP-seq experiment validated our previous RNA-seq (**Fig. S15**) and ATAC-seq (**Fig. 4g-k**) datasets that indicated that upregulation of HMGB2 promotes a pro-invasive neuronal phenotype.

We next wished to extend these observations to an additional melanoma cell line capable of phenotype switching. It has long been observed that phenotype switching in melanoma is controlled by a regulatory axis of expression of melanocytic (e.g. *MITF*) versus invasive transcription factors (e.g. *BRN2*, *AP1*, *SOX9*). Cells with high levels of melanocytic TFs tend to be proliferative but not invasive, whereas those with high levels of invasive TFs slowly proliferate and become invasive. We therefore wanted to more directly assess whether HMGB2 was sufficient to induce the invasion program by modulating expression of these transcription factors or their targets. To test this, we utilized the SKMEL5 human melanoma cell line, which is typically characterized by high expression of melanocytic factors such as *MITF* but can be induced to become invasive. For this experiment, we used an SKMEL5 cell line that we had generated that stably expressed a V5-tagged form of HMGB2. To perform the ChIP experiment as robustly as possible, we decided to use both a HMGB2 antibody and a V5 antibody, and compare the signal from each to generate the peak atlas. As the V5 antibody provided a “cleaner” signal that allowed us to more clearly define HMGB2 targets (**new Fig. S16a-b**), we used the V5 signal to generate the subsequent peak atlas, comparing targets to the empty vector control. We then filtered the list to keep only peaks found within 1 kb of the associated gene’s TSS, yielding a set of 1361 peaks corresponding to 1286 unique target genes (**new Fig. S16b** and **Table S8**).

Interestingly, we identified binding by HMGB2 to the *MITF* promoter region (**new Fig. S16c**), but found by RNA-seq that loss of HMGB2 did not affect *MITF* expression at the transcriptional level ($\log_2FC = 0.157$; **new Fig. S16e-g**), suggesting that HMGB2 might instead be positively regulating pro-invasive transcription factors rather than negatively regulating melanocytic TFs. We found that HMGB2 robustly bound the promoter of genes associated with Notch signaling, not only in A375 cells (as mentioned above, **new Fig. S14d** and **Table S6**) but also in the more plastic SKMEL5 cell line (**new Fig. S16d** and **Table S8**). Notch-family genes were also upregulated by human interface cells (*DLL1*, *DLL3*, *DLK2*; **Table S1**) and confined human melanoma cells (*NOTCH2NLA*, *DLK2*, *DLL4*; **Table S2**). Notch signaling was recently found to drive melanoma invasion through a reciprocal relationship with the pro-invasive transcription factor brain-2 (*BRN2*, also known as *POU3F2*; PMID: 18983536), one of the most well known pro-invasive transcription factors in melanoma (PMID: 29781575). In addition to its role in invasion, *BRN2* also has a well-characterized role in neuronal development and has been referred to as a master neural transcription factor (PMID: 27784708). This suggests that both the invasive and the neuronal phenotypes we identified in confined interface cells are likely mediated by HMGB2-mediated upregulation of Notch/*BRN2* signaling. In support of this model, a *BRN2/POU3F2* motif was enriched in the promoter region of HMGB2 target genes identified from ChIP-seq (**new Fig. S16h**). While we cannot rule out a role for other chromatin-independent functions of HMGB2 in phenotype switching, our ChIP-seq experiments demonstrate that HMGB2/chromatin interactions are capable of activating the invasive neuronal signature through these mechanisms.

Finally, regarding the third point around druggability: the editor, Dr. Marte, specifically mentioned in her letter to us that she felt this was outside of the scope of the present manuscript. However, in response to another reviewer, we did perform mouse xenografts in which we overexpressed HMGB2 and tested sensitivity to BRAF/MEK inhibitors (a mainstay of melanoma therapy), and found that HMGB2 makes the cells more drug resistant (**new Fig. 5n-o**). This finding opens up the possibility that drugs currently being tested to overcome BRAF/MEK resistance (i.e. HGF inhibitors, immunotherapy) could have utility in HMGB2-overexpressing tumors.

Nevertheless, I hope that the more detailed comments I am offering below will be of some use to the authors.

Secondary comments:

- UMAP representation of scRNA-seq data can be highly misleading due to how UMAP directionality reduction is performed. I would advise the authors to use tSNE representations instead.

Response: We replotted all of the data represented in UMAPs in the manuscript as tSNE and added these plots in **new Figure S1**.

- The elongated nuclei in Fig. 1j,k are not necessarily also more flat, an attribute that can only be shown via a 3D imaging approach.

Response: We agree that the use of “flattened” was inaccurate here and have removed all such references from the manuscript.

- The similarity claimed for gene expression changes in melanoma cells lines under mechanical confinement and the melanoma “interface” cells in vivo is not sufficiently back up by GO term analysis in Fig. 1n. The authors should provide 1-to-1 correlation of gene expression changes between the in vitro and the in vivo model (using central melanoma cells as control for the latter comparison). Then we can see which genes are indeed convergently and which divergently regulated.

Response: We performed this analysis as suggested, using the Jerby-Arnon human melanoma scRNA-seq dataset to calculate a list of interface markers relative to the rest of the tumor cells (listed in Table S1), and compared this list to the confined cell markers found in Table S2. We filtered each list to only include genes with an adjusted P-value < 0.05, and then calculated a list of overlapping genes that were significantly up- or downregulated in both datasets ($n = 649$ genes; **new Fig. S3a**). As requested, we have provided a 1:1 comparison of log2FC for each co-expressed gene in the human interface and confined datasets as **new Table S3**. To determine if the co-regulated genes support the emergence of a neuronal phenotype in confined interface cells, we performed pathway analysis on the list of concurrently upregulated genes and found that the co-upregulated genes were enriched in those related to neuronal development and architecture (**new Fig. S3b**). This result suggests that both confined melanoma cells and interface cells from human samples upregulate a common set of genes associated with a neuronal phenotype.

We note that we also identified a number of genes that were not co-expressed by both datasets. While these genes may provide important insight into biological processes active in interface cells distinct from the mechanoreponse, we feel that with the current data, we are unable to determine whether these divergently regulated genes represent biological or technical variation. Technical differences between the two datasets can be attributed to (1) the inherent differences between single-cell and bulk RNA-seq data, where single-cell datasets typically have high dropout rates and relatively shallow sequencing depth; (2) increased noise between datasets due to batch effect; (3) the differences between a cultured immortalized cell line versus cells from patients. For these reasons, we have highlighted only the co-upregulated genes in the manuscript. We also note that some authors in the field generally discourage 1:1 comparison between genes across datasets for the reasons described above, instead focusing on network- or pathway-level analyses (see PMID: 20005852).

- It would be important to understand the extent of HMGB2 upregulation due to mechanical stress. The data presented in Fig. 3a,c,f,j do not make this clear. For example, scaled expression in Fig. 3a is not immediately interpretable (can we see counts and log2FC?),

Response: we have added plots showing normalized expression of *hmgb2a* and *hmgb2b* on a per cell level as **new Fig. 3b**. We calculated a log2FC for *hmgb2a* relative to the tumor of 1.57 ($P = 1.78e-87$) and log2FC for *hmgb2b* relative to the tumor of 1.53 ($P = 1.03e-98$) and added these values to the plots in **Fig. 3b**. In addition, the raw counts are shown below in **Fig. R2**.

Figure R2: Raw scRNA-seq counts for hmgb2a and hmgb2b in zebrafish melanoma cells. Each data point represents an individual cell.

...while in Fig. 3g the high-low intensity scale seems arbitrary.

Response: While the high-low scale in Fig. 3g is indeed arbitrary, the increase in HMGB2 levels upon compression is quantified in **Figs. 3i and 3k**. We also quantified HMGB2-GFP increase over time in compressed cells using live imaging, which is shown and quantified in **Fig. S6a-b**. We feel that these complementary approaches show that HMGB2 levels increase in confined cells.

Then, looking at Fig. 3h-j (where I am not sure what the Hoechst intensity increase really means, cannot be simply compaction), the effect seems to be <2-fold increase. If this is not also reflected in the RNA-seq data, then is it rather protein stabilization?

Response: We specifically used Hoescht staining because we felt it likely that there would be some degree of nuclear compaction, and that this needed to be accounted for when quantifying HMGB2 levels. Although it is true that things other than compaction could account for the increase in Hoechst signal here, this was the most reasonable method we could find to quantify compaction. Once we had that measurement, we were then able to normalize the HMGB2 levels to the Hoechst signal, which is what is shown in Fig. 3k. This shows that the increase in HMGB2 is greater than the increase in Hoechst signal, suggesting the increase in HMGB2 here is not solely attributable to compaction of chromatin. We clarified the details of this analysis in the text (lines 217-220). As to your point about stabilization, we agree, and this is what the FRAP data in Fig. 4a-f suggests, in which the increase in HMGB2 upon compression is due to stabilization of interactions of HMGB2 with chromatin.

Panel 3f would suggest a less steep increase in the model though. It would also be important to understand how HMGB2 was prioritized in the first place amongst many other (possibly even stronger upregulated) factors?

Response: HMGB2 is a protein that we have been interested in since our 2021 publication where we first uncovered the interface identity (Hunter et al., *Nat Comms* 2021; PMID: 34725363), and we highlighted HMGB2 as a notable interface marker in that paper, where we showed in that study that HMGB2 was highly upregulated by interface cells identified using spatial transcriptomics, single-nucleus RNA-seq, and single-cell RNA-seq (Figs. S10d and S11 in Hunter et al., 2021). To better explain why and how we prioritized HMGB2 as an interface marker, we added a description of our previous results to the text (lines 205-206). We acknowledge that there are many other genes upregulated in the interface which likely also have important roles in melanoma progression, and characterizing these will be a focus of a future study.

- Moreover, in Fig. 3c the "interface" levels should be compared not to bulk tumor but to the central, unconfined part only.

Response: The data presented in the previous Fig. 3c (now Fig. 3d) is human scRNA-seq data, so this type of data does not allow us to do comparisons based on spatial organization. However, we have now added new imaging data (**new Fig. S7**) where we identified HMGB2+ and acetylated tubulin+ interface cells in human samples. We stained human melanoma tissue microarrays (TMAs) to look for evidence of interface cells,

characterized by elongated nuclei with high levels of HMGB2 and perinuclear enrichment of acetylated tubulin. Similar to our zebrafish results, in human samples the invasive front was often characterized by the presence of elongated nuclei with high levels of HMGB2 and acetylated tubulin expression (**new Fig. S7a-c**), although as expected there was variability across patients. Of the 40 patient samples we analyzed, 20 samples (50%) contained interface cells (elongated nuclei, HMGB2+, AcTub+), 9 samples (22.5%) contained tumor cells exhibiting perinuclear acetylated tubulin enrichment only (AcTub+, HMGB2-), 3 samples (7.5%) contained tumor cells with elongated HMGB2+ nuclei, but no acetylated tubulin signal (AcTub-, HMGB2+), and the remaining 8 samples (20%) did not contain any tumor cells with enrichment of HMGB2 or acetylated tubulin (**new Fig. S7d-e**). Thus, this result validates the scRNA-seq data shown in Fig. 3d (and Fig. 1c) indicating that interface cells are present in human samples.

Moreover, the UMAP plot in Fig. 3b suggests that expression is not really higher in a per cell level in "interface" cells, but rather that most cells show this increase. The authors should therefore ask whether HMGB2-high cells are actually selected for this spatial destination by looking at the evolution of the tumor from earlier stages until this one here in their fish model.

Response: We agree that the idea that HMGB2-high interface cells are pre-selected at earlier stages via tumor evolution is an intriguing one. However, we feel that this is unlikely for several reasons. First, our data showing that HMGB2 increases in direct response to compression (**Figs. 3h-i** and **S6a-b**) suggests that the HMGB2-high cells at the interface are not pre-selected, but rather are induced by mechanical pressure at the invasive front. Our TMA imaging data in **new Fig. S7** (described above) also shows the same correlation between nuclear shape and HMGB2 upregulation at the invasive front that we observed in the zebrafish, further supporting a role for mechanically-induced nuclear deformation in upregulating HMGB2. Adding further evidence to this is our new data, described above, that links mechanical force to HMGB2 upregulation via the LINC complex (**new Figs. 3l-q** and **S13**).

We are not entirely sure what you meant in that the data in previous Fig. 3b (now **Fig. 3c**) shows that most cells demonstrate increased expression of *HMGB2*. In the UMAP and the corresponding violin plot in **Fig. 3d**, we demonstrate very low levels of expression of *HMGB2* in the bulk of the tumor in human samples. The zebrafish data plotted in **Fig. 3a**, **new Fig. 3b**, and **Fig. R2** also show low expression of zebrafish *hmgb2a/hmgb2b* in the bulk of the tumor as well. We are happy to further address this point with additional clarification.

- Please note that what appears to be nucleolar staining in Fig. 3g (which is the main point of fluorescent intensity increase) is an artefact of formaldehyde fixation of HMGBs in general. It is important that this experiment is repeated using more suitable fixation parameters (see Zirkel et al, Mol Cell 2018 and Mensah et al, Nature 2023).

Response: Thank you for this suggestion. We are using very similar and, often, identical fixation protocols to those in the cited papers. For fixation of cultured cells, we typically use 4% PFA for 15 minutes at room temperature as described in the Methods. For cultured cells, Zirkel et al. also used 4% PFA for 15 minutes at room temperature, and Mensah et al. used 4% PFA for 10 minutes at room temperature. Thus, our fixation protocol is in line with those used by others in the field. Additionally, while we agree that we do see some nucleolar localization of HMGB2 in fixed cells, we also generated a stable A375 cell line expressing GFP-tagged HMGB2, and used live imaging to show that the GFP-tagged protein increases in confined cells (**Fig. S6a-b**). Thus, we feel that the live imaging data, in particular, convincingly shows confinement-induced upregulation of HMGB2 is not attributable to fixation artifacts.

- I would suggest caution concerning the acetylation modulating experiments. HMGBs in general are known to be post-translationally modified by acetylation at various sites along their length, which can change their DNA binding and localization properties. It would therefore be important to control for this in all these experiments using tubacin/taxol/NZ.

Response: We agree that it is important to consider the role of acetylation in HMGB2 localization and behavior, especially in cases where we are broadly manipulating the acetylation state of the cell. We should clarify that

we specifically used taxol and nocodazole because they act on tubulin through *acetylation-independent* mechanisms: nocodazole binds to tubulin monomers and blocks assembly of tubulin filaments, whereas taxol binds tubulin filaments and prevents them from disassembling. Thus, taxol and nocodazole should not affect HMGB2 acetylation. Tubacin is a highly specific inhibitor of HDAC6, the major tubulin deacetylase. We could not find any published evidence of HDAC6 acting on HMGB2, and HDAC6 was not identified as a potential interactor with HMGB2 in our TurboID experiment (**new Table S4**). We also note that HDAC6 is localized primarily to the cytoplasm (PMID: 10873806), and we showed that tubacin treatment did not affect histone acetylation (**Fig. S11**), both of which indicate that tubacin is unlikely to target the largely nuclear pool of HMGB2. We also note that treating confined cells with taxol vs tubacin gave nearly identical results (**Fig. S12a-c**), and since taxol is acetylation-independent, this suggests that our tubacin results are likely not attributable to off-target effects on HMGB2, but rather specific effects on tubulin stability. While we can never fully rule out the possibility of acetylation-mediated effects, our approach was specifically designed to mitigate these effects as much as possible. We do feel it was important to address this in the manuscript, so we modified the text to note potential off-target effects of tubacin on HMGB2 acetylation (lines 282-284), and emphasize the acetylation-independent effects of taxol and nocodazole (lines 278-279 and 289-290).

In data not included in the manuscript, we did observe significant relocalization of HMGB2-GFP in response to HDAC inhibition via treatment with the pan-HDAC inhibitor trichostatin A (TSA; **Fig. R3**), indicating, as you suggested, that hyperacetylation of HMGB2 could influence its localization in melanoma cells. We did not include these data in the manuscript as we were concerned that, as you mentioned, this effect may be due to TSA-induced hyperacetylation of HMGB2 in addition to TSA's effects on tubulin acetylation. Below, we present side-by-side results showing the dramatic effect of TSA (far right) on HMGB2-GFP localization compared to the minimal change in localization we observed with tubacin or taxol, again suggesting that tubacin and taxol do not significantly influence the acetylation state and localization of HMGB2.

Figure R3: Trichostatin A (TSA) treatment causes HMGB2 to become cytoplasmic. Time-lapse images of HMGB2-GFP expression in confined human melanoma cells, treated with tubacin (HDAC6 inhibitor), taxol (acetylation-independent tubulin stabilizer), or TSA (pan-HDAC inhibitor). Time post-confinement is indicated. Scale bars, 25 μ m.

- I am probably misunderstanding something in the interpretation of the data in Fig. S5. Since early NZ treatment does abolish the perinuclear MT mesh, but accumulation persists, wouldnt this suggest that the two events are not coupled? The authors seem to suggest otherwise in the main text.

Response: We acknowledge that our wording may have been confusing here. NZ treatment abolishes the bulk MT network with the exception of the perinuclear acetylated tubulin mesh, as shown in **Fig. 2i-j** (bottom right

panels) where the perinuclear network is intact in NZ-treated cells. We have modified the text to better clarify this point (lines 293-294).

- The FRAP analysis, at best, shows some mild increase in chromatin association with HMGB2. I am also not sure how the change of nuclear volume/shape during confinement might affect FRAP outcomes? Therefore, to convincingly show such an effect, one would need to perform ChIP-seq (with spike-ins) to quantify binding patterns genome-wide.

Response: We agree that it is likely that the change in nuclear volume and shape during confinement is influencing HMGB2's dynamics within the nucleus. In fact, we feel this is likely the main factor causing a more 'crowded' environment in confined cell nuclei, which is most likely impairing HMGB2 diffusion and leading to more stable interactions with chromatin. This is in addition to the well known role for HMGB2 in binding chromatin under mechanical strain (PMIDs: 11246022, 8339930), which we would also expect to be increased in confined cells. Thus, the changes in nuclear architecture in confined cells are very likely to influence HMGB2's dynamics here. To clarify this point, we added a description of these potential effects to the text (lines 331-333).

As noted above, we have now performed ChIP-seq to assess HMGB2 binding in A375 and SKMEL5 cells. We have also performed ChIP-seq of HMGB2 in confined and unconfined A375 cells using *Drosophila* chromatin as a spike-in, as you suggested, to quantify the amount of HMGB2 bound genome-wide. This was a very technically challenging experiment. The confinement device is very small, limiting the number of cells we could process for ChIP-seq, in addition to the inherent limitations of performing ChIP on a protein that binds DNA significantly less strongly than a typical transcription factor (discussed in PMID: 29706538). While there was some variability across replicates, likely for these technical reasons, we found that confined cells exhibit increased binding of HMGB2 genome-wide (**Fig. R1**).

Figure R1: Differentially enriched HMGB2 targets in confined A375 cells. Number of HMGB2 target peaks quantified in confined cells relative to unconfined cells.

The HMGB2-OE experiments are welcome, but in the absence of an HMGB2 motif, it cannot be deduced that it is this factor that directly associates with the de novo opened positions. In fact, HMGBs have been shown to also bind RNA pretty specifically and this might explain more of the phenotype that chromatin effects (see Sofiadis et al, Mol Syst Biol 2021 for an example of HMGB1).

Response: We completely agree that we cannot rule out a role for HMGB2 in binding RNA on the basis of our current data. In the present manuscript, we have chosen to focus on HMGB2's interactions with chromatin for several reasons. First, HMGB2's well-characterized role in alleviating mechanical strain on chromatin has relevance to nuclear mechanobiology, a major focus of the current study. Second, we have now performed ChIP-sequencing targeting HMGB2 in two different melanoma cell lines as well as in confined cells (**new Figs. S14 and S16**). These ChIP experiments show that HMGB2 is binding DNA in melanoma cells, including in the promoter regions of several genes critical for melanoma tumorigenesis, plasticity and invasion.

While HMGB2's reported interactions with RNA are likely relevant in melanoma, we feel that characterizing how HMGB2 interacts with RNA in melanoma, and the biological relevance of these interactions, is outside the scope of the current study and would likely take us several years to thoroughly explore. We have added text to the Discussion section acknowledging that the HMGB2-RNA interaction may be relevant here (lines 507-511).

- The final result of the manuscript is intriguing, as higher levels of HMGB2 across multiple different types of cancer actually correlate with increased proliferation and anti-senescent gene expression programs. It is therefore surprising to see these anti-proliferative effects, and can only hypothesize that the mechanical stress of confinement (somehow) imposes on the usual HMGB2 effects. However, the authors do not dissect how this might occur and only present us with the observation.

Response: We agree that this result was surprising, and that our data indicate that the mechanical stress of confinement is overpowering HMGB2's typical pro-proliferative effects to drive invasion. We feel this is related to the "go or grow" phenotype switching paradigm which is well known in melanoma (shown in the schematic in **Fig. 5a**), where tumor cells can be either invasive or proliferative, but not both - if you gain one phenotype you lose the other (PMIDs: 18245463, 27124452, 32753671). As mentioned above, phenotypic plasticity in melanoma is controlled by a regulatory axis of expression of melanocytic (e.g. *MITF*) and invasive transcription factors (e.g. *BRN2*), where high expression of melanocytic TFs and low expression of invasive TFs drive proliferation, and high expression of pro-invasive TFs and low expression of melanocytic TFs promotes invasion. Our ChIP-seq data, described above, suggests that HMGB2 promotes invasion by activating Notch/BRN2 signaling. While our data also shows that HMGB2 binds the *MITF* promoter, which we initially thought may be indicative of transcriptional repression of *MITF* promoting anti-proliferative effects, loss of HMGB2 did not affect *MITF* expression at the transcriptional level ($\log_2FC = 0.157$; **new Fig. S16g**), suggesting the anti-proliferative effects of HMGB2/confinement are not mediated solely by repression of the *MITF* transcriptional program. However, our new data that shows activation of pro-invasive Notch/BRN2 signaling by HMGB2 suggests that this invasive program essentially tips the scales towards an invasive phenotype and away from a proliferative phenotype.

Reviewer 3

Hunter et al use a zebrafish model of melanoma to study the mechanobiology of a subset of rare cancer cells localized at the interface between the tumor and its surroundings. These cells display high upregulation of genes characteristic of an invasive state as well as markers of neuronal development, a profile that is also found in published datasets of human melanoma patients. The authors show that these cells display highly elongated nuclei, which leads them to hypothesize that nuclear deformation plays a key role in defining the cellular transcriptional state and function. To test this hypothesis, they subject a micropatterned human melanoma cell line to compressive stress and find that this confinement is sufficient to express the neuronal pathways that were upregulated in interface cells. In addition, they show that both interface cells in vivo and compressed cells in vitro assemble a network of stable acetylated microtubules at the nuclear periphery. This type of nuclear cage has been observed previously in other cell types in response to mechanical compression and is thought to shield the nucleus from mechanical stresses during migration through narrow pores. The authors then identify that, as in other cancer types, the high mobility group (HMG)-family proteins (mainly HMGB2) is upregulated in interface cells. This protein, which bends DNA in a sequence unspecific manner for transcription factor binding, increases its expression in response to compression in a way that is linked to microtubule stability. Using FRAP experiments, the authors identify a significant increase in the bound fraction of HMGB2 in confined cells. Using ATAC-seq, they show that upregulation of HMGB2 increases chromatin accessibility at neuronal loci, promoting the expression of genes that have been associated with mesenchymal migration. They also show in vitro and in vivo that HMGB2 contributes to phenotypic switching between proliferative and invasive phenotypes. Finally, they report that the confined cells are resistant to taxol-induced cell death, providing a link between confinement and therapy.

Overall, this is an insightful paper that connects a mechanobiological feature (nuclear deformation) with phenotypic switching in melanoma. Mechanisms are dissected in detail by combining in vivo zebrafish work and in vitro biophysical assays. The paper is clearly written and the narrative is solid, but some conclusions are insufficiently supported by the experiments. My major points are the following:

1) Nuclear deformation has been previously associated with nuclear translocation of transcription factors and regulators. YAP is the best known of them, but others associated with EMT have also been shown to be mechanosensitive (twist, snail, smad3, etc). Translocation of these factors might have a relevant role in phenotypic switching and in driving melanoma invasion. The authors should consider this possibility in their model and test it experimentally.

Response: We were initially surprised that some of these factors did not emerge in our original RNA-seq and ATAC-seq analysis. But we wanted to further test this, so we have now performed staining of confined and unconfined cells with antibodies targeting YAP, Twist, Snail, and SMAD3. Consistent with our initial observations, none of these proteins exhibited nuclear translocation upon confinement, suggesting they might not be key factors in this setting. We added this data as **new Fig. S9**.

2) Compression of cancer cells is sufficient to drive their migration independently of transcriptional changes, see Lomakin et al (Science, 2020), Venturini et al. (Science, 2020), Conti et al (biorxiv). The authors should rule out that these acute changes do not dominate the phenotypic switching they report. The proposed mechanism is compelling, but how dominant is it? Can the authors measure cell migration during compression as in the papers cited above?

Response: We quantified acute migration of compressed A375 cells and measured (as our videos also show) very little effect of confinement on cell velocity over the time of compression (mean velocity = 0.0838 ± 0.0025 $\mu\text{m}/\text{min}$; $n = 38$ cells). We added these analyses to the manuscript as **new Fig. S17**. These results are in line with the migration speeds of confined non-amoeboid cells measured by, for example, Venturini et al. who found that confined non-amoeboid cells migrate at speeds of $\sim 0\text{-}1$ $\mu\text{m}/\text{min}$, relative to confined amoeboid cells that migrated at ~ 6 $\mu\text{m}/\text{min}$ (Fig. S2G in Venturini et al., 2020). Given the importance of the seminal papers you have cited (and which have served as an inspiration for our own work), it is likely that differences in the conditions between our work and theirs might help explain the discrepancy. In the Lomakin et al., Venturini et al., and Conti et al. papers, they specifically plated their cells on extremely low adhesion PEG-coated

substrates, whereas in all of our experiments we plated our cells on high adhesion fibronectin-coated substrates. We felt that in our particular situation, this was a better mimic of the *in vivo* microenvironment faced by the interface cells.

3) The relevance of the authors' findings in the advance of cancer cell biology is clear, but its direct impact on human cancer is not. The authors should show that some of the key features of the mechanobiology of interface cells are present in human (or at least mouse) melanoma. For example, they should show that the correlation between nuclear shape, MT perinuclear localization, and HMGB2 overexpression found in zebrafish are also present in human/mouse samples.

Response: We examined human melanoma tissue microarrays (TMAs) to look for evidence of interface cells, characterized by elongated nuclei with high levels of HMGB2 and perinuclear enrichment of acetylated tubulin. Similar to our zebrafish results, in human samples the invasive front was typically characterized by the presence of elongated nuclei with high levels of HMGB2 and acetylated tubulin expression. Of the 40 patient samples we analyzed, 20 samples (50%) contained interface cells (elongated nuclei, HMGB2+, AcTub+), 9 samples (22.5%) contained tumor cells exhibiting perinuclear acetylated tubulin enrichment only (AcTub+, HMGB2-), 3 samples (7.5%) contained tumor cells with elongated HMGB2+ nuclei, but no acetylated tubulin signal (AcTub-, HMGB2+), and the remaining 8 samples (20%) did not contain any tumor cells with enrichment of HMGB2 or acetylated tubulin. This result validates our scRNA-seq analyses that also indicated that interface cells are present in human samples (**Fig. 1c**). We added these data as **new Fig. S7**.

Related to this point about human relevance, we have now also performed mouse experiments to measure tumor growth over time in animals transplanted with human melanoma cells overexpressing HMGB2. Two findings from these experiments are relevant in this context. First, we find that the HMGB2 overexpressing cells exhibit increased tolerance to clinically used BRAF/MEK inhibitor combinations (dabrafenib and trametinib), which would be consistent with the more invasive state being linked to drug resistance (**new Fig. 5n-o**). Second, when we performed histology on xenografts with HMGB2 overexpression constructs, we observed a striking accumulation of acetylated tubulin in interface cells at the tumor border (**new Fig. S18j**).

4) Large nuclear deformations have been identified by pathologists for decades in different types of cancers. Are the authors' results specific to melanoma? This could be easily addressed using cells from other types of cancer and subject them to compression to assess the perinuclear accumulation of MTs and the overexpression of HMGB2.

Response: We agree and think this is an important point. To test this, we examined HMGB2 and acetylated tubulin levels upon confinement in the human pancreatic adenocarcinoma (PDAC) cell lines Panc-1 and MIA-PaCa-2, and the bladder cancer cell lines HTB-4 and HTB-9. In all 4 cell lines, confinement induced significant upregulation of HMGB2 and acetylated tubulin. These data indicate that mechanical force, and subsequent changes in cell state, are likely to be generalizable outside of melanoma. We added these data as **new Fig. S8**.

5) How the microtubule cage drives changes in expression of HMGB2 should be clarified. The authors carried out experiments showing that strengthening the microtubule cage leads to HMGB2 over-expression, but the mechanism is unclear. Can they also weaken the cage? Is the microtubule cage needed at all for HMGB2 over-expression?

Response: To weaken the microtubule cage, we generated two A375 melanoma cell lines in which the main tubulin acetyltransferase, *ATAT1*, was inactivated via CRISPR (**new Fig. S4a-b**). In both *ATAT1*^{KO} cell lines, CRISPR-mediated inactivation of *ATAT1* almost completely abolished acetylated tubulin in both unconfined and confined cells (in a mosaic fashion due to mosaic expression of the CRISPR-Cas construct), including the assembly of the perinuclear tubulin network (**new Fig. S4c-e**). This indicates *ATAT1* is the acetyltransferase that responds to mechanical stress in melanoma cells by stabilizing the tubulin cytoskeleton. We next quantified HMGB2 accumulation in response to confinement in *A375-ATAT1*^{KO} cells lacking acetylated tubulin. Unexpectedly, HMGB2 accumulation was not impaired in *ATAT1*^{KO} cells (**new Fig. S4c-d,f**), suggesting that acetylated tubulin is sufficient, but not necessary for enrichment of HMGB2 in response to confinement.

Given this finding, we wanted to further investigate the factors that may cooperate with the MT cytoskeleton to promote upregulation of HMGB2 in confined cells. To identify proteins that interact with HMGB2 in melanoma, we performed proximity labeling proteomics using the TurboID system (**new Fig. 3I**). We generated stable A375 cell lines expressing either HMGB2-TurboID or NLS-TurboID fusion proteins, pulsed the cells with 10 mM biotin to induce biotinylation of proteins proximal to the TurboID cassette, and quantified enrichment of biotinylated proteins using mass spectrometry. To identify proteins that specifically interact with HMGB2 above the 'background' of the overall nuclear proteome, for each protein detected we calculated fold enrichment in HMGB2-TurboID relative to NLS-TurboID. One highly enriched protein was nesprin-2 (gene name: *SYNE2*), a component of the LINC (Linker of Nucleoskeleton and Cytoskeleton) complex that physically connects the cytoskeleton, nuclear lamina and chromatin (**new Fig. 3m**). We hypothesized that nesprin-2 cooperates with HMGB2 to respond to mechanical force and stabilize the nucleus against mechanical stress. Accordingly, confined A375 cells upregulated nesprin-2 (**new Fig. S13a-b**), and targeting *SYNE2* with siRNA (**new Fig. S13c**) abolished confinement-mediated accumulation of HMGB2 (**new Fig. 3n-o**) and formation of the perinuclear acetylated tubulin network (**new Fig. 3p-q**). Thus, our results suggest that nesprin-2 and the acetylated tubulin network cooperate to promote upregulation of HMGB2 in confined cells.

6) The higher invasive capacity of HMGB2OE cells should be tested functionally (the snapshots of Fig S6 are insufficiently conclusive). There are many simple assays in vitro that can ascertain whether these cells are indeed more invasive.

Response: To address this, we have performed in vitro assays using the Cultrex Collagen I Cell Invasion Assay (see **Methods**) to measure the invasive capacity of HMGB2OE cells. A375 cells overexpressing HMGB2 were serum-starved overnight and subsequently allowed to invade through a collagen-coated insert for ~18 hours. Overexpression of HMGB2 significantly increased the invasive ability of melanoma cells. We added these data as **new Fig. 5h-i**.

Minor

1) Line 152: please correct "unconfined and unconfined"

Response: We made the indicated change to the text.

2) The authors qualify the interface cells are "rare". Can they provide a quantitative statement of how rare they are?

Response: We calculated that 12.1% (211 out of 1743 cells) of zebrafish tumor cells and 12.3% (269 out of 2187 cells) of human tumor cells adopted an interface identity, defined as relative expression of the interface gene signature identified in our previous paper (PMID: 34725363) above 0.75 on a scale of 0-1. We added these figures to the text (lines 62-63 and 70-71) and removed the word "rare" from the abstract.

Referee #1 (Remarks to the Author):

Mechanical confinement governs phenotypic plasticity in melanoma.
Hunter et al, 2023. Nature.

Summary: The revised manuscript by Hunter et al. has identified a transcriptional network driving a pro-invasion phenotype in melanoma that is mediated by HMGB2. This revision has strengthened the overall manuscript particularly the addition of multiple ChIP-seq experiments and refining the model of how HMGB2 functions. However, some concerns still arise, particularly regarding drug tolerance, and are detailed below. Upon resolution of these issues, this manuscript would be of broad interest to the readers of Nature and should be accepted.

Major:

1. The addition of the mouse experiment to address drug tolerance (Fig 5o), although appreciated, is not sufficient to conclude that HMGB2 leads to sustained drug tolerance. Individual spaghetti plots for each mouse in the cohort should be shown in supplemental to help clarify if it is a single mouse driving the increase in tumor volume or if it's a cohort effect. Further, it appears that at Day 25 both the control and HMGB2 OE mice are gaining tolerance, although HMGB2 OE is occurring at a faster rate.

We added a spaghetti plot showing tumor volume over time for each animal in the drug cohort as **new Fig. S20** (reproduced below as **Fig. R1**), and also added all of the raw data for each animal as **new Table S10**. As shown below, the effect of HMGB2 is seen across the cohort, extending out beyond day 25 to day 27-28, when we had to euthanize the mice at the humane endpoint. To measure differences in growth rates across groups, we performed a series of likelihood-ratio tests by fitting a biexponential model to each individual growth curve. Using this approach, we calculated a significant increase in growth rate in the OE dab/tram condition relative to the EV dab/tram ($P = 0.043$).

Figure R1. Tumor volume over time for A375-HMGB2EV and A375-HMGB2OE xenografts treated with dabrafenib/trametinib. $n = 11$ mice per condition across 2 replicates.

2. The co-expression of MITF and BRN2 is known to be an artifact of cell culture in comparison to 3D culture or human melanoma (see PMIDs: 19826052, 18829533, 25132268). In addition, BRN2 is a known repressor of MITF expression (see PMIDs: 18829533, 21435193).

a. The manuscript would benefit from staining human samples or harvested zebrafish tumors/mouse xenografts for MITF and BRN2 to show distinct localization.

We agree that *MITF* and *BRN2* are often inversely expressed in melanoma, although some studies have also shown that they can be co-expressed as well, including in 3D culture/spheroids (see Fig. S7A in PMID: 28119061 and Fig. 3A in PMID: 21358674), mouse xenografts (Fig. 3B in PMID: 21358674), and human samples (Fig. 3C in PMID: 21358674). To better understand this, we examined this question at both the RNA and protein levels.

We analyzed two well-known human patient scRNA-seq datasets from Jerby-Arnon et al., 2018 and Pozniak et al., 2024. In both datasets, we identified a subset of tumor cells (3-12%) that co-express both *MITF* and *BRN2* at the transcriptional level (**Fig. R2**).

Figure R2. Co-expression of *MITF* and *BRN2* in melanoma cells from human patients. Human melanoma scRNA-seq from Pozniak et al., Cell 2024 (a) and Jerby-Arnon et al., Cell 2018 (b).

To examine this at the protein level, we stained human melanoma samples with antibodies targeting MITF and BRN2. We identified subpopulations of tumor cells co-expressing MITF and BRN2 in each sample (**Fig. R3a**). We quantified co-expression by thresholding the intensity measurements, setting a threshold to classify a given cell as MITF+ and/or BRN2+ if intensity was above the mean of the given factor +1 standard deviation. Consistent with the RNA-seq, we found that 56/831 (6.7%) of cells co-expressed both MITF and BRN2 protein (**Fig. R3b**).

Figure R3. Co-expression of MITF and BRN2 protein in human tumor samples. *a. Representative images of MITF and BRN2 expression in human melanoma. Arrows indicate double positive cells. Scale bars, 25 μ m. b. Quantification of MITF and BRN2 intensity. Each data point represents a single cell. Dotted lines indicate thresholding cutoff (mean intensity + 1 standard deviation). 13 images from 3 tumors were quantified for a total of $n = 831$ cells.*

b. The discussion would benefit from more explanation of why MITF was not affected upon nuclear confinement with an upregulation of HMGB2 and BRN2 activity.

Thank you for this suggestion. Our data and the publications cited above indicate that while *MITF* and *BRN2* are often inversely expressed, they can also be co-expressed at both the transcriptional and protein levels within the same cell in 3D culture, mouse, and human samples. We added a discussion of this point to the manuscript text (lines 420-423).

Minor:

1. Size of the tumors upon treatment (Fig 5o) should be noted in the methods as they appear quite small.

We have included our dosing procedure in the “Mouse experiments” section within the Methods (lines 819-822):

Once tumors reached an average volume of 100 mm³, mice were randomized into two treatment groups (n = 4-6 mice/group) to receive either a vehicle control, or trametinib (1 mg/kg) in combination with dabrafenib (30 mg/kg).

We have also added all of the measurements for each individual mouse as **new Table S10**.

2. If collected, mouse xenografts should be stained with proliferation, pERK, and other known markers to be modulated following HMGB2 OE and drug treatment to show translation of the identified mechanism to the in vivo mouse model.

Unfortunately we did not collect these tumor samples so we are unable to perform the suggested experiment. We agree that characterizing the molecular mechanisms by which HMGB2 promotes drug tolerance is an important point, and this will be the focus of a future study.

Referee #2 (Remarks to the Author):

The authors have carefully addressed the majority of the points I raised. Although I still believe that the HMGB2-RNA interactions are equally important in the context of HMGB2 upregulation, I understand how this might be a challenging bit to investigate. Therefore, I am happy to suggest that the manuscript is published in its current form.

One minor note: I find the filtration of HMGB2 peaks to be excessively strict, thereby resulting in <100 targets. I would suggest that the authors consider a broader set in their analysis.

A. Papantonis

Thank you for these constructive comments. We think that the lower number of HMGB2 targets that we identified from our A375 ChIP experiment is likely due to the fact that we manually filtered the original list of 843 targets to include only high-quality peaks, as described in the text (lines 369-374):

We generated a peak atlas by comparing peaks present in A375 cells expressing baseline levels of HMGB2 to those not present in A375-HMGB2^{KO} cells. This approach resulted in 843 peaks, in line with previous studies of HMGB2 targets in other cell lines⁷⁹⁻⁸¹. We then manually filtered the list of peaks to remove intergenic, low-quality, or otherwise non-specific peaks, resulting in a final conservative, high-confidence set of 96 HMGB2 target genes (Table S6 and Fig. S14b).

However, to account for our stringent filtering we have now also added the original unfiltered set of 843 peaks to Table S6.

Referee #3 (Remarks to the Author):

The authors have addressed each of my comments thoroughly. While some of their responses reinforce their original conclusions, others suggest that the underlying mechanisms may be more complex than initially proposed. For example, the finding that acetylated tubulin is sufficient but not necessary for HMGB2 enrichment in response to confinement, along with the newly discovered involvement of nesprin, raises alternative mechanistic interpretations rather than the original linear argument linking the tubulin cage to HMGB2. Furthermore, the authors now show that transcription factors and regulators leave the nucleus in response to compression, opening the possibility of additional transcriptional mechanisms besides those studied here. I see this not as a weakness but as a strength. For instance, the translocation of transcription factors from the nucleus to the cytoplasm challenges current understanding in mechanobiology and will attract the interest of many research groups. Overall, I find that this article advances the fields of cancer cell biology and mechanobiology significantly and brings them further together.

Thank you for this positive feedback.

I recommend acceptance with just one minor comment.

1) In their discussion of Supp Fig. 9 the authors should emphasize that compression causes the translocation of transcription factors and regulators from the nucleus to the cytoplasm.

We clarified this point in the text (line 255).